# Computationally Efficient RL under Linear Bellman Completeness for Deterministic Dynamics

**Runzhe Wu**[*]
Cornell University
rw646@cornell.edu

**Ayush Sekhari**[*]
MIT
sekhari@mit.edu

**Akshay Krishnamurthy**
Microsoft Research
akshaykr@microsoft.com

**Wen Sun**
Cornell University
ws455@cornell.edu

## Abstract

We study computationally and statistically efficient Reinforcement Learning algorithms for the *linear Bellman Complete* setting. This setting uses linear function approximation to capture value functions and unifies existing models like linear Markov Decision Processes (MDP) and Linear Quadratic Regulators (LQR). While it is known from the prior works that this setting is statistically tractable, it remained open whether a computationally efficient algorithm exists. Our work provides a computationally efficient algorithm for the linear Bellman complete setting that works for MDPs with large action spaces, random initial states, and random rewards but relies on the underlying dynamics to be deterministic. Our approach is based on randomization: we inject random noise into least squares regression problems to perform optimistic value iteration. Our key technical contribution is to carefully design the noise to only act in the null space of the training data to ensure optimism while circumventing a subtle error amplification issue.

## 1 Introduction

Various application domains of Reinforcement Learning (RL)—including game playing, robotics, self-driving cars, and foundation models—feature environments with large state and action spaces. In such settings, the learner aims to find a well performing policy by repeated interactions with the environment to acquire knowledge. Due to the high dimensionality of the problem, function approximation techniques are used to generalize the knowledge acquired across the state and action space. Under the broad category of function approximation, model-free RL stands out as a particularly popular approach due to its simple implementation and relatively better sample efficiency in practice. In model-free RL, the learner uses function approximation (e.g., an expressive function class like deep neural networks) to model the state-action value function of various policies in the underlying MDP. In fact, the combination of model-free RL with various empirical exploration heuristics has led to notable empirical advances, including breakthroughs in game playing (Silver et al., 2016; Berner et al., 2019), robot manipulation (Andrychowicz et al., 2020), and self-driving (Chen et al., 2019).

Theoretical advancements have paralleled the practical successes in RL, with tremendous progress in recent years in building rigorous statistical foundations to understand what structures in the environment and the function class suffice for sample-efficient RL. These advancements are supported by optimal exploration strategies that align with the corresponding structural assumptions, and by now we have a rich set of tools and techniques for sample-efficient RL in MDPs with large state/action spaces (Russo & Van Roy, 2013; Jiang et al., 2017; Sun et al., 2019; Wang et al., 2020; Du et al., 2021; Jin et al., 2021; Foster et al., 2021; Xie et al., 2022). However, despite a rigorous statistical foundation, a significant challenge remains: many of these theoretically rigorous approaches

---

[*]Equal contribution.

for rich function approximation are not computationally feasible, and thus have limited practical applicability. For example, some require solving complex optimization problems that are computationally intractable in practice (Zanette et al., 2020b); others require deterministic dynamics and initial states (Du et al., 2020); and some methods depend on maintaining large and complex version spaces (Jin et al., 2021; Du et al., 2021) which are intractable in terms of memory and computation.

One of the most striking examples of this statistical-computational gap is observed in the *Linear Bellman Completeness* setting, which is perhaps one of the simplest learning settings. Linear Bellman completeness serves as a bridge between RL and control theory literature as it provides a unified framework to capture Linear MDPs (Jin et al., 2020; Agarwal et al., 2019; Zanette et al., 2020b) and the Linear Quadratic Regulator (LQR), two popular models in RL and control respectively. In particular, the linear Bellman completeness setting captures MDPs where the state-action value function of the optimal policy is a linear function of some pre-specified feature representations (of states and actions), and the Bellman backups of linear state-action value functions are linear (w.r.t. some feature representation). Naturally, for this setting, the learner utilizes the function class $\mathcal{F}$ consisting of all linear functions over the given feature representation as the value function class for model-free RL. In addition to considering a linear class, we also assume that the class $\mathcal{F}$ exhibits low inherent Bellman error—a structural assumption that quantifies the error in approximating the Bellman backup of functions within $\mathcal{F}$. The first assumption, i.e., linearity of optimal state-action value function, is perhaps the simplest modeling assumption one can make in RL with function approximation. Furthermore, emerging evidence suggests that linearity is practically useful, as with adequate feature representation, linear functions can represent value functions in various domains. The second assumption, i.e. low inherent Bellman error of the class, while being a bit mysterious, is a natural condition for statistical tractability for classic algorithms such as Fitted Q-iteration (FQI) and temporal difference (TD) learning with linear function approximation (Munos, 2005; Zanette et al., 2020b). It is also well-known that linearity alone does not suffice for efficient RL (Wang et al., 2021; Weisz et al., 2021).

While the prior works have shown that RL with linear bellman completeness is statistically tractable, and one can learn with sample complexity that scales polynomially with both $d$ and $H$ (where $d$ is the dimensionality of the feature representation and $H$ is the horizon of the RL problem), the proposed algorithms that obtain such sample complexity in the online RL setting are not computationally efficient. Given the simplicity of the problem, it was conjectured that a computationally efficient algorithm should exist. However, no such algorithms were proposed. Unfortunately, the classical approaches of combining supervised learning techniques with RL in the online setting, e.g., value function iteration, which are computationally efficient by design, fail to extend to be statistically tractable due to exponential blowups from error compounding, especially without making norm-boundedness assumptions. On the other hand, the techniques of adding quadratic exploration bonuses, e.g., the one proposed in LinUCB (Li et al., 2010) and used in LSVI for linear MDPs, also fail here as Bellman backups of quadratic functions are not necessarily within the linear class $\mathcal{F}$. In fact, the search for a computationally efficient algorithm with large action spaces is open even when the transition dynamics are deterministic.

In this work, we provide the first computationally efficient algorithm for the linear Bellman complete setting with deterministic dynamics, that enjoys regret bound of $\widetilde{O}(d^{5/2}H^{5/2} + d^2 H^{3/2} T^{1/2})$ for feature dimension $d$, horizon $H$, and number of rounds $T$. Importantly, our algorithm works with large action spaces, stochastic reward functions, and stochastic initial states. The key ideas of our algorithm are twofold: using *randomization* to encourage exploration and leveraging a *span argument* to bound the regret. While adding random noise to the learned parameters has been quite successful in linear function approximation, unfortunately, for our specific setting, since we need to add sufficiently large noise to cancel out the estimation error, blind randomization can cause the corresponding parameters to grow exponentially with the horizon. We avoid paying for this blow-up by only adding noise to the null space of the data. In particular, when the dynamics are deterministic, by adding exploration noise only in the null space, we can learn the value function exactly for any trajectories that lie within the span of the data seen so far. Additionally, a simple span argument bounds the number of times the trajectories fall outside the span of the historical data. Together, these techniques leads to our polynomial sample complexity bound. The resulting algorithm relies on linear regression oracles under convex constraints, which we show can be approximately solved via a random-walk-based algorithm (Bertsimas & Vempala, 2004).

## 2 RELATED WORKS

**Computational Efficient RL under Linear Bellman Completeness.** Numerous works have focused on computationally efficient RL within the scope of linear Bellman completeness (LBC). The simplest setting is tabular MDPs where computationally efficient and near-optimal algorithms have been well known (Azar et al., 2017; Zhang et al., 2020; Jin et al., 2018). Tabular MDPs can be extended to linear MDPs (Jin et al., 2020), where computationally efficient algorithms are also known (Jin et al., 2020; Agarwal et al., 2023; He et al., 2023). However, in the setting of linear Bellman completeness, which captures linear MDPs, the existence of computationally efficient algorithms remain unclear. Previous works have resorted to various assumptions to achieve computational efficiency, such as few actions (Golowich & Moitra, 2024) and assuming MDPs are "explorable" (Zanette et al., 2020c). We provide a detailed overview of the literature in Section 3.2.

**Exploration via Randomization.** Random noise has been a powerful alternative to bonus-based exploration in RL literature. A typical approach is Randomized Least-Squares Value Iteration (RLSVI) (Osband et al., 2016), which injects Gaussian noise into the least-squares estimate and achieves near-optimal worst-case regret for linear MDPs (Agrawal et al., 2021; Zanette et al., 2020a); Ishfaq et al. (2023) instead propose posterior sampling via Langevin Monte Carlo for Q-function and also obtain regret bounds for linear MDPs; Ishfaq et al. (2021) developed randomization algorithms for general function approximation assuming bounded eluder dimension and Bellman completeness for any function. Randomization is also explored in preference-based RL, leading to the first computationally efficient algorithm with near-optimal regret guarantees for linear MDPs (Wu & Sun, 2024). However, these approaches either have strong assumptions (e.g., Bellman completeness for any function), or inject random noise larger than the estimation error, causing exponential blow-up of parameter values—to mitigate it, they truncate the value, but this is feasible only in low-rank MDPs and challenging under linear Bellman completeness as the Bellman backup of truncated value may no longer be linear. Consequently, existing algorithms cannot handle linear Bellman complete problems, and new techniques capable of managing exponential parameter values are needed.

**Beyond Linear Bellman Completeness.** Many structural conditions capture linear Bellman completeness, such as Bilinear class (Du et al., 2021), Bellman eluder dimension (Jin et al., 2021), Bellman rank (Jiang et al., 2017), witness rank (Sun et al., 2019), and decision-estimation coefficient (Foster et al., 2021). While statistically efficient algorithms exist for these settings, no computationally efficient algorithms are known.

## 3 PRELIMINARIES

A finite-horizon Markov Decision Process (MDP) is given by a tuple $\mathcal{M} = (\mathcal{S}, \mathcal{A}, H, \mathsf{T}, r, \mu)$ where $\mathcal{S}$ is the state space, $\mathcal{A}$ is the action space, $H \in \mathbb{N}$ is the horizon, $\mathsf{T} : \mathcal{S} \times \mathcal{A} \to \Delta(\mathcal{S})$ is the transition function, $r : \mathcal{S} \times \mathcal{A} \to [0, 1]$ is the reward function and $\mu \in \Delta(\mathcal{S})$ is the initial state distribution. Given a policy $\pi : \mathcal{S} \mapsto \Delta(\mathcal{A})$, we denote $Q_h^\pi(s, a) = \mathbb{E}_\pi \left[ \sum_{i=h}^H r_i \,|\, s_h = s, a_h = a \right]$ as the layered state-action value function of policy $\pi$ and $V_h^\pi(s) = Q_h^\pi(s, \pi(s))$ as the state value function. The optimal value function is denoted by $V_h^\star(s) = \max_\pi V_h^\pi(s)$, and the optimal policy is $\pi^\star$.

We focus on the setting of linear function approximation and consider the following linear Bellman completeness, which ensures that the Bellman backup of a linear function remains linear.

**Definition 1** (Linear Bellman Completeness). *An MDP is said to be linear Bellman complete with respect to a feature mapping $\phi$ if there exists a mapping $\mathcal{T} : \mathbb{R}^d \to \mathbb{R}^d$ so that, for all $\theta \in \mathbb{R}^d$ and all $(s, a) \in \mathcal{S} \times \mathcal{A}$, it holds that*

$$\langle \mathcal{T}\theta, \phi(s, a) \rangle = \mathbb{E}_{s' \sim \mathsf{T}(s,a)} \max_{a'} \langle \theta, \phi(s', a') \rangle.$$

*Moreover, we require that, for all $h \in [H]$ and $(s, a) \in \mathcal{S} \times \mathcal{A}$, the random reward is bounded in $[0, 1]$ with mean $r_h(s, a) = \langle \omega_h^\star, \phi(s, a) \rangle$ for some unknown $\omega_h^\star \in \mathbb{R}^d$.*

We assume $\|\phi(s, a)\|_2 \le 1$ for all $s \in \mathcal{S}$ and $a \in \mathcal{A}$. Notably, we do not impose any upper bound on $\|\omega_h^\star\|_2$ or any $\ell_2$-norm non-expansiveness of the Bellman backup, distinguishing us from some existing works—in Section 3.1, we discuss why many existing definitions of linear Bellman completeness fail to capture even tabular MDPs or linear MDPs due to certain $\ell_2$-norm boundedness assumptions.

We further assume the feature space spans $\mathbb{R}^d$, i.e., $\text{span}(\{\phi(s,a) : s \in \mathcal{S}, a \in \mathcal{A}\}) = \mathbb{R}^d$; otherwise, we can project the feature space onto its span or use pseudo-inverse in the analysis when needed. We can verify that the linear Bellman completeness captures both linear MDPs and Linear Quadratic Regulators (for a convex subset of linear functions). The proof is in Appendix E.

Next, we consider deterministic state transition.

**Assumption 1** (Deterministic transitions). *For all $s \in \mathcal{S}$ and $a \in \mathcal{A}$, there is a unique state $s' \in \mathcal{S}$ to which the system transitions to after taking action $a$ on state $s$.*

We emphasize that, although the transition is deterministic, the initial state distribution is stochastic (although we assume that $\{s_{t,1}\}_{t \leq T}$ is independently sampled from an initial distribution $\mu$, our results extend to the scenarios when $\{s_{t,1}\}_{t \leq T}$ are adversarially chosen). Additionally, the reward signals can be stochastic. Hence, learning is still challenging in this case. The goal is to achieve low regret over $T$ rounds. The regret is defined as

$$\text{Reg}_T := \mathbb{E}\left[\sum_{t=1}^{T} \left(V^\star(s_{t,1}) - V^{\pi_t}(s_{t,1})\right)\right].$$

The expectation here is taken over the randomness of algorithm and reward signals. While it is defined as an average for simplicity, a concentration inequality can yield the high-probability regret. In this paper, we use asymptotic notations $\widetilde{\Theta}(\cdot)$ and $\widetilde{O}(\cdot)$ to hides logarithmic and constant factors.

### 3.1 OTHER LINEAR BELLMAN COMPLETENESS DEFINITIONS IN THE LITERATURE

Several closely related definitions of Linear Bellman Completeness have been considered in the literature. In the following, we demonstrate that some of these variant definitions face limitations due to additional $\ell_2$-norm assumptions. We present two commonly imposed assumptions in existing works below, and subsequently provide examples illustrating their potential limitations.

**(1) Assuming Bounded $\ell_2$-norm of Parameters.** Golowich & Moitra (2024); Zanette et al. (2020b;c) assume that any value function under consideration has its parameters bounded in $\ell_2$-norm, i.e., when we apply the Bellman backup, the resulting state-action value function always lies in $\{Q : Q(s,a) = \langle \phi(s,a), \theta \rangle, \|\theta\|_2 \leq R\}$ where $R$ is a pre-fixed polynomial in the dimension of the feature space. We will show that this assumption might not hold true since $\|\theta\|_2$ is unnecessarily bounded under linear Bellman completeness.

**(2) Assuming Non-expansiveness of Bellman Backup in $\ell_2$-norm.** Song et al. (2022) assume that, after applying the Bellman backup, the $\ell_2$-norm of the value function parameters will not increase, i.e., for any $\theta$, they assume the existence of parameter $\theta'$ such that $\|\theta'\|_2 \leq \|\theta\|_2$ and $\langle \phi(s,a), \theta' \rangle = \mathbb{E}_{s' \sim \mathsf{T}(s,a)} \max_{a'} \langle \phi(s',a'), \theta \rangle$ for all $s, a$. This assumption is stronger than the previous one and does not hold even in tabular MDPs, as we will show in the second example below.

The following example demonstrates that the two assumptions above do not generally hold under linear Bellman completeness as the $\ell_2$-norm amplification can actually be arbitrarily large.

**Example 1** (Arbitrarily Large $\ell_2$-norm on Parameters). *Consider a layered linear MDP with three states, $s_1, s_2, s_3$, and a single action $a_1$. Here $s_1$ is in the first layer and $s_2$ and $s_3$ are in the second layer. For some $\varepsilon$ and $p$, we define $\phi(s_1, a_1) = (\sqrt{\varepsilon}, \sqrt{p-\varepsilon})$, $\mu(s_2) = (p/\sqrt{\varepsilon}, 0)$, and $\mu(s_3) = (0, (1-p)/\sqrt{p-\varepsilon})$. We further define $r(s_2, \cdot) = \varepsilon$ and $r(s_3, \cdot) = 1$. We can verify that $P(s_2|s_1, a_1) = p$ and $P(s_3|s_1, a_1) = 1 - p$. Hence $Q(s_1, a_1) = p\varepsilon + 1 - p$. We assume Q-function is parameterized by $\theta$. Then, since $\|\phi(s_1, a_1)\| = p$, it must hold that $\|\theta\| \geq (p\varepsilon + 1 - p)/p = \varepsilon + p^{-1} - 1$. While $p$ can be arbitrarily small, the norm of $\theta$ can be arbitrarily large.*

We may hope to "normalize" the features in this example so that the $\ell_2$-norm of the parameters is bounded. However, it is unclear how to do so since changing either $\varepsilon$ or $p$ will change the MDP, and feature search is likely a hard problem. Essentially, this example breaks one of the assumptions in the original linear MDP (Jin et al., 2020) which requires the integral $\int g\mu$ to be bounded for any function $g \in [0, 1]$. Thus, while being a linear MDP, the original LSVI-UCB algorithm (Jin et al., 2020) indeed will not work for this example. However, we note that our algorithm can still work.

Nevertheless, as the above example leverages a careful design of the feature, we might hope that some non-expansiveness properties could hold under stronger representation assumptions (e.g.,

when state space is tabular). Unfortunately, the following example shows that Bellman backup can be expansive even in tabular MDPs.

**Example 2** (Expansiveness of Bellman Backup in $\ell_2$-norm). *Consider a tabular MDP with horizon $H = 2$, $S$ states $\{s_1, \ldots, s_S\}$ in the first layer, a single state $\overline{s}$ in the second layer, and a single action $a$. On taking action $a$ in any state in the first-layer, the agent deterministically transitions to $\overline{s}$, and on taking action $a$ in $\overline{s}$ deterministically yields a reward of 1. Since linear Bellman completeness captures tabular MDPs with one-hot encoded features where $\phi(s_i, a) = e_i \in \mathbb{R}^{S+1}$ for $i \leq S$ and $\phi(\overline{s}, a) = e_{S+1} = (0, \ldots, 0, 1)^\top$, the state-action value function at the second layer can be parameterized by $\theta_2 = (0, \ldots, 0, 1)^\top$. However, applying the Bellman backup, since the return-to-go for any first-layer state $s_i$ is 1 (because $\overline{s}$ always yields a reward of 1), the backed-up value function must be parameterized by $\theta_1 = (1, 1, \ldots, 1)^\top$. Here, we find that $\|\theta_1\|_2 / \|\theta_2\|_2 = \sqrt{S}$, thus showing that Bellman backup cannot guarantee non-expansiveness of the $\ell_2$-norm.*

Hence, in this paper, we aim not to assume any $\ell_2$-norm bound or $\ell_2$-norm non-expansiveness of the parameters. Unfortunately, without these assumptions, the ground truth parameter of the optimal value function can exponentially grow with the horizon as evidenced by the examples above, thus invalidating prior methods requiring bounded parameter. Our key contribution is an algorithm that remains efficient even if the parameter norm blows up but requiring deterministic transition.

## 3.2 Other Prior Works on Linear Bellman Completeness

In this section, we review prior efforts on RL under linear Bellman completeness and discuss various assumptions underlying these approaches.

**Efficient Algorithms under Generative Access.** A generative model takes as input a state-action pair $(s, a)$ and returns a sample $s' \sim \mathsf{T}(\cdot \mid s, a)$ and the reward signal. With such a generative model, Linear Least-Squares Value Iteration (LSVI) can achieve statistical and computational efficiency (Agarwal et al., 2019). However, generative access is a big assumption, and our work aims to operate with only online access.

**Efficient Algorithms under Explorability Assumption.** Zanette et al. (2020c) propose a reward-free algorithm under the assumption that every direction in the parameter space is reachable. This assumption, when translated into tabular MDPs, means that any state can be reached with a probability bounded below by some (large enough) positive constant. This does not hold if there are unreachable states or if the probability of reaching them is exponentially small.

**Computationally Intractable Algorithms.** Zanette et al. (2020b) present a computationally intractable algorithm that requires solving an intractable optimization problem. In our work, we aim to only utilize a tractable squared loss minimization oracle.

**Few action MDPs.** Golowich & Moitra (2024) propose a computationally efficient algorithm under linear Bellman completeness, inspired by the bonus-based exploration approach in LSVI-UCB (Jin et al., 2020) for Linear MDPs. While their algorithm extends to stochastic MDPs, both the sample complexity and running time have exponential dependence on the size of the action space. In comparison, our algorithm extends to infinite action spaces but relies on the transition dynamics to be deterministic.

**Deterministic Rewards or Deterministic Initial State.** Several existing studies provide computationally and statistically efficient algorithms for more general settings but under stronger assumptions; these methods can be extended to linear Bellman completeness settings but similarly strong assumptions will also apply. Du et al. (2020) provide an algorithm based on a span argument that is efficient for MDPs that have linear optimal state-action value function (a.k.a. the Linear $Q^\star$ setting), deterministic transition dynamics, deterministic initial state, and stochastic rewards. Unfortunately, their approach cannot extend to settings with stochastic initial states, as we consider in our paper. Another line of work due to Wen & Van Roy (2017) considers the $Q^\star$-realizable setting with deterministic dynamics, deterministic rewards, stochastic initial states, and bounded eluder dimension. Their approach can be extended to the linear bellman completeness setting when both rewards and dynamics are deterministic. However, their algorithm fails to converge when rewards are stochastic and thus may not apply to the problem setting that we consider.

**Efficient Algorithm in the hybrid RL setting.** Song et al. (2022) develop efficient algorithms for the hybrid RL setting, where the learner has access to both online interaction and an offline dataset. However, they do not have a fully online algorithms.

In summary, no previous work addresses the problem with stochastic initial states, stochastic rewards, and large action spaces. This is the gap that we aim to fill with this work.

## 4 ALGORITHM

In this section, we present our algorithm for online RL under linear Bellman completeness. See Algorithm 1 for pseudocode. The input to the algorithm consists of three components. First, the noise variances, $\{\sigma_h\}_{h=1}^H$ and $\sigma_R$, control the scale of the random noise. Second, a D-optimal design (defined below) for the feature space.

**Definition 2** (D-optimal design). *The D-optimal design for the set of features* $\Phi = \{\phi(s,a) : s \in \mathcal{S}, a \in \mathcal{A}\}$ *is a distribution* $\rho$ *over* $\Phi$ *that maximizes* $\log \det\left(\sum_{\phi \in \Phi} \rho(\phi)\phi\phi^\top\right)$.

There always exist D-optimal designs with at most $O(d^2)$ support points (Lemma 23). Many efficient algorithms can be applied to find approximate D-optimal designs such as the Frank-Wolfe. The algorithm also requires a constrained squared loss minimization oracle $\mathcal{O}^{\mathrm{sq}}$, and we introduce an instantiation of $\mathcal{O}^{\mathrm{sq}}$ in Section 6.

---

**Algorithm 1** Null Space Randomization for Linear Bellman Completeness

**Require:** • Noise variances $\{\sigma_h\}_{h=1}^H$ and $\sigma_R$.
      • A D-optimal design for $\Phi = \{\phi(s,a) : s \in \mathcal{S}, a \in \mathcal{A}\}$ given by $\{(\phi_i, \rho_i)\}_{i=1}^m$.
      • Squared loss minimization oracle $\mathcal{O}^{\mathrm{sq}}$.

1: Define $\Sigma_{1,h} := \sum_{i=1}^m \rho_i \phi_i \phi_i^\top$ for all $h \in [H]$.
2: **for** $t = 1, \ldots, T$ **do**
3:      Let $\overline{\theta}_{t,H+1} \leftarrow 0, \overline{Q}_{t,H+1} \leftarrow 0, \overline{V}_{t,H+1} \leftarrow 0$.
4:      **for** $h = H, \ldots, 1$ **do**
5:          Let $P_{t,h}$ be the orthogonal projection matrix onto $\mathrm{span}(\{\phi(s_{i,h}, a_{i,h}) : i = 1, \ldots, t-1\})$
6:          For $i \in [m]$, define $\phi_{t,h,i}^\| = P_{t,h}\phi_i$ and $\phi_{t,h,i}^\perp = (I - P_{t,h})\phi_i$
7:          Let $\Lambda_{t,h} \leftarrow \sum_{i=1}^m \rho_i(\phi_{t,h,i}^\|(\phi_{t,h,i}^\|)^\top + \phi_{t,h,i}^\perp(\phi_{t,h,i}^\perp)^\top)$
8:          // Fit value function and reward using squared loss regression //
9:          Compute $\widehat{\theta}_{t,h}$ and $\widehat{\omega}_{t,h}$ using the squared loss minimization oracle $\mathcal{O}^{\mathrm{sq}}$ as:

$$\widehat{\theta}_{t,h} \leftarrow \operatorname*{argmin}_{\theta \in \mathcal{O}(W_h)} \sum_{i=1}^{t-1} \left(\langle \theta, \phi(s_{i,h}, a_{i,h})\rangle - \overline{V}_{t,h+1}(s_{i,h+1})\right)^2 \tag{1}$$

$$\widehat{\omega}_{t,h} \leftarrow \operatorname*{argmin}_{\omega \in \mathcal{O}(1)} \sum_{i=1}^{t-1} \left(\langle \omega, \phi(s_{i,h}, a_{i,h})\rangle - r_{i,h}\right)^2 \tag{2}$$

10:        // Perturb the estimated parameters by adding Gaussian noise //
11:        Update the parameters by sampling:

$$\overline{\theta}_{t,h} \sim \widehat{\theta}_{t,h} + \mathcal{N}\left(0, \sigma_h^2(I - P_{t,h})\Lambda_{t,h}^{-1}(I - P_{t,h})\right)$$

$$\overline{\omega}_{t,h} \sim \widehat{\omega}_{t,h} + \mathcal{N}(0, \sigma_R^2 \Sigma_{t,h}^{-1})$$

12:        Define $\overline{Q}_{t,h}(s,a) \leftarrow \langle \overline{\omega}_{t,h} + \overline{\theta}_{t,h}, \phi(s,a)\rangle$ and $\overline{V}_{t,h}(s) \leftarrow \max_a \overline{Q}_{t,h}(s,a)$ for all $(s,a)$
13:      **end for**
14:      Define the policy $\pi_t$ such that $\pi_{t,h}(s) = \operatorname{argmax}_a \overline{Q}_{t,h}(s,a)$
15:      Generate trajectory $(s_{t,1}, a_{t,1}, r_{t,1}, \ldots, s_{t,H}, a_{t,H}, r_{t,H}) \sim \pi_t$
16:      Define $\Sigma_{t+1,h} := \Sigma_{t,h} + \phi(s_{t,h}, a_{t,h})\phi^\top(s_{t,h}, a_{t,h})$ for all $h \in [H]$
17: **end for**

---

The algorithm begins by initializing the covariance matrix $\Sigma_{1,h}$ for all $h \in [H]$ using the optimal design, which differs from most standard LSVI-type algorithms where it is initialized to the identity matrix. We believe that the identity matrix is unsuitable here since we do not assume any $\ell_2$-norm bound on the parameters. Additionally, recalling that we assume the feature space spans $\mathbb{R}^d$, it ensures $\Sigma_{t,h}$ is invertible for all $t$ and $h$. Otherwise, pseudo-inverses can be used instead.

At each round $t \in [T]$, the algorithm operates in a backward manner starting from the last horizon $H$. For each $h \in [H]$, it first constructs the orthogonal projection matrix $P_{t,h}$ onto the span of the historical data. It then decomposes the D-optimal design points into the span and null space components using the projection and constructs $\Lambda_{t,h}$. By separating the span and null space components, it facilitates clearer concentration bounds for the subsequent Gaussian noise.

The algorithm then performs constrained squared loss regression to estimate the value function and reward function. Here we define $\mathscr{O}(W) \coloneqq \{\theta \in \mathbb{R}^d : |\langle \theta, \phi(s,a) \rangle| \le W \text{ for all } s \in \mathcal{S}, a \in \mathcal{A}\}$ for any $W > 0$. This *convex* constrained set is defined by the $\ell_\infty$-functional-norm bound instead of the $\ell_2$-norm because we do not assume any bound on the $\ell_2$-norm of the learned parameters. Here we define $W_h = \widetilde{\Theta}((d\sqrt{mH})^{H-h}(d^{3/2} + d\sqrt{mH}))$ (detailed definition deferred to Appendix C). We note that although $W_h$ appears exponential, which may seem suspicious, this does not affect our sample efficiency due to the span argument that we introduce in the analysis. We note that prior RLSVI algorithms used truncation on value functions to explicitly avoid such an exponential blow-up. However, truncation does not work for linear Bellman completeness setting since the Bellman backup on a truncated value function is not necessarily a linear function anymore.

Next, the algorithm perturbs the estimated parameters by adding Gaussian noise. The noise for the value function act *only in the null space* of the data covariance matrix. This ensures optimism while keeping the estimate accurate in the span space. It is a key modification from the standard RLSVI algorithm. The perturbation for the reward function is standard. Finally, the algorithm constructs the value function for the current horizon and the greedy policy with respect to it. It then generates the trajectory by executing the greedy policy, and the covariance matrix is updated.

## 5 ANALYSIS

In this section, we provide the theoretical guarantees of Algorithm 1. A high-level proof sketch can be found in Appendix B and detailed proofs are in Appendix C. We first consider the case where the squared loss minimization oracle is exact. We then extend the analysis to the approximate oracle and the low inherent linear Bellman error setting in subsequent sections.

### 5.1 PRELUDE: LEARNING WITH EXACT SQUARE LOSS MINIMIZATION ORACLE

We first consider the most ideal setting where the squared loss minimization oracle is exact.

**Assumption 2** (Exact Squared Loss Minimization Oracle). *Line 9 of Algorithm 1 is solved exactly.*

Then, we have the following regret bound. A proof sketch is provided in Appendix B for the readers convenience.

**Theorem 1** (Regret Bound with Exact Oracle). *Under Assumptions 1 and 2, executing Algorithm 1 with parameters $\sigma_{\mathrm{R}} = \widetilde{\Theta}(\sqrt{dH\log(HT)})$ and $\sigma_h = \widetilde{\Theta}((d\sqrt{mH})^{H-h+1}(\sqrt{d} + \sqrt{mH}))$, we have*

$$\mathrm{Reg}_T = \widetilde{O}(d^{5/2}H^{5/2} + d^2 H^{3/2}\sqrt{T}).$$

This result has several notable features. First, it does not depend on the number of actions. The only requirement for the action space is the ability to compute the argmax. Second, the $\sqrt{T}$-dependence on $T$ is optimal, as it is necessary even in the bandit setting. Additionally, we emphasize that the dependence on $\sqrt{T}$ arises solely from reward learning due to the application of elliptical potential lemma. In fact, if the reward function is known, our regret bound can be as small as $\widetilde{O}(dH^2)$, depending on $T$ up to logarithmic factors. We elaborate on this observation in Appendix B. As a standard practice, Theorem 1 can be converted into a sample complexity bound below.

**Corollary 1** (Sample Complexity Bound). *Let $\varepsilon \le 1$. Under the same setting as Theorem 1, letting $T \ge \Omega(d^4 H^3/\varepsilon^2)$, we get that the policy $\widehat{\pi}$ chosen uniformly from the set $\pi_1, \ldots, \pi_T$ enjoys performance guarantee $\mathbb{E}[V^\star - V^{\widehat{\pi}}] \le \varepsilon$.*

## 5.2 Learning with Approximate Square Loss Minimization Oracle

**Assumption 3** (Approximate Squared Loss Minimization Oracle). *We assume access to an approximate squared loss minimization oracle $\mathcal{O}^{\mathrm{sq}}_{\mathrm{apx}}$ that takes as input a problem of the form:* $\mathrm{argmin}_{\theta \in \mathscr{O}(W)} g(\theta) \coloneqq \sum_{(\phi(s,a),u) \in \mathcal{D}} (\langle \theta, \phi(s,a) \rangle - u)^2$ *where $\mathscr{O}(W) = \{\theta \in \mathbb{R}^d \mid |\langle \theta, \phi(s,a) \rangle| \le W\}$ for some $W \in \mathbb{R}$ is a convex set, and $\mathcal{D}$ is a dataset of tuples $\{(\phi(s,a), u)\}$. The oracle returns a point $\widehat{\theta}$ that satisfies $g(\widehat{\theta}) - \min_{\theta \in \mathscr{O}(W)} g(\theta) \le \varepsilon_1^2$ and $\widehat{\theta} \in \mathscr{O}(W + \varepsilon_2)$ where $\varepsilon_1, \varepsilon_2 \le 1$ are precision parameters of the oracle.*

With an approximate oracle, the regret bound depends on an additional quantity defined below.

**Assumption 4.** *There exists a constant $\gamma > 1$ such that, for any $r \le d$, and for any $\phi_1, \phi_2, \ldots, \phi_r \in \Phi$, the eigenvalues of the matrix $\Sigma \coloneqq \sum_{i=1}^r \phi_i \phi_i^\intercal$ are either zero or at least $1/\gamma^2$.*

As a concrete example, it holds with $\gamma = 1$ when the MDP is tabular. This assumption implies that the eigenvalues of $\Sigma^\dagger$ are at most $\gamma^2$. Consequently, for any vector $\phi \in \Phi$, we have $\|\phi\|_{\Sigma^\dagger} \le \|\phi\|_2 \gamma \le \gamma$—this lower bound on the norm of any vector is exactly what we need for the analysis of an approximate oracle, while Assumption 4 simply serves as a sufficient condition for it. The following theorem provides the regret bound with the approximate oracle in terms of parameters $\varepsilon_1, \varepsilon_2$ and $\gamma$.

**Theorem 2** (Regret Bound with Approximate Oracle). *Under Assumptions 1, 3 and 4, executing Algorithm 1 with $\sigma_{\mathrm{R}} = \widetilde{\Theta}(\sqrt{dH})$ and $\sigma_h = \widetilde{\Theta}((d\sqrt{mH})^{H-h+1}(\varepsilon_1 \gamma \sqrt{H} + \sqrt{d} + \sqrt{mH}))$, we have*

$$\mathrm{Reg}_T = \widetilde{O}\big(d^{5/2} H^{5/2} + d^2 H^{3/2} \sqrt{T} + \varepsilon_1 \gamma \big(dH^2 + d^{3/2} H \sqrt{T}\big)\big).$$

Compared to Theorem 1, the regret bound has an additional term that depends on the approximation error $\varepsilon_1 \gamma$. Typically, $\varepsilon_1$ is from optimization and thus can be exponentially small with respect to the relevant parameters, as we later discuss in Section 6. Hence, we allow $\gamma$ to be exponentially large. Moreover, we note that $\varepsilon_2$ does not appear in the regret bound since it only affects the constraint violation of the regression, whose effect to the statistical guarantees is of lower order and thus ignored. In addition, we note that the regret bound does not depend on the number of actions, and the dependence on $T$ remains optimal, similar to the previous theorem.

## 5.3 Learning with Low Inherent Linear Bellman Error

Now we consider the setting where the MDP has low inherent linear Bellman error.

**Definition 3** (Inherent Linear Bellman Error). *Given $\varepsilon_{\mathrm{B}} \le 1$, an MDP $\mathcal{M}$ is said to have $\varepsilon_{\mathrm{B}}$-inherent linear Bellman error with respect to a feature mapping $\phi$ if there exists a mapping $\mathcal{T} : \mathbb{R}^d \to \mathbb{R}^d$ so that, for all $\theta \in \mathbb{R}^d$ and all $(s,a) \in \mathcal{S} \times \mathcal{A}$, it holds that $|\langle \mathcal{T}\theta, \phi(s,a) \rangle - \mathbb{E}_{s' \sim \mathsf{T}(s,a)} \max_{a'} \langle \theta, \phi(s',a') \rangle| \le \varepsilon_{\mathrm{B}}$. Moreover, we require that, for all $h \in [H]$ and $(s,a) \in \mathcal{S} \times \mathcal{A}$, the random reward is bounded in $[0,1]$ with $|r_h(s,a) - \langle \omega_h^\star, \phi(s,a) \rangle| \le \varepsilon_{\mathrm{B}}$ for some unknown $\omega_h^\star \in \mathbb{R}^d$.*

With low inherent Bellman error, Assumption 4 is still necessary. The following theorem provides the regret bound in this case. We assume the exact oracle for simplicity.

**Theorem 3** (Regret Bound with Low Inherent Bellman Error). *Assume the MDP has $\varepsilon_{\mathrm{B}}$-inherent Bellman error. Under Assumptions 1, 2 and 4, when executing Algorithm 1 with parameters $\sigma_{\mathrm{R}} = \widetilde{\Theta}(\sqrt{dH} + \varepsilon_{\mathrm{B}} HT)$ and $\sigma_h = \widetilde{\Theta}((d\sqrt{mH})^{H-h+1}(\varepsilon_{\mathrm{B}} \gamma \sqrt{HT} + \sqrt{\varepsilon_{\mathrm{B}} T} + \sqrt{d} + \sqrt{mH}))$, we have*

$$\mathrm{Reg}_T = \widetilde{O}\big(d^{5/2} H^{5/2} + d^2 H^{3/2} \sqrt{T} + \sqrt{\varepsilon_{\mathrm{B}}}\big(d^2 H^{5/2} \sqrt{T} + d^{3/2} H^{3/2} T\big) + \varepsilon_{\mathrm{B}} \gamma \big(dH^2 \sqrt{T} + d^{3/2} HT\big)\big).$$

Compared to Theorem 1, the regret bound includes two additional terms that depend on the inherent linear Bellman error $\varepsilon_{\mathrm{B}}$. For both terms, the dependence on $T$ is linear. We believe the $T$-dependence is unavoidable, as it also appears in similar settings (Zanette et al., 2020b). In addition, it is worth noting that the regret bound does not depend on the number of actions, and the other dependence on $T$ remains optimal, similar to previous theorems.

# 6 OPENING THE BLACK-BOX: IMPLEMENTING SQUARED LOSS MINIMIZATION ORACLES IN ALGORITHM 1

In this section, we detail a practical implementation of the desired squared loss oracle need by our algorithm. The implementation relies on the observation that a square loss minimization objective over a convex domain can be cast as a convex set feasibility problem—given a convex set $\mathcal{K}$, return a point $\widehat{\theta} \in \mathcal{K}$. Thus, we can use algorithms for convex set feasibility to implement the squared loss minimization oracles. However, even given this observation, our key challenge for an efficient algorithm is that the corresponding convex set could be exponentially large and only be described using exponentially many number of linear constraints. Fortunately, various works in the optimization literature propose computationally efficient procedures to find feasible points within such ill-defined sets, under mild oracle assumptions.

## 6.1 COMPUTATIONALLY EFFICIENT CONVEX SET FEASIBILITY

We first paraphrase the work of Bertsimas & Vempala (2004) that provide a computationally efficient procedure for finding feasible points within a convex set by random walks. Notably, the computational complexity of their algorithm only depends logarithmically on the size of the convex set, and thus their approach is well suited for the corresponding convex feasibility problems that appear in our approach. At a high level, they provide an algorithm that takes an input an arbitrary convex set $\mathcal{K} \subseteq \mathbb{R}^d$, and returns a feasible point $\widehat{z} \in \mathcal{K}$. Their algorithm accesses the convex set $\mathcal{K}$ via a separation oracle defined as follows.

**Definition 4** (Separation oracle). *A separation oracle for a convex set $\mathcal{K}$, denoted by $\mathcal{O}_{\mathcal{K}}^{\mathrm{sep}}$, is defined such that on any input $z \in \mathbb{R}^d$, the oracle either confirms that $z \in \mathcal{K}$ or returns a hyperplane $\langle a, z \rangle \le b$ that separates the point $z$ from the set $\mathcal{K}$.*

In order to ensure finite time convergence for their procedure, they assume that the convex set $\mathcal{K}$ is not degenerate and is bounded in any direction. This is formalized by the following assumption.

**Assumption 5.** *The convex set $\mathcal{K}$ is $(r, R)$–Bounded, i.e. there exist parameters $0 < r \le R$ such that (a) $\mathcal{K} \subseteq \mathbb{R}_{\infty}(R)$, and (b) there exists a vector $z \in \mathbb{R}^d$ such that the shifted cube $(z + \mathbb{R}_{\infty}(r)) \subseteq \mathcal{K}$.*

The computational efficiency and the convergence guarantee of their algorithm are below.

**Theorem 4** (Bertsimas & Vempala (2004)). *Let $\delta \in (0, 1)$ and $\mathcal{K} \subset \mathbb{R}^d$ be an arbitrary convex set that satisfies Assumption 5 for some $0 \le r \le R$. Then, Algorithm 2 (given in the appendix), when invoked with the separation oracle $\mathcal{O}_{\mathcal{K}}^{\mathrm{sep}}$ w.r.t. $\mathcal{K}$, returns a feasible point $\widehat{z} \in \mathcal{K}$ with probability at least $1 - \delta$. Moreover, Algorithm 2 makes $O(d \log(R/\delta r))$ calls to the oracle $\mathcal{O}_{\mathcal{K}}^{\mathrm{sep}}$ and runs in time $O(d^7 \log(R/\delta r))$.*

Notice that both the number of oracle calls and the running time only depend logarithmically on $R$ and $r$, and thus their procedure can be efficiently implemented for our applications where $R/r$ may be exponentially large in the corresponding problem parameters.

## 6.2 COMPUTATIONALLY EFFICIENT ESTIMATION OF VALUE FUNCTION (EQN (1))

We now described how to leverage the method by Bertsimas & Vempala (2004) to estimate the parameters for the value functions in (1) in Algorithm 1. Note that for any time $t$ and horizon $h \in [H]$, the objective in (1) is the optimization problem

$$\widehat{\theta}_{t,h} \leftarrow \operatorname*{argmin}_{\theta \in \mathscr{O}(W_h)} \sum_{i=1}^{t-1} \left( \langle \theta, \phi(s_{i,h}, a_{i,h}) \rangle - \overline{V}_{t,h+1}(s_{i,h+1}) \right)^2, \tag{3}$$

where $W_h = \widetilde{\Theta}((d\sqrt{mH})^{H-h}(\varepsilon_1 d\gamma\sqrt{H} + d^{3/2} + d\sqrt{mH}))$. We provide a computationally efficient procedure to approximately solve the above given a linear optimization oracle over the feature space.

**Assumption 6** (Linear optimization oracle over the feature space). *Learner has access to a linear optimization oracle $\mathcal{O}^{\mathrm{lin}}$ that on taking input $\theta \in \mathbb{R}^d$, returns a feature $\phi(s', a') \in \operatorname{argmax}_{s,a} \langle \theta, \phi(s, a) \rangle$.*

The key observation we use is that under linear Bellman completeness (Definition 1) and deterministic dynamics (Assumption 1), any solution $\theta$ for (3) must satisfy $\sum_{i=1}^{t-1}(\langle\theta, \phi(s_{i,h}, a_{i,h})\rangle - \overline{V}_{t,h+1}(s_{i,h+1}))^2 = 0$. On the other hand, the converse also holds that any point $\theta \in \mathscr{O}(W_h)$ for which the objective value is $0$ must be a solution to (3). Thus, the minimization problem in (3) is equivalent to finding a feasible point within the convex set

$$\mathcal{K} := \left\{ \theta \in \mathbb{R}^d \;\middle|\; \begin{array}{c} \left(\langle\theta, \phi(s_{i,h}, a_{i,h})\rangle - \overline{V}_{t,h+1}(s_{i,h+1})\right)^2 = 0 \text{ for all } i \le t \\ |\langle\theta, \phi(s,a)\rangle| \le W_h \text{ for all } s,a \end{array} \right\}. \tag{4}$$

Given the above reformulation of the optimization objective (3) as a feasibility problem, we can now use the procedure of Bertsimas & Vempala (2004) for finding $\theta_{t,h} \in \mathcal{K}$. However, we first need to define a separation oracle for the set $\mathcal{K}$ and verify Assumption 5. Unfortunately, there may not exist any $r > 0$ for which $(z + \mathbb{R}_\infty(r)) \subseteq \mathcal{K}$ for some $z \in \mathbb{R}^d$ in our case and thus the above $\mathcal{K}$ may not satisfy Assumption 5. This can, however, be easily fixed by artificially increasing the set $\mathcal{K}$ to allow for some approximation errors. In particular, let $\varepsilon > 0$ and define the convex set

$$\mathcal{K}_{\text{APX}} := \left\{ \theta \in \mathbb{R}^d \;\middle|\; \begin{array}{c} \langle\theta, \phi(s_{i,h}, a_{i,h})\rangle - \overline{V}_{t,h+1}(s_{i,h+1}) \le \varepsilon \text{ for all } i \le t \\ \langle\theta, \phi(s_{i,h}, a_{i,h})\rangle - \overline{V}_{t,h+1}(s_{i,h+1}) \ge -\varepsilon \text{ for all } i \le t \\ |\langle\theta, \phi(s,a)\rangle| \le W_h + \varepsilon \text{ for all } s,a \end{array} \right\}. \tag{5}$$

Clearly, since there exists at least one point $\theta_{t,h} \in \mathcal{K}$, we must have that $(\theta_{t,h} + \mathbb{R}_\infty(\varepsilon)) \subseteq \mathcal{K}_{\text{APX}}$. To ensure an outer bounding box for the set $\mathcal{K}_{\text{APX}}$, we need to make an additional assumption.

**Assumption 7.** *Let $\Phi = \{\phi(s,a) \mid s, a \in \mathcal{S} \times \mathcal{A}\}$. There exist some $R \ge 0$ such that $\frac{1}{R}e_i \in \Phi$, where $e_i$ denotes the unit basis vector along the $i$-th direction in $\mathbb{R}^d$.*

The above assumption ensures that $\mathcal{K} \subseteq \mathbb{B}_\infty(W_h R)$. Recall that we can tolerate the parameter $R$ to be exponential in the dimension $d$ or the horizon $H$. Finally, a separation oracle can be implemented using $\mathscr{O}^{\text{lin}}$ (see Algorithm 4 for details). Thus, one can use Algorithm 2 (given in appendix), due to Bertsimas & Vempala (2004), and the guarantee in Theorem 4 to find a feasible point in $\mathcal{K}_{\text{APX}}$, which corresponds to an approximate solution to (3).

**Theorem 5.** *Let $\varepsilon > 0$, $\delta \in (0, 1)$, and suppose Assumption 7 holds with some parameter $R > 0$. Additionally, suppose Assumption 6 holds with the linear optimization oracle denoted by $\mathscr{O}^{\text{lin}}$. Then, there exists a computationally efficient procedure (given in Algorithm 4 in the appendix), that for any $t \in [T]$ and $h \in [H]$, returns a point $\widehat{\theta}_{t,h}$ that, with probability at least $1 - \delta$, satisfies*

$$\sum_{i=1}^{t-1} \left( \langle\widehat{\theta}_{t,h}, \phi(s_{i,h}, a_{i,h})\rangle - \overline{V}_{t,h+1}(s_{i,h+1}) \right)^2 \le T\varepsilon \qquad \text{and} \qquad \widehat{\theta}_{t,h} \in \mathscr{O}(W_h + \varepsilon).$$

*Furthermore, Algorithm 4 takes $O(d^7 \log(\frac{R}{\delta\varepsilon}))$ time in addition to $O(d \log(\frac{THR}{\delta\varepsilon}))$ calls to $\mathscr{O}^{\text{lin}}$.*

The above techniques and Algorithm 4 can be similarly extended to get a computationally efficient procedure to estimate the reward parameter in (2). The main difference is that the value of the optimization objective in (2) is not zero at the minimizer (due to stochasticity). Thus, we need to construct a set feasibility problem for every desired target value of the objective function within the grid $[0, \varepsilon, 2\varepsilon, \ldots, 2 - \varepsilon, 2]$ and use a separating hyperplane w.r.t. the ellipsoid constraint in (2) to implement the separating hyperplane for $\mathcal{K}_{\text{APX}}$ (which can be implemented using projections).

## 7 CONCLUSION

In this paper, we develop a computationally efficient RL algorithm under linear Bellman completeness with deterministic dynamics, aiming to bridge the statistical-computational gap in this setting. Our algorithm injects random noise into regression estimates only in the null space to ensure optimism and leverages a span argument to bound regret. It handles large action spaces, random initial states, and stochastic rewards. Our key observation is that deterministic dynamics simplifies the learning process by ensuring accurate value estimates within the data span, allowing noise injection to be confined to the null space. Extending our algorithm to stochastic dynamics remains an open challenge.

## ACKNOWLEDGMENTS

We thank Yuda Song, Zeyu Jia, Noah Golowich, and Sasha Rakhlin for useful discussions. AS acknowledges support from the Simons Foundation and NSF through award DMS-2031883, as well as from the DOE through award DE-SC0022199. WS acknowledges support from NSF IIS-2154711, NSF CAREER 2339395, and DARPA LANCER: LeArning Network CybERagents.

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

CONTENTS OF APPENDIX

## A    TABLE OF NOTATION

We list the notation used in this paper in table 1, for the convenience of reference.

Table 1: Notation used in the paper.

| Symbol | Description |
|--------|-------------|
| $\mathscr{O}(W)$ | $\{\theta \in \mathbb{R}^d : |\langle \theta, \phi(s,a)\rangle| \leq W \text{ for all } s \in \mathcal{S}, a \in \mathcal{A}\}$ |
| $\mathbb{R}_\infty(W)$ | $\{\theta \in \mathbb{R}^d : \|\theta\|_\infty \leq W\}$ |
| $\mathbb{R}_2(W)$ | $\{\theta \in \mathbb{R}^d : \|\theta\|_2 \leq W\}$ |
| $\eta_{t,h}$ | $\mathcal{T}(\overline{\omega}_{t,h+1} + \overline{\theta}_{t,h+1}) - \widehat{\theta}_{t,h}$ |
| $\eta_{t,h}^{\mathrm{R}}$ | $\omega_h^\star - \widehat{\omega}_{t,h}$ |
| $\xi_t^{\mathrm{R}}$ | $\overline{\omega}_{t,h} - \widehat{\omega}_{t,h}$ |
| $\xi_{t,h}^{\mathrm{P}}$ | $\overline{\theta}_{t,h} - \widehat{\theta}_{t,h}$ |
| $\mathfrak{E}^{\mathrm{high}}$ | High probability event, defined in Definition 5 |
| $\mathfrak{E}_t^{\mathrm{span}}$ | Event that trajectory at round $t$ is within the span of historical data, defined in (6) |
| $\mathfrak{E}_t^{\mathrm{optm}}$ | Optimism event at round $t$, defined in Lemma 14 |
| $U_{t,h}$ | Value function lower bound, defined in Appendix C.2 |
| $B_{\mathrm{err}}^{\mathrm{R}}$ | Upper bound of $\|\widehat{\omega}_{t,h} - \omega_h^\star\|_{\Sigma_t}$, defined in Definition 5 |
| $B_{\mathrm{err}}^{\mathrm{P}}$ | Upper bound of $\|\widehat{\theta}_{t,h} - \mathcal{T}(\omega_{t,h} + \theta_{t,h+1})\|_{\widehat{\Sigma}_{t,h}}$, defined in Lemma 7 |
| $B_{\mathrm{noise}}^{\mathrm{R}}$ | Upper bound of $\|\xi_{t,h}^{\mathrm{R}}\|_{\Sigma_{t,h}}$, defined in Definition 5 |
| $B_{\mathrm{noise},h}^{\mathrm{P}}$ | Upper bound of $\|\xi_{t,h}^{\mathrm{P}}\|_{\Lambda_{t,h}}$, defined in Definition 5 |
| $B_\phi^{\mathrm{R}}$ | Upper bound of $\sum_{t=1}^{T} \|\phi(s_{t,h}, a_{t,h})\|_{\Sigma_{t,h}^{-1}}$ defined in Lemma 16 |
| $B_\phi^{\mathrm{P}}$ | Upper bound of $\sum_{t=1}^{T} \mathbf{1}\{\mathfrak{E}_t^{\mathrm{span}}\}\|\phi(s_{t,h}, a_{t,h})\|_{\widehat{\Sigma}_{t,h}^\dagger}$, defined in Lemma 16 |
| $B_V$ | Upper bound of $|\overline{V}_t|$ conditioning on $\mathfrak{E}_t^{\mathrm{span}}$ and $\mathfrak{E}^{\mathrm{high}}$, defined in Lemma 13 |
| $\Sigma_{t,h}$ | $\sum_{i=1}^{m} \rho_i \phi_i \phi_i^\top + \sum_{i=1}^{t-1} \phi(s_{i,h}, a_{i,h})\phi^\top(s_{i,h}, a_{i,h})$ |
| $\widehat{\Sigma}_{t,h}$ | $\sum_{i=1}^{t-1} \phi(s_{i,h}, a_{i,h})\phi^\top(s_{i,h}, a_{i,h})$ |
| $W_h$ | Recursively defined as $W_{h-1} = W_h + 2\varepsilon_2 + \sqrt{2d} \cdot B_{\mathrm{noise},h}^{\mathrm{P}} + \sqrt{2d} \cdot B_{\mathrm{noise}}^{\mathrm{R}} + 1$ with $W_{H+1} = 1$ |

## B  PROOF OVERVIEW

In this section, we provide a sketch of the proof of Theorem 1 (exact oracle and zero inherent linear Bellman error) with the full proofs deferred to Appendix C. To better convey the intuition, we now assume that the reward function is known, as reward learning is largely standard. In particular, we temporarily remove the estimation and perturbation of rewards (Lines 9 and 11) and simply assume $\overline{\omega}_{t,h} = \omega_h^\star$ in Line 12.

### B.1  SPAN ARGUMENT

The very first step of our analysis revolve around two complimentary cases – whether the trajectory at round $t$ is in the span of the historical data or not. Let $\mathcal{D}_{t,h} := \{\phi(s_{i,h}, a_{i,h})\}_{i=1}^t$ and define $\mathfrak{E}_t^{\mathrm{span}}$ as the event that the trajectory at round $t$ is in the span of the historical data, i.e.,

$$\mathfrak{E}_t^{\mathrm{span}} := \{\forall h \in [H] : \phi(s_{t,h}, a_{t,h}) \in \mathrm{span}(\mathcal{D}_{t-1,h})\}. \tag{6}$$

*(1) In-span case.* When the trajectory generated in the round $t$ is completely within the span of historical data, we can assert that the value function estimation is accurate under $\pi_t$. Particularly, by linear Bellman completeness, the Bayes optimal of the regression in Line 9 zeros the empirical risk, as formally stated in the following lemma.

**Lemma 1.** *For any $t \in [T]$, we have $\sum_{i=1}^{t-1} (\langle \widehat{\theta}_{t,h}, \phi(s_{i,h}, a_{i,h}) \rangle - \overline{V}_{t,h+1}(s_{i,h+1}))^2 = 0$.*

Define $U_t(\cdot)$ as a version of $\overline{V}_t(\cdot)$ that minimizes $\overline{V}_t(s_{t,1})$ while satisfying the high probability bound (precise definition provided at the beginning of Appendix C.2). It implies the following.

**Lemma 2.** *For any $t \in [T]$, whenever $\mathfrak{E}_t^{\mathrm{span}}$ holds, we have $\overline{V}_t(s_{t,1}) = U_t(s_{t,1}) = V^{\pi_t}(s_{t,1})$.*

To understand Lemma 2, we consider two fact: (1) $\pi_t$ is the optimal policy for the estimated value function $\overline{V}_t$, and (2) both $\overline{V}_t$ and $U_t$ has accurate value estimate for the trajectory induced by $\pi_t$, starting from $s_{t,1}$, because it is in the span of the historical data when $\mathfrak{E}_t^{\mathrm{span}}$ holds.

*(2) Out-of-span case.* When any segment of the trajectory is not within the span, we simply pay $H$ in regret and can assert that this will not occur too many times. To see this, we observe the following fact: whenever $\mathfrak{E}_t^{\mathrm{span}}$ does not hold, there must exists $h \in [H]$ such that $\dim \mathrm{span}(\mathcal{D}_{t,h}) = \dim \mathrm{span}(\mathcal{D}_{t-1,h}) + 1$ by definition. Since the dimension of spans cannot exceed $d$ for any $h \in [H]$, the case that $\mathfrak{E}_t^{\mathrm{span}}$ does not hold cannot happen for more than $dH$ times. We formally state it in the following lemma.

**Lemma 3.** *We have $\sum_{t=1}^T \mathbf{1}\{(\mathfrak{E}_t^{\mathrm{span}})^{\mathsf{C}}\} \leq dH$.*

Hence, we have the following decomposition:

$$V^\star(s_{t,1}) - V^{\pi_t}(s_{t,1}) = \mathbf{1}\{\mathfrak{E}_t^{\mathrm{span}}\}\left(V^\star(s_{t,1}) - V^{\pi_t}(s_{t,1})\right) + \underbrace{\mathbf{1}\{(\mathfrak{E}_t^{\mathrm{span}})^{\mathsf{C}}\}\left(V^\star(s_{t,1}) - V^{\pi_t}(s_{t,1})\right)}_{\leq dH^2 \text{ when summed over } t}$$

Therefore, we only need to focus on the rounds where $\mathfrak{E}_t^{\mathrm{span}}$ holds. This will be the aim of the subsequent sections.

### B.2  EXPLORATION IN THE NULL SPACE

Lemma 1 implies that the estimation error only comes from the null space of the historical data, i.e., $\mathrm{null}(\{\phi(s_{i,h}, a_{i,h}) : i = 1, \ldots, t-1\})$. Therefore, we only need to explore in this null space. While adding explicit bonus is infeasible under linear Bellman completeness, we add noise (Line 11) that can cancel out the estimation error in the null space. This achieves the following:

**Lemma 4** (Optimism with constant probability)**.** *Denote $\mathfrak{E}_t^{\mathrm{optm}}$ as the event that $V^\star(s_{t,1}) \leq \overline{V}_t(s_{t,1})$. Then, for any $t \in [T]$, we have $\Pr(\mathfrak{E}_t^{\mathrm{optm}}) \geq \Gamma^2(-1)$ where $\Gamma$ is the cumulative distribution function of the standard normal distribution.*

This result has been the key idea in randomized RL algorithms, such as RLSVI. In the next section, we will explore how this lemma is utilized.

### B.3 PROOF OUTLINE

In this section, we outline the structure of the whole proof. Let $\widetilde{V}$ denote an i.i.d. copy of $\overline{V}$, and $\widetilde{\mathfrak{E}}_t^{\mathrm{span}}, \widetilde{\mathfrak{E}}_t^{\mathrm{optm}}$ denote the counterpart of $\mathfrak{E}_t^{\mathrm{span}}, \mathfrak{E}_t^{\mathrm{optm}}$ for $\widetilde{V}$. We first invoke Lemma 2 and get

$$\mathbf{1}\{\mathfrak{E}_t^{\mathrm{span}}\}\Big(V^\star(s_{t,1}) - V^{\pi_t}(s_{t,1})\Big) = \mathbf{1}\{\mathfrak{E}_t^{\mathrm{span}}\}\Big(V^\star(s_{t,1}) - U_t(s_{t,1})\Big) \le V^\star(s_{t,1}) - \mathbf{1}\{\mathfrak{E}_t^{\mathrm{span}}\}U_t(s_{t,1})$$

where the last step is by the non-negativity of $V^\star$. Next, we apply Lemma 4 and get

$$\le \mathop{\mathbb{E}}_{\widetilde{V}_t}\Big[\min\{\widetilde{V}_t(s_{t,1}), H\} - \mathbf{1}\{\mathfrak{E}_t^{\mathrm{span}}\}U_t(s_{t,1})\Big|\widetilde{\mathfrak{E}}_t^{\mathrm{optm}}\Big]$$

Split it into two parts:

$$= \mathop{\mathbb{E}}_{\widetilde{V}_t}\Big[\mathbf{1}\{\widetilde{\mathfrak{E}}_t^{\mathrm{span}}\}\Big(\min\{\widetilde{V}_t(s_{t,1}), H\} - \mathbf{1}\{\mathfrak{E}_t^{\mathrm{span}}\}U_t(s_{t,1})\Big)\Big|\widetilde{\mathfrak{E}}_t^{\mathrm{optm}}\Big]$$

$$+ \mathop{\mathbb{E}}_{\widetilde{V}_t}\Big[\mathbf{1}\{(\widetilde{\mathfrak{E}}_t^{\mathrm{span}})^{\mathsf{C}}\}\Big(\min\{\widetilde{V}_t(s_{t,1}), H\} - \mathbf{1}\{\mathfrak{E}_t^{\mathrm{span}}\}U_t(s_{t,1})\Big)\Big|\widetilde{\mathfrak{E}}_t^{\mathrm{optm}}\Big]$$

Note that the quantity inside the first expectation is non-negative, so we can peel off the conditioning event; the quantity in the second term is simply upper bounded by $H$. Hence, we have

$$\le \frac{1}{\Gamma^2(-1)} \mathop{\mathbb{E}}_{\widetilde{V}_t}\Big[\mathbf{1}\{\widetilde{\mathfrak{E}}_t^{\mathrm{span}}\}\Big(\min\{\widetilde{V}_t(s_{t,1}), H\} - \mathbf{1}\{\mathfrak{E}_t^{\mathrm{span}}\}U_t(s_{t,1})\Big)\Big] + \frac{1}{\Gamma^2(-1)} \mathop{\mathbb{E}}_{\widetilde{V}_t}\Big[\mathbf{1}\{(\widetilde{\mathfrak{E}}_t^{\mathrm{span}})^{\mathsf{C}}\}H\Big]$$

Now we split the first term into two parts again:

$$= \frac{1}{\Gamma^2(-1)} \mathop{\mathbb{E}}_{\widetilde{V}_t}\Big[\mathbf{1}\{\widetilde{\mathfrak{E}}_t^{\mathrm{span}}\}\min\{\widetilde{V}_t(s_{t,1}), H\} - \mathbf{1}\{\mathfrak{E}_t^{\mathrm{span}}\}U_t(s_{t,1})\Big]$$

$$+ \frac{1}{\Gamma^2(-1)} \mathop{\mathbb{E}}_{\widetilde{V}_t}\Big[\mathbf{1}\{(\widetilde{\mathfrak{E}}_t^{\mathrm{span}})^{\mathsf{C}} \cap \mathfrak{E}_t^{\mathrm{span}}\}U_t(s_{t,1})\Big] + \frac{1}{\Gamma^2(-1)} \mathop{\mathbb{E}}_{\widetilde{V}_t}\Big[\mathbf{1}\{(\widetilde{\mathfrak{E}}_t^{\mathrm{span}})^{\mathsf{C}}\}H\Big]$$

$$\le \frac{1}{\Gamma^2(-1)} \mathop{\mathbb{E}}_{\widetilde{V}_t}\Big[\mathbf{1}\{\widetilde{\mathfrak{E}}_t^{\mathrm{span}}\}\min\{\widetilde{V}_t(s_{t,1}), H\} - \mathbf{1}\{\mathfrak{E}_t^{\mathrm{span}}\}U_t(s_{t,1})\Big] + \frac{2}{\Gamma^2(-1)} \mathop{\mathbb{E}}_{\widetilde{V}_t}\Big[\mathbf{1}\{(\widetilde{\mathfrak{E}}_t^{\mathrm{span}})^{\mathsf{C}}\}H\Big]$$

where we used the fact that $\mathbf{1}\{\mathfrak{E}_t^{\mathrm{span}}\}U_t(s_{t,1}) \le H$. Taking the expectation over the randomness of the algorithm and use the tower property, which converts $\widetilde{V}$ into $\overline{V}$, we obtain

$$\le \frac{1}{\Gamma^2(-1)} \mathbb{E}\Big[\mathbf{1}\{\mathfrak{E}_t^{\mathrm{span}}\}\min\{\overline{V}_t(s_{t,1}), H\} - \mathbf{1}\{\mathfrak{E}_t^{\mathrm{span}}\}U_t(s_{t,1})\Big] + \frac{2}{\Gamma^2(-1)} \mathbb{E}\Big[\mathbf{1}\{(\mathfrak{E}_t^{\mathrm{span}})^{\mathsf{C}}\}H\Big]$$

The first term is upper bounded by zero due to Lemma 2, and the second term is upper bounded by $dH^2$ by Lemma 3 when summed over $t$. This finishes the proof.

**Remark 1** (Span Argument and Exponential Blow-Up). *In the proof sketch above, we did not utilize any $\ell_2$-norm bound on $\overline{\theta}_{t,h}$ or $\widehat{\theta}_{t,h}$ as did in many prior works. We actually cannot leverage them since they can be exponentially large due to the addition of exponentially large noise. This phenomenon is widely observed in the literature (e.g., Agrawal et al. (2021); Zanette et al. (2020a)) and is addressed through truncation. However, truncation does not work under linear Bellman completeness, as the Bellman backup of a truncated value function is not necessarily linear. This is why we use the span argument to circumvent this issue.*

## C    FULL PROOF FOR SECTION 5

In this section, we present and prove the following main theorem, which provides the regret bound in terms of parameters $\varepsilon_1$, $\varepsilon_2$, and $\varepsilon_{\mathrm{B}}$. Setting $\varepsilon_1 = \varepsilon_2 = \varepsilon_{\mathrm{B}} = 0$ yields Theorem 1, setting $\varepsilon_{\mathrm{B}} = 0$ yields Theorem 2, and setting $\varepsilon_1 = \varepsilon_2 = 0$ yields Theorem 3.

**Theorem 6.** *Assume the MDP has $\varepsilon_{\mathrm{B}}$-inherent linear Bellman error. Under Assumptions 1, 3 and 4, when executing Algorithm 1 with parameters $\sigma_{\mathrm{R}} = \sqrt{H}B_{\mathrm{err}}^{\mathrm{R}}$ and $\sigma_h \ge \sqrt{H}(\sqrt{3}\gamma B_{\mathrm{err}}^{\mathrm{P}} + \sqrt{8m}(W_h + \varepsilon_2))$, we have*

$$\mathbb{E}\left[\sum_{t=1}^{T}\Big(V^\star(s_{t,1}) - V^{\pi_t}(s_{t,1})\Big)\right] = \widetilde{O}\Big(d^{5/2}H^{5/2} + d^2 H^{3/2}\sqrt{T} + \varepsilon_1\gamma\Big(dH^2 + d^{3/2}H\sqrt{T}\Big)$$

$$+ \sqrt{\varepsilon_{\mathrm{B}}}\Big(d^2 H^{5/2}\sqrt{T} + d^{3/2}H^{3/2}T\Big) + \varepsilon_{\mathrm{B}}\gamma\Big(dH^2\sqrt{T} + d^{3/2}HT\Big)\Big).$$

**Exact value of parameters** $\sigma_{\mathrm{R}}$ **and** $\sigma_h$ **in Theorem 6.** We define $W_{H+1} = 1$ and recursively define $W_{h-1} = W_h + 2\varepsilon_2 + \sqrt{2d} \cdot B_{\mathrm{noise},h}^{\mathrm{P}} + \sqrt{2d} \cdot B_{\mathrm{noise}}^{\mathrm{R}} + 1$. Plugging the definition of all these symbols involved and ignoring lower order terms (i.e., logarithmic and constant terms), we get

$$W_{h-1} \approx d\sqrt{mH} \cdot W_h + \varepsilon_1 \cdot d\gamma\sqrt{H} + \varepsilon_{\mathrm{B}} \cdot d\gamma\sqrt{HT} + \sqrt{\varepsilon_{\mathrm{B}}} \cdot d\sqrt{T} + d^{3/2}. \tag{7}$$

Solving this recursion, we get

$$W_h \approx \left(d\sqrt{mH}\right)^{H+1-h} + \left(d\sqrt{mH}\right)^{H-h}\left(\varepsilon_1 \cdot d\gamma\sqrt{H} + \varepsilon_{\mathrm{B}} \cdot d\gamma\sqrt{HT} + \sqrt{\varepsilon_{\mathrm{B}}} \cdot d\sqrt{T} + d^{3/2}\right)$$
$$\approx \left(d\sqrt{mH}\right)^{H-h}\left(\varepsilon_1 \cdot d\gamma\sqrt{H} + \varepsilon_{\mathrm{B}} \cdot d\gamma\sqrt{HT} + \sqrt{\varepsilon_{\mathrm{B}}} \cdot d\sqrt{T} + d^{3/2} + d\sqrt{mH}\right).$$

We insert this into the value of $\sigma_h$ and get

$$\sigma_h \approx \left(d\sqrt{mH}\right)^{H-h+1}\left(\varepsilon_1 \cdot \gamma\sqrt{H} + \varepsilon_{\mathrm{B}} \cdot \gamma\sqrt{HT} + \sqrt{\varepsilon_{\mathrm{B}}} \cdot \sqrt{T} + d^{1/2} + \sqrt{mH}\right).$$

We can also get the value of $\sigma_{\mathrm{R}}$ as

$$\sigma_{\mathrm{R}} \approx \sqrt{H}\left(\sqrt{d\log(HT)} + \varepsilon_1 + \sqrt{\varepsilon_{\mathrm{B}}T}\right).$$

Define $\Lambda = \sum_{i=1}^m \rho_i \phi_i \phi_i^\top$. It is straightforward that both $\Lambda$ and $\Lambda_{t,h}$ (constructed in Line 7 of Algorithm 1) are invertible. We define $\lambda \coloneqq \max_{s,a} \|\phi(s,a)\|_{\Lambda^{-1}}$ and $\lambda_{t,h} \coloneqq \max_{s,a} \|\phi(s,a)\|_{\Lambda_{t,h}^{-1}}$.

**Lemma 5.** *The matrices $\Lambda$ and $\Lambda_{t,h}$ are invertible. Furthermore, we also have that*

- $\lambda \le \sqrt{d}$;
- $\lambda_{t,h} \le \sqrt{2d}$ *for all $t \in [T]$ and all $h \in [H]$.*

**Proof of Lemma 5.** By the last item in Lemma 23, we have $\lambda \le \sqrt{d}$. In what follows, we will show that $\Lambda \preceq 2\Lambda_{t,h}$, which implies $\lambda_{t,h} \le \sqrt{2}\lambda \le \sqrt{2d}$.

For any $x \in \mathbb{R}^d$, we have

$$\begin{aligned} x^\top \Lambda x &= \sum_{i=1}^m \rho_i (x^\top \phi_i)^2 = \sum_{i=1}^m \rho_i \left(x^\top \phi_{t,h,i}^{\|} + x^\top \phi_{t,h,i}^{\perp}\right)^2 \\ &\le 2\sum_{i=1}^m \rho_i \left(x^\top \phi_{t,h,i}^{\|}\right)^2 + 2\sum_{i=1}^m \rho_i \left(x^\top \phi_{t,h,i}^{\perp}\right)^2 \qquad \text{(using } (a+b)^2 \le 2a^2 + 2b^2\text{)} \\ &= 2x^\top \Lambda_{t,h} x. \end{aligned}$$

This implies that $\Lambda \preceq 2\Lambda_{t,h}$. $\qquad\square$

## C.1 High-probability Event and Boundedness

**Lemma 6** (Reward estimation). *With probability at least $1 - \delta$, for any $t \in [T]$ and $h \in [H]$,*

$$\|\widehat{\omega}_{t,h} - \omega_h^\star\|_{\Sigma_t} \le \sqrt{1030(1+\varepsilon_2)^4 d\log\left(8(1+\varepsilon_2)e^2 T^2 H/\delta\right) + 4\varepsilon_1^2 + 16(1+\varepsilon_2)(1+\varepsilon_{\mathrm{B}}T)}.$$

**Proof of Lemma 6.** For the ease of notation, we fixed $t$ and $h$ in the proof and simply write the regression problem as

$$\widehat{\omega} \leftarrow \operatorname*{argmin}_{\omega \in \mathscr{O}(1)} \sum_{i=1}^n \left(\omega^\top \phi_i - r_i\right)^2$$

where we have dropped the subscripts $t$ and $h$ for notational simplicity. Here $\phi_i$ and $r_i$ are abbreviated notations for $\phi(s_{i,h}, a_{i,h})$ and $r_{i,h}$, respectively, and $n = t - 1$.

Note that, due to approximate oracle (Assumption 3), $\widehat{\omega}$ actually belongs to $\mathscr{O}(1 + \varepsilon_2)$ instead of $\mathscr{O}(1)$. Denote $\mathcal{C}$ as an $\ell_1$-norm $\alpha$-cover (Definition 6) on $\mathscr{O}(1 + \varepsilon_2)$ such that for any $\omega \in \mathscr{O}(1 + \varepsilon_2)$,

there exists a $\widetilde{\omega} \in \mathcal{C}$, such that $\sum_{i=1}^{n} |\omega^{\top}\phi_i - \widetilde{\omega}^{\top}\phi_i|/n \le \alpha$. Since $\mathcal{O}(1 + \varepsilon_2)$ is a linear function class, which has pseudo-dimension $d$ (Definition 8), we have

$$|\mathcal{C}| \le \left(8(1 + \varepsilon_2)e^2/\alpha\right)^d \tag{8}$$

by Lemma 27. Now define $z_i^{\omega} = (\omega^{\top}\phi_i - r_i)^2 - ((\omega^{\star})^{\top}\phi_i - r_i)^2$. Then we have $|z_i^{\omega}| \le 4(1 + \varepsilon_2)^2$, and

$$\begin{aligned}
\mathbb{E}_i[z_i^{\omega}] &= \mathbb{E}_i\left[(\omega^{\top}\phi_i - (\omega^{\star})^{\top}\phi_i)(\omega^{\top}\phi_i + (\omega^{\star})^{\top}\phi_i - 2r_i)\right] \\
&= \mathbb{E}_i\left[(\omega^{\top}\phi_i - (\omega^{\star})^{\top}\phi_i)\left(\omega^{\top}\phi_i - (\omega^{\star})^{\top}\phi_i + 2((\omega^{\star})^{\top}\phi_i - r_i)\right)\right] \\
&\ge (\omega^{\top}\phi_i - (\omega^{\star})^{\top}\phi_i)^2 - 4(1 + \varepsilon_2)\varepsilon_{\mathrm{B}},
\end{aligned}$$

and moreover,

$$\mathbb{E}_i[(z_i^{\omega})^2] = \mathbb{E}_i\left[(\omega^{\top}\phi_i - (\omega^{\star})^{\top}\phi_i)^2(\omega^{\top}\phi_i + (\omega^{\star})^{\top}\phi_i - 2r_i)^2\right] \le 16(1 + \varepsilon_2)^2(\omega^{\top}\phi_i - (\omega^{\star})^{\top}\phi_i)^2$$

We note that $z_i^{\omega} - \mathbb{E}_i z_i^{\omega}$ is a martingale difference sequence and $|z_i^{\omega} - \mathbb{E}_i z_i^{\omega}| \le 8(1 + \varepsilon_2)^2$. Applying Freedman's inequality (Lemma 22) and a union bound over $\omega \in \mathcal{C}$, we have with probability at least $1 - \delta$, for all $\omega \in \mathcal{C}$,

$$\begin{aligned}
&\sum_{i=1}^{n}(\omega^{\top}\phi_i - (\omega^{\star})^{\top}\phi_i)^2 - \sum_{i=1}^{n} z_i^{\omega} \\
&\le \eta \sum_{i=1}^{n} 16(1 + \varepsilon_2)^2(\omega^{\top}\phi_i - (\omega^{\star})^{\top}\phi_i)^2 + \frac{8(1 + \varepsilon_2)^2 \log(|\mathcal{C}|/\delta)}{\eta} + 4(1 + \varepsilon_2)\varepsilon_{\mathrm{B}}T. \tag{9}
\end{aligned}$$

Recall that $\widehat{\omega}$ is the least square solution. Denote $\widetilde{\omega} \in \mathcal{C}$ as the point that is closest to $\widehat{\omega}$, which means that: $\sum_{i=1}^{n} |\widehat{\omega}^{\top}\phi_i - \widetilde{\omega}^{\top}\phi_i| \le n\alpha$. We can derive the following relationship between $\widehat{\omega}$ and $\widetilde{\omega}$:

$$\sum_{i=1}^{n}(\widehat{\omega}^{\top}\phi_i - (\omega^{\star})^{\top}\phi_i)^2 \le 2\sum_{i=1}^{n}(\widehat{\omega}^{\top}\phi_i - \widetilde{\omega}^{\top}\phi_i)^2 + 2\sum_{i=1}^{n}(\widetilde{\omega}^{\top}\phi_i - (\omega^{\star})^{\top}\phi_i)^2 \le 2n^2\alpha^2 + 2\sum_{i=1}^{n}(\widetilde{\omega}^{\top}\phi_i - (\omega^{\star})^{\top}\phi_i)^2,$$

$$\sum_{i=1}^{n} z_i^{\widetilde{\omega}} - \sum_{i=1}^{n} z_i^{\widehat{\omega}} = \sum_{i=1}^{n}(\widetilde{\omega}^{\top}\phi_i - \widehat{\omega}^{\top}\phi_i)(\widetilde{\omega}^{\top}\phi_i + \widehat{\omega}^{\top}\phi_i - 2r_i) \le 4(1 + \varepsilon_2)n\alpha.$$

Now plug $\widetilde{\omega}$ into (9) and re-arrange terms, we get:

$$\sum_{i=1}^{n}(\widetilde{\omega}^{\top}\phi_i - (\omega^{\star})^{\top}\phi_i)^2 \le \frac{1}{1 - 16(1 + \varepsilon_2)^2\eta}\sum_{i=1}^{n} z_i^{\widetilde{\omega}} + \frac{8(1 + \varepsilon_2)^2}{\eta(1 - 16(1 + \varepsilon_2)^2\eta)} \cdot \log(|\mathcal{C}|/\delta) + \frac{4(1 + \varepsilon_2)\varepsilon_{\mathrm{B}}T}{1 - 16(1 + \varepsilon_2)^2\eta}.$$

Setting $\eta = (32(1 + \varepsilon_2)^2)^{-1}$, we get

$$\sum_{i=1}^{n}(\widetilde{\omega}^{\top}\phi_i - (\omega^{\star})^{\top}\phi_i)^2 \le 2\sum_{i=1}^{n} z_i^{\widetilde{\omega}} + 512(1 + \varepsilon_2)^4 \log(|\mathcal{C}|/\delta) + 8(1 + \varepsilon_2)\varepsilon_{\mathrm{B}}T.$$

Using the relationships between $\widehat{\omega}$ and $\widetilde{\omega}$ that we derived above, we have:

$$\begin{aligned}
&\sum_{i=1}^{n}(\widehat{\omega}^{\top}\phi_i - (\omega^{\star})^{\top}\phi_i)^2 \\
&\le 2n^2\alpha^2 + 4\sum_{i=1}^{n} z_i^{\widetilde{\omega}} + 1024(1 + \varepsilon_2)^4 \log(|\mathcal{C}|/\delta) + 16(1 + \varepsilon_2)\varepsilon_{\mathrm{B}}T. \\
&\le 2n^2\alpha^2 + 4\sum_{i=1}^{n} z_i^{\widehat{\omega}} + 1024(1 + \varepsilon_2)^4 \log(|\mathcal{C}|/\delta) + 16(1 + \varepsilon_2)n\alpha + 16(1 + \varepsilon_2)\varepsilon_{\mathrm{B}}T.
\end{aligned}$$

Since $\widehat{\omega}$ is the (approximate) least square solution, we have $\sum_i z_i^{\widehat{\omega}} \le \varepsilon_1^2$. This implies that:

$$\sum_{i=1}^{n}(\widehat{\omega}^{\top}\phi_i - (\omega^{\star})^{\top}\phi_i)^2 \le 2n^2\alpha^2 + 4\varepsilon_1^2 + 1024(1 + \varepsilon_2)^4 \log(|\mathcal{C}|/\delta) + 16(1 + \varepsilon_2)(n\alpha + \varepsilon_{\mathrm{B}}T).$$

Now plugging the covering number (8) and setting $\alpha = 1/n$, we obtain

$$\sum_{i=1}^{n}(\widehat{\omega}^\top\phi_i - (\omega^\star)^\top\phi_i)^2 \le 2 + 4\varepsilon_1^2 + 1024(1+\varepsilon_2)^4 d\log(8(1+\varepsilon_2)e^2 n/\delta) + 16(1+\varepsilon_2)(1+\varepsilon_{\mathrm{B}}T)$$

$$\le 1026(1+\varepsilon_2)^4 d\log(8(1+\varepsilon_2)e^2 n/\delta) + 4\varepsilon_1^2 + 16(1+\varepsilon_2)(1+\varepsilon_{\mathrm{B}}T).$$

Finally, we have

$$\|\widehat{\omega} - \omega_h^\star\|_{\Sigma_t}^2 = \sum_{i=1}^{n}(\widehat{\omega}^\top\phi_i - (\omega^\star)^\top\phi_i)^2 + \sum_{i=1}^{m}\rho_i(\widehat{\omega}^\top\phi_i - (\omega^\star)^\top\phi_i)^2.$$

Here, with some abuse of notation, the $\phi_i$'s in the right term are the support points of the optimal design. The first term is already bounded above. The second term can be bounded by

$$\sum_{i=1}^{m}\rho_i(\widehat{\omega}^\top\phi_i - (\omega^\star)^\top\phi_i)^2 \le \sum_{i=1}^{m}\rho_i \cdot 4(1+\varepsilon_2) = 4(1+\varepsilon_2).$$

We add it into the constant of the first term. Then, we apply the union bound over all $t \in [T]$ and $h \in [H]$ to get the desired result. $\qquad\square$

**Lemma 7** (Value function estimation). *Suppose that* $\mathcal{T}(\overline{\omega}_{t,h} + \overline{\theta}_{t,h+1}) \in \mathcal{O}(W_h)$. *Then,*

$$\sum_{i=1}^{t-1}\left(\langle\widehat{\theta}_{t,h}, \phi(s_{i,h}, a_{i,h})\rangle - \overline{V}_{t,h+1}(s_{i,h+1})\right)^2 \le \varepsilon_1^2 + T\varepsilon_{\mathrm{B}}^2.$$

*Furthermore,* $\|\widehat{\theta}_{t,h} - \mathcal{T}(\overline{\omega}_{t,h} + \overline{\theta}_{t,h+1})\|_{\widehat{\Sigma}_{t,h}} \le \sqrt{2\varepsilon_1^2 + 4T\varepsilon_{\mathrm{B}}^2} =: B_{\mathrm{err}}^{\mathrm{P}}$.

**Proof of Lemma 7.** The Bayes optimal $\mathcal{T}(\overline{\omega}_{t,h} + \overline{\theta}_{t,h+1})$ should achieve the empirical risk of at most $\varepsilon_{\mathrm{B}}$, i.e.,

$$\forall i \in [t-1]: \quad \left|\langle\phi(s_{i,h}, a_{i,h}), \mathcal{T}(\overline{\omega}_{t,h} + \overline{\theta}_{t,h+1})\rangle - \overline{V}_{t,h+1}(s_{i,h+1})\right| \le \varepsilon_{\mathrm{B}}.$$

Since $\mathcal{T}(\overline{\omega}_{t,h} + \overline{\theta}_{t,h+1})$ is realizable (i.e., $\mathcal{T}(\overline{\omega}_{t,h} + \overline{\theta}_{t,h+1}) \in \mathcal{O}(W_h)$), and $\widehat{\theta}_{t,h}$ minimizes the objective up to precision $\varepsilon_1$, it should satisfy the following

$$\sum_{i=1}^{t-1}\left(\langle\widehat{\theta}_{t,h}, \phi(s_{i,h}, a_{i,h})\rangle - \overline{V}_{t,h+1}(s_{i,h+1})\right)^2 \le \varepsilon_1^2 + T\varepsilon_{\mathrm{B}}^2.$$

Combining the above two results, we arrive at the following:

$$\sum_{i=1}^{t-1}\langle\phi(s_{i,h}, a_{i,h}), \widehat{\theta}_{t,h} - \mathcal{T}(\overline{\omega}_{t,h} + \overline{\theta}_{t,h+1})\rangle^2$$

$$\le 2\sum_{i=1}^{t-1}\left(\langle\phi(s_{i,h}, a_{i,h}), \widehat{\theta}_{t,h}\rangle - \overline{V}_{t,h+1}(s_{i,h+1})\right)^2 + 2\sum_{i=1}^{t-1}\left(\overline{V}_{t,h+1}(s_{i,h+1}) - \langle\phi(s_{i,h}, a_{i,h}), \mathcal{T}(\overline{\omega}_{t,h} + \overline{\theta}_{t,h+1})\rangle\right)^2$$

$$\text{(using } (a+b)^2 \le 2a^2 + 2b^2\text{)}$$

$$\le 2\varepsilon_1^2 + 4T\varepsilon_{\mathrm{B}}^2.$$

This implies that

$$\left\|\widehat{\theta}_{t,h} - \mathcal{T}(\overline{\omega}_{t,h} + \overline{\theta}_{t,h+1})\right\|_{\widehat{\Sigma}_{t,h}}^2 \le 2\varepsilon_1^2 + 4T\varepsilon_{\mathrm{B}}^2.$$

$\qquad\square$

**Definition 5** (High-probability events). *Define event* $\mathfrak{E}^{\mathrm{high}}$ *as*

$$\mathfrak{E}^{\mathrm{high}} := \left\{\forall t \in [T], \forall h \in [H]: \|\xi_{t,h}^{\mathrm{P}}\|_{\Lambda_{t,h}} \le \sigma_h\sqrt{2d\log(6dH^2T^2)} =: B_{\mathrm{noise},h}^{\mathrm{P}}\right\}$$

$$\cap \left\{\forall t \in [T], \forall h \in [H]: \|\xi_{t,h}^{\mathrm{R}}\|_{\Sigma_{t,h}} \le \sigma_{\mathrm{R}}\sqrt{2d\log(6dHT^2)} =: B_{\mathrm{noise}}^{\mathrm{R}}\right\}$$

$$\cap \left\{\forall t \in [T], \forall h \in [H]: \|\eta_{t,h}^{\mathrm{R}}\|_{\Sigma_{t,h}} \le B_{\mathrm{err}}^{\mathrm{R}}\right\}$$

*where* $B_{\mathrm{err}}^{\mathrm{R}} := \sqrt{1030(1+\varepsilon_2)^4 d\log\left(24(1+\varepsilon_2)e^2 T^3 H^2\right) + 4\varepsilon_1^2 + 16(1+\varepsilon_2)(1+\varepsilon_{\mathrm{B}}T)}$.

**Lemma 8.** *We have* $\Pr(\mathfrak{E}^{\mathrm{high}}) > 1 - 1/(HT)$.

**Proof of Lemma 8.** Below we show that each event defined in Definition 5 holds with probability at least $1 - 1/(3HT)$. Then, by union bound, we have the desired result.

*Proof of the first event.* The way we generate $\xi_{t,h}^{\mathrm{P}}$ is equivalent to first sampling $\zeta_{t,h} \sim \mathcal{N}(0, (\sigma_h)^2 \Lambda_{t,h}^{-1})$ and then set $\xi_{t,h}^{\mathrm{P}} \leftarrow (I - P_{t,h})\zeta_{t,h}$. By Lemma 20 and the union bound, we have

$$\Pr\left(\forall t \in [T], \forall h \in [H] : \|\zeta_{t,h}\|_{\Lambda_{t,h}} > \sigma_h \sqrt{2d \log(6dH^2 T^2)}\right) \leq 1/(3HT).$$

Then, by definition, we have

$$
\begin{aligned}
\|\xi_{t,h}^{\mathrm{P}}\|_{\Lambda_{t,h}}^2 &= \|(1 - P_{t,h})\zeta_{t,h}\|_{\Lambda_{t,h}}^2 \\
&= \zeta_{t,h}^{\top}(I - P_{t,h}) \sum_{i=1}^m \left(\phi_{t,h,i}^{\|}(\phi_{t,h,i}^{\|})^{\top} + \phi_{t,h,i}^{\perp}(\phi_{t,h,i}^{\perp})^{\top}\right)(I - P_{t,h})\zeta_{t,h} \\
&= \zeta_{t,h}^{\top} \sum_{i=1}^m \phi_{t,h,i}^{\perp}(\phi_{t,h,i}^{\perp})^{\top}\zeta_{t,h} \\
&\leq \zeta_{t,h}^{\top} \sum_{i=1}^m \left(\phi_{t,h,i}^{\|}(\phi_{t,h,i}^{\|})^{\top} + \phi_{t,h,i}^{\perp}(\phi_{t,h,i}^{\perp})^{\top}\right)\zeta_{t,h}
\end{aligned}
$$

where the third step holds by the fact that $\phi^{\perp}$ is in the null space and $\phi^{\|}$ is in the span. Hence, we conclude that $\|\xi_{t,h}^{\mathrm{P}}\|_{\Lambda_{t,h}} \leq \|\zeta_{t,h}\|_{\Lambda_{t,h}}$.

*Proof of the second event.* Applying Lemma 20 and the union bound, we have

$$\Pr\left(\forall t \in [T] : \|\xi_t^{\mathrm{R}}\|_{\Sigma_t} > \sigma_{\mathrm{R}} \sqrt{2d \log(6dHT^2)}\right) \leq 1/(3HT).$$

*Proof of the third event.* This is directly from Lemma 6. $\qquad\square$

**Lemma 9** (Boundness of parameters). *Under Assumption 4, conditioning on $\mathfrak{E}^{\mathrm{high}}$, the following hold for all $t \in [T]$ and $h \in [H]$:*

1. $\max_{s,a} |\langle \phi(s,a), \widehat{\theta}_{t,h} \rangle| \leq W_h + \varepsilon_2$;

2. $\max_{s,a} |\langle \phi(s,a), \mathcal{T}(\overline{\omega}_{t,h} + \overline{\theta}_{t,h+1}) \rangle| \leq W_h$;

3. $\|\eta_{t,h}\|_{\widehat{\Sigma}_{t,h}} \leq B_{\mathrm{err}}^{\mathrm{P}}$;

4. $\|\eta_{t,h}\|_{\Lambda} \leq 2(W_h + \varepsilon_2)\sqrt{m}$;

5. $\|\eta_{t,h}\|_{\Lambda_{t,h}} \leq \sqrt{3}\gamma B_{\mathrm{err}}^{\mathrm{P}} + \sqrt{8m}(W_h + \varepsilon_2)$ ;

6. $\max_{s,a} |\langle \phi(s,a), \overline{\theta}_{t,h} \rangle| \leq W_{h-1} - \sqrt{2d} \cdot B_{\mathrm{noise}}^{\mathrm{R}} - 1 - \varepsilon_2$

7. $\max_s \overline{V}_{t,h}(s) = \max_{s,a} |\overline{Q}_{t,h}(s,a)| \leq W_{h-1}$.

**Proof of Lemma 9.** Fix $t \in [T]$. We prove these items by induction on $h$. The base case ($h = H+1$) clearly holds since there is actually nothing at $(H+1)$-th step. Now assume they hold for $h+1$, and we will show that they hold for $h$ as well.

*Proof of Item 1.* It is simply by Line 9 of Algorithm 1 and Assumption 3.

*Proof of Item 2.* By linear Bellman completeness (Definition 1), for any $s, a$, we have,

$$
\begin{aligned}
|\langle \phi(s,a), \mathcal{T}(\overline{\omega}_{t,h} + \overline{\theta}_{t,h+1}) \rangle| &= \left| \mathbb{E}_{s' \sim \mathsf{T}(s,a)} \max_{a'} \langle \phi(s',a'), \overline{\omega}_{t,h} + \overline{\theta}_{t,h+1} \rangle \right| \\
&\leq \max_{s,a} \left| \langle \phi(s,a), \overline{\omega}_{t,h} + \overline{\theta}_{t,h+1} \rangle \right|
\end{aligned}
$$

$$\le \max_{s,a}\left|\langle\phi(s,a),\widehat{\omega}_{t,h}\rangle\right| + \max_{s,a}\left|\langle\phi(s,a),\xi_{t,h}^{\mathrm{R}}\rangle\right| + \max_{s,a}\left|\langle\phi(s,a),\overline{\theta}_{t,h+1}\rangle\right|$$

$$\le (1+\varepsilon_2) + \max_{s,a}\|\phi(s,a)\|_{\Sigma_{t,h}^{-1}}\|\xi_{t,h}^{\mathrm{R}}\|_{\Sigma_{t,h}} + (W_h - \sqrt{2d}\cdot B_{\mathrm{noise}}^{\mathrm{R}} - 1 - \varepsilon_2)$$

$$\le 1 + \varepsilon_2 + \sqrt{2d}\cdot B_{\mathrm{noise}}^{\mathrm{R}} + (W_h - \sqrt{2d}\cdot B_{\mathrm{noise}}^{\mathrm{R}} - 1 - \varepsilon_2) = W_h.$$

*Proof of Item 3.* This is directly from Lemma 7.

*Proof of Item 4.* By triangle inequality, we have

$$\|\eta_{t,h}\|_\Lambda = \left\|\widehat{\theta}_{t,h} - \mathcal{T}(\overline{\omega}_{t,h} + \overline{\theta}_{t,h+1})\right\|_\Lambda \le \left\|\widehat{\theta}_{t,h}\right\|_\Lambda + \left\|\mathcal{T}(\overline{\omega}_{t,h} + \overline{\theta}_{t,h+1})\right\|_\Lambda \le 2(W_h + \varepsilon_2)\sqrt{m}.$$

where the last step is by

$$\left\|\widehat{\theta}_{t,h}\right\|_\Lambda = \sqrt{\sum_{i=1}^m \langle\phi_i, \widehat{\theta}_{t,h}\rangle^2} \le \sqrt{\sum_{i=1}^m (W_h + \varepsilon_2)^2} = (W_h + \varepsilon_2)\sqrt{m}$$

and the similar for $\mathcal{T}(\overline{\omega}_{t,h} + \overline{\theta}_{t,h+1})$.

*Proof of Item 5.* By definition, we have

$$\|\eta_{t,h}\|_{\Lambda_{t,h}}^2 = \sum_{i=1}^m \rho_i\left(\langle\phi_{t,h,i}^\|, \eta_{t,h}\rangle^2 + \langle\phi_{t,h,i}^\perp, \eta_{t,h}\rangle^2\right)$$

$$= \sum_{i=1}^m \rho_i\left(\langle P_{t,h}\phi_i, \eta_{t,h}\rangle^2 + \langle(I - P_{t,h})\phi_i, \eta_{t,h}\rangle^2\right)$$

$$\le \sum_{i=1}^m \rho_i\left(3\langle P_{t,h}\phi_i, \eta_{t,h}\rangle^2 + 2\langle\phi_i, \eta_{t,h}\rangle^2\right) \qquad \text{(using } (a+b)^2 \le a^2 + b^2)$$

$$\le 3\sum_{i=1}^m \rho_i\left(\left\|\phi_{t,h,i}^\|\right\|_{\widehat{\Sigma}_{t,h}^\dagger}^2 \|\eta_{t,h}\|_{\widehat{\Sigma}_{t,h}}^2\right) + 2\|\eta_{t,h}\|_\Lambda^2 \qquad \text{(Cauchy-Schwartz, Lemma 25)}$$

We have $\|\phi_{t,h,i}^\|\|_{\widehat{\Sigma}_{t,h}^\dagger} = \|P_{t,h}\phi_i\|_{\widehat{\Sigma}_{t,h}^\dagger} = \|\phi_i\|_{\widehat{\Sigma}_{t,h}^\dagger}$ by Lemma 26. By Assumption 4, this is upper bounded by $\gamma$. The second term, $\|\eta_{t,h}\|_{\widehat{\Sigma}_{t,h}}$, is upper bounded by $B_{\mathrm{err}}^{\mathrm{P}}$ by Item 3.

Hence, we have

$$\|\eta_{t,h}\|_{\Lambda_{t,h}}^2 \le 3\gamma^2 (B_{\mathrm{err}}^{\mathrm{P}})^2 + 2\|\eta_{t,h}\|_\Lambda^2$$

$$\le 3\gamma^2 (B_{\mathrm{err}}^{\mathrm{P}})^2 + 8(W_h + \varepsilon_2)^2 m. \qquad \text{(Item 4)}$$

*Proof of Item 6.* We have

$$\max_{s,a}\left|\langle\phi(s,a),\overline{\theta}_{t,h}\rangle\right| = \max_{s,a}\left|\langle\phi(s,a),\widehat{\theta}_{t,h} + \xi_{t,h}^{\mathrm{P}}\rangle\right|$$

$$\le \max_{s,a}\left|\langle\phi(s,a),\widehat{\theta}_{t,h}\rangle\right| + \max_{s,a}\left|\langle\phi(s,a),\xi_{t,h}^{\mathrm{P}}\rangle\right|$$

$$\le W_h + \varepsilon_2 + \max_{s,a}\|\phi(s,a)\|_{\Lambda_{t,h}^{-1}}\|\xi_{t,h}^{\mathrm{P}}\|_{\Lambda_{t,h}}$$

$$\le W_h + \varepsilon_2 + \sqrt{2d}\cdot B_{\mathrm{noise},h}^{\mathrm{P}} \qquad \text{(Lemma 5)}$$

$$= W_{h-1} - \sqrt{2d}\cdot B_{\mathrm{noise}}^{\mathrm{R}} - 1 - \varepsilon_2.$$

*Proof of Item 7.* We have

$$|\overline{Q}_{t,h}(s,a)| = |\langle\phi(s,a),\overline{\theta}_{t,h}\rangle + \langle\phi(s,a),\overline{\omega}_{t,h}\rangle|$$

$$\le |\langle\phi(s,a),\overline{\theta}_{t,h}\rangle| + |\langle\phi(s,a),\widehat{\omega}_{t,h}\rangle| + |\langle\phi(s,a),\xi_t^{\mathrm{R}}\rangle|$$

$$\le (W_{h-1} - \sqrt{2d}\cdot B_{\mathrm{noise}}^{\mathrm{R}} - 1 - \varepsilon_2) + (1 + \varepsilon_2) + \sqrt{2d}\cdot B_{\mathrm{noise}}^{\mathrm{R}}$$

$$= W_{h-1}.$$

and also, $\max_s |\overline{V}_{t,h}(s)| = \max_{s,a}|\overline{Q}_{t,h}(s,a)| \le W_{h-1}.$ $\qquad\qquad \square$

## C.2 VALUE DECOMPOSITION

We note that, at any round $t \in [T]$, conditioning on all information collected up to round $t-1$, the randomness of $\overline{V}_t$ only comes from the Gaussian noise $\{\xi_{t,h}^{\mathrm{P}}, \xi_{t,h}^{\mathrm{R}}\}_{h=1}^H$. In other words, $\overline{V}_t$ can be considered *a functional of the Gaussian noise*. In light of this, we define

$$V_{t,h}[\check{\xi}_1^{\mathrm{P}}, \ldots, \check{\xi}_H^{\mathrm{P}}, \check{\xi}_1^{\mathrm{R}}, \ldots, \check{\xi}_H^{\mathrm{R}}](\cdot)$$

as a functional of the noise variable, which maps the given noise variable to the value function produced by the algorithm by replacing the random Gaussian noise with the variable $\check{\xi}_1^{\mathrm{P}}, \ldots, \check{\xi}_H^{\mathrm{P}}, \check{\xi}_1^{\mathrm{R}}, \ldots, \check{\xi}_H^{\mathrm{R}}$. By definition, we immediately have

$$\overline{V}_{t,h}(\cdot) = V_{t,h}[\xi_{t,1}^{\mathrm{P}}, \ldots, \xi_{t,H}^{\mathrm{P}}, \xi_{t,1}^{\mathrm{R}}, \ldots, \xi_{t,H}^{\mathrm{R}}](\cdot).$$

Next, we define $U_t$ as the minimum of the following program

$$\min_{\check{\xi}_1^{\mathrm{P}}, \ldots, \check{\xi}_H^{\mathrm{P}}, \check{\xi}_1^{\mathrm{R}}, \ldots, \check{\xi}_H^{\mathrm{R}}} V_{t,1}\left[\check{\xi}_1^{\mathrm{P}}, \ldots, \check{\xi}_H^{\mathrm{P}}, \check{\xi}_1^{\mathrm{R}}, \ldots, \check{\xi}_H^{\mathrm{R}}\right](s_{t,1})$$

$$\text{s.t.} \quad \forall h \in [H] : \|\check{\xi}_{t,h}^{\mathrm{P}}\|_{\Lambda_{t,h}} \le B_{\mathrm{noise},h}^{\mathrm{P}}, \quad \|\check{\xi}_{t,h}^{\mathrm{R}}\|_{\Sigma_{t,h}} \le B_{\mathrm{noise}}^{\mathrm{R}}.$$

In other words, $U_t$ achieves the minimum value at $s_{t,1}$ while satisfying the high-probability constraints ($\mathfrak{E}^{\mathrm{high}}$) on the noise variable. We denote $\underline{\xi}_1^{\mathrm{P}}, \ldots, \underline{\xi}_H^{\mathrm{P}}, \underline{\xi}_1^{\mathrm{R}}, \ldots, \underline{\xi}_H^{\mathrm{R}}$ as the minimizer of the above program, and will always use underlined variables to represent the intermediate variables corresponding to $U_t$ (such as $\underline{\widehat{\theta}}, \underline{\overline{\theta}}, \underline{\widehat{\omega}}, \underline{\overline{\omega}}, \underline{\overline{Q}}, \underline{\overline{V}}$) to distinguish them from the variables corresponding to $\overline{V}_t$, $(\widehat{\theta}, \overline{\theta}, \widehat{\omega}, \overline{\omega}, \overline{Q}, \overline{V})$. We note that, under $\mathfrak{E}^{\mathrm{high}}$, we directly have $U_t(s_{t,1}) \le \overline{V}_t(s_{t,1})$.

Below is a decomposition lemma under deterministic transition. Note that it slightly differs from the usual value decomposition lemma under stochastic transitions, where we have to take the expectation over trajectory randomness. This distinction is crucial to our analysis: by not accounting for trajectory randomness, we can effectively leverage our span argument.

We denote $\{s_{t,h}, a_{t,h}\}_{h=1}^H$ as the trajectory generated by executing $\pi_t$ with initial state $s_{t,1}$, and $\{s_{t,h}^\star, a_{t,h}^\star\}_{h=1}^H$ as the trajectory generated by executing $\pi^\star$ with initial state $s_{t,1}^\star = s_{t,1}$.

**Lemma 10** (Value decomposition under deterministic transition). *Under deterministic transition (Assumption 1), we have*

$$V^{\pi_t}(s_{t,1}) - \overline{V}_t(s_{t,1}) = \sum_{h=1}^H \left( \overline{V}_{t,h+1}(s_{t,h+1}) - \langle \overline{\theta}_{t,h}, \phi(s_{t,h}, a_{t,h}) \rangle + \langle \omega_h^\star - \overline{\omega}_{t,h}, \phi(s_{t,h}, a_{t,h}) \rangle \right); \tag{10}$$

$$V^\star(s_{t,1}) - \overline{V}_t(s_{t,1}) \le \sum_{h=1}^H \left( \overline{V}_{t,h+1}(s_{t,h+1}^\star) - \langle \overline{\theta}_{t,h}, \phi(s_{t,h}^\star, a_{t,h}^\star) \rangle + \langle \omega_h^\star - \overline{\omega}_{t,h}, \phi(s_{t,h}^\star, a_{t,h}^\star) \rangle \right). \tag{11}$$

*Similarly, we have*

$$V^{\pi_t}(s_{t,1}) - U_t(s_{t,1}) \le \sum_{h=1}^H \left( U_{t,h+1}(s_{t,h+1}) - \langle \underline{\overline{\theta}}_{t,h}, \phi(s_{t,h}, a_{t,h}) \rangle + \langle \omega_h^\star - \underline{\overline{\omega}}_{t,h}, \phi(s_{t,h}, a_{t,h}) \rangle \right). \tag{12}$$

**Proof of Lemma 10.** We will prove (10) and (11) altogether, and then prove (12).

*Proof of* (10) *and* (11). We consider an arbitrary policy $\pi$. Let $\{s_{t,h}', a_{t,h}'\}_{h=1}^H$ denote the deterministic trajectory generated by $\pi$ with initial state $s_{t,1}' = s_{t,1}$. By definition, we have

$$\begin{aligned}
&V^\pi(s_{t,1}') - \overline{V}_t(s_{t,1}') \\
&= Q_1^\pi(s_{t,1}', \pi(s_{t,1}')) - \max_a \overline{Q}_{t,1}(s_{t,1}', a) \\
&\le Q_1^\pi(s_{t,1}', \pi(s_{t,1}')) - \overline{Q}_{t,1}(s_{t,1}', \pi(s_{t,1}')) \\
&= V_2^\pi(s_{t,2}') + r_h(s_{t,1}', a_{t,1}') - \langle \overline{\theta}_{t,1}, \phi(s_{t,1}', \pi(s_{t,1}')) \rangle - \langle \overline{\omega}_{t,h}, \phi(s_{t,1}', \pi(s_{t,1}')) \rangle \quad \text{(by definition)}
\end{aligned} \tag{13}$$

$$= \left(V_2^\pi(s_{t,2}') - \overline{V}_{t,2}(s_{t,2}')\right) + \left(\overline{V}_{t,2}(s_{t,2}') - \langle\overline{\theta}_{t,1}, \phi(s_{t,1}', \pi(s_{t,1}'))\rangle\right) + \langle\omega_h^\star - \overline{\omega}_{t,h}, \phi(s_{t,1}', a_{t,1}')\rangle$$

Recursively expanding the first term, we obtain

$$V^\pi(s_{t,1}') - \overline{V}_t(s_{t,1}') \le \sum_{h=1}^H \left(\overline{V}_{t,h+1}(s_{t,h+1}') - \langle\overline{\theta}_{t,h}, \phi(s_{t,h}', a_{t,h}')\rangle + \langle\omega_h^\star - \overline{\omega}_{t,h}, \phi(s_{t,h}', a_{t,h}')\rangle\right).$$

This proves (11) by specifying $\pi = \pi^\star$. Similarly, (10) can be proved by observing that the only inequality (13) becomes equality when $\pi = \pi_t$.

*Proof of* (12). The proof is quite similar. We have

$$\begin{aligned}
&V^{\pi_t}(s_{t,1}) - U_t(s_{t,1}) \\
&= Q_1^{\pi_t}(s_{t,1}, \pi_t(s_{t,1})) - \max_a \overline{Q}_{t,1}(s_{t,1}, a) \\
&\le Q_1^{\pi_t}(s_{t,1}, \pi_t(s_{t,1})) - \underline{Q}_{t,1}(s_{t,1}, \pi_t(s_{t,1})) \\
&= V_2^{\pi_t}(s_{t,2}) + r_h(s_{t,1}, a_{t,1}) - \langle\underline{\theta}_{t,1}, \phi(s_{t,1}, \pi_t(s_{t,1}))\rangle - \langle\underline{\omega}_{t,h}, \phi(s_{t,1}, a_{t,1})\rangle \qquad \text{(by definition)}
\end{aligned}$$

$$= \left(V_2^{\pi_t}(s_{t,2}) - U_{t,2}(s_{t,2})\right) + \left(U_{t,2}(s_{t,2}) - \langle\underline{\theta}_{t,1}, \phi(s_{t,1}, \pi_t(s_{t,1}))\rangle\right) + \langle\omega_h^\star - \underline{\omega}_{t,h}, \phi(s_{t,1}, a_{t,1})\rangle$$

Recursively expanding the first term, we obtain

$$V^{\pi_t}(s_{t,1}) - U_t(s_{t,1}) \le \sum_{h=1}^H \left(U_{t,h+1}(s_{t,h+1}) - \langle\underline{\theta}_{t,h}, \phi(s_{t,h}, a_{t,h})\rangle + \langle\omega_h^\star - \underline{\omega}_{t,h}, \phi(s_{t,h}, a_{t,h})\rangle\right).$$

This completes the proof. $\qquad\square$

**Lemma 11.** *For any $t \in [T]$, conditioning on $\mathfrak{E}_t^{\mathrm{span}}$, we have the following (in)equalities:*

$$\overline{V}_t(s_{t,1}) = \sum_{h=1}^H \left(\langle\widehat{\theta}_{t,h} - \mathcal{T}(\overline{\theta}_{t,h+1} + \overline{\omega}_{t,h+1}), \phi(s_{t,h}, a_{t,h})\rangle + \langle\overline{\omega}_{t,h}, \phi(s_{t,h}, a_{t,h})\rangle\right),$$

$$U_t(s_{t,1}) \ge \sum_{h=1}^H \left(\langle\widehat{\underline{\theta}}_{t,h} - \mathcal{T}(\underline{\theta}_{t,h+1} + \underline{\omega}_{t,h+1}), \phi(s_{t,h}, a_{t,h})\rangle + \langle\underline{\omega}_{t,h}, \phi(s_{t,h}, a_{t,h})\rangle\right).$$

**Proof of Lemma 11.** We will prove the two statements separately, but the proofs are quite similar.

*Proof of the first statement.* By Lemma 10, we have

$$\begin{aligned}
&\overline{V}_t(s_{t,1}) - V^{\pi_t}(s_{t,1}) \\
&= \sum_{h=1}^H \left(\langle\widehat{\theta}_{t,h}, \phi(s_{t,h}, a_{t,h})\rangle + \langle\xi_{t,h}^{\mathrm{P}}, \phi(s_{t,h}, a_{t,h})\rangle - \overline{V}_{t,h+1}(s_{t,h+1}) + \langle\overline{\omega}_{t,h} - \omega_h^\star, \phi(s_{t,h}, a_{t,h})\rangle\right)
\end{aligned}$$

By linear Bellman completeness (Definition 1), there exists a vector, denoted by $\mathcal{T}(\overline{\theta}_{t,h+1} + \overline{\omega}_{t,h+1})$, such that $\overline{V}_{t,h+1}(\cdot) = \langle\phi(\cdot, a), \mathcal{T}(\overline{\theta}_{t,h+1} + \overline{\omega}_{t,h+1})\rangle$. Hence, we can rewrite the above as

$$\begin{aligned}
&\overline{V}_t(s_{t,1}) - V^{\pi_t}(s_{t,1}) \\
&= \sum_{h=1}^H \left(\langle\widehat{\theta}_{t,h} - \mathcal{T}(\overline{\theta}_{t,h+1} + \overline{\omega}_{t,h+1}), \phi(s_{t,h}, a_{t,h})\rangle + \langle\xi_{t,h}^{\mathrm{P}}, \phi(s_{t,h}, a_{t,h})\rangle + \langle\overline{\omega}_{t,h} - \omega_h^\star, \phi(s_{t,h}, a_{t,h})\rangle\right).
\end{aligned}$$

Note that by definition of $V^{\pi_t}$ we have $V^{\pi_t}(s_{t,1}) = \sum_{h=1}^H \langle\omega^\star, \phi(s_{t,h}, a_{t,h})\rangle$. Hence, the above implies

$$\overline{V}_t(s_{t,1}) = \sum_{h=1}^H \left(\langle\widehat{\theta}_{t,h} - \mathcal{T}(\overline{\theta}_{t,h+1} + \overline{\omega}_{t,h+1}) + \xi_{t,h}^{\mathrm{P}}, \phi(s_{t,h}, a_{t,h})\rangle + \langle\overline{\omega}_{t,h}, \phi(s_{t,h}, a_{t,h})\rangle\right).$$

We can remove $\xi_{t,h}^{\mathrm{P}}$ since $\langle\xi_{t,h}^{\mathrm{P}}, \phi(s_{t,h}, a_{t,h})\rangle = 0$ conditioning on $\mathfrak{E}_t^{\mathrm{span}}$.

*Proof of the second statement.* By Lemma 10, we have

$$V^{\pi_t}(s_{t,1}) - U_t(s_{t,1}) \le \sum_{h=1}^{H} \left( U_{t,h+1}(s_{t,h+1}) - \langle \underline{\overline{\theta}}_{t,h}, \phi(s_{t,h}, a_{t,h}) \rangle + \langle \omega_h^\star - \overline{\omega}_{t,h}, \phi(s_{t,h}, a_{t,h}) \rangle \right)$$

Again, by the definition of $V^{\pi_t}$, we conclude that

$$U_t(s_{t,1}) \ge \sum_{h=1}^{H} \left( \langle \overline{\omega}_{t,h}, \phi(s_{t,h}, a_{t,h}) \rangle + \langle \underline{\widehat{\theta}}_{t,h} - \mathcal{T}(\overline{\theta}_{t,h+1} + \overline{\omega}_{t,h+1}) + \underline{\xi}_{t,h}^{\mathrm{P}}, \phi(s_{t,h}, a_{t,h}) \rangle \right).$$

We can remove $\underline{\xi}_{t,h}^{\mathrm{P}}$ since $\langle \underline{\xi}_{t,h}^{\mathrm{P}}, \phi(s_{t,h}, a_{t,h}) \rangle = 0$ conditioning on $\mathfrak{E}_t^{\mathrm{span}}$. $\qquad \square$

The following lemma shows that, conditioning on the span event $\mathfrak{E}_t^{\mathrm{span}}$, the value function $\overline{V}_t$ cannot deviate too much from the value function $V^{\pi_t}$ on average.

**Lemma 12.** *For any $t \in [T]$, under Assumption 4 and conditioning on $\mathfrak{E}_t^{\mathrm{span}}$ and $\mathfrak{E}^{\mathrm{high}}$, we have*

$$\sum_{t=1}^{T} \left( \overline{V}_t(s_{t,1}) - V^{\pi_t}(s_{t,1}) \right) \le B_{\mathrm{err}}^{\mathrm{P}} \gamma H + (B_{\mathrm{noise}}^{\mathrm{R}} + B_{\mathrm{err}}^{\mathrm{R}}) \cdot B_\phi^{\mathrm{R}}.$$

**Proof of Lemma 12.** We apply Lemma 11 to decompose $\overline{V}_t$ and obtain

$$\sum_{t=1}^{T} \left( \overline{V}_t(s_{t,1}) - V^{\pi_t}(s_{t,1}) \right)$$

$$= \sum_{t=1}^{T} \left( \langle \widehat{\theta}_{t,h} - \mathcal{T}(\overline{\theta}_{t,h+1} + \overline{\omega}_{t,h+1}), \phi(s_{t,h}, a_{t,h}) \rangle + \langle \overline{\omega}_{t,h} - \omega_h^\star, \phi(s_{t,h}, a_{t,h}) \rangle \right)$$

Applying Cauchy-Schwartz yields

$$\le \sum_{t=1}^{T} \left( \left\| \widehat{\theta}_{t,h} - \mathcal{T}(\overline{\theta}_{t,h+1} + \overline{\omega}_{t,h+1}) \right\|_{\widehat{\Sigma}_{t,h}} \| \phi(s_{t,h}, a_{t,h}) \|_{\widehat{\Sigma}_{t,h}^\dagger} + \| \overline{\omega}_{t,h} - \omega_h^\star \|_{\Sigma_{t,h}} \| \phi(s_{t,h}, a_{t,h}) \|_{\Sigma_{t,h}^{-1}} \right)$$

We apply Lemma 7 and Assumption 4 to the left term and Lemmas 6 and 16 and Definition 5 to the right. Then, we obtain

$$\le H \cdot B_{\mathrm{err}}^{\mathrm{P}} \gamma + (B_{\mathrm{noise}}^{\mathrm{R}} + B_{\mathrm{err}}^{\mathrm{R}}) \cdot B_\phi^{\mathrm{R}}.$$

This completes the proof. $\qquad \square$

The following lemma establishes upper bounds on the value functions when conditioning on the span event $\mathfrak{E}_t^{\mathrm{span}}$.

**Lemma 13.** *For any $t \in [T]$, conditioning on $\mathfrak{E}_t^{\mathrm{span}}$ and $\mathfrak{E}^{\mathrm{high}}$, we have*

$$|U_t(s_{t,1})| \le H \cdot (B_{\mathrm{noise}}^{\mathrm{R}} + B_{\mathrm{err}}^{\mathrm{R}}) \cdot \sqrt{d} + H \cdot (1 + B_{\mathrm{err}}^{\mathrm{P}} \gamma).$$

*Moreover, we have*

$$|\overline{V}_t(s_{t,1})| \le H \cdot (B_{\mathrm{noise}}^{\mathrm{R}} + B_{\mathrm{err}}^{\mathrm{R}}) \cdot \sqrt{d} + H \cdot (1 + B_{\mathrm{err}}^{\mathrm{P}} \gamma).$$

*We abbreviate $B_V := H \cdot (B_{\mathrm{noise}}^{\mathrm{R}} + B_{\mathrm{err}}^{\mathrm{R}}) \cdot \sqrt{d} + H \cdot (1 + B_{\mathrm{err}}^{\mathrm{P}} \gamma)$.*

**Proof of Lemma 13.** We will first prove the second statement and then the first statement.

*Proof of the second statement.* Applying Lemma 11 and the triangle inequality, we have the following

$$|\overline{V}_t(s_{t,1})| \le \left| \sum_{h=1}^{H} \langle \overline{\omega}_{t,h}, \phi(s_{t,h}, a_{t,h}) \rangle \right| + \left| \sum_{h=1}^{H} \langle \widehat{\theta}_{t,h} - \mathcal{T}(\overline{\theta}_{t,h+1} + \overline{\omega}_{t,h+1}), \phi(s_{t,h}, a_{t,h}) \rangle \right|$$

$$=: \mathrm{T}_1 + \mathrm{T}_2.$$

We bound the two terms separately. For $\mathtt{T}_1$, we have

$$\mathtt{T}_1 = \left| \sum_{h=1}^{H} \langle (\overline{\omega}_{t,h} - \widehat{\omega}_{t,h}) + (\widehat{\omega}_{t,h} - \omega^\star) + \omega_h^\star, \phi(s_{t,h}, a_{t,h}) \rangle \right|$$

$$\leq \sum_{h=1}^{H} \left( \|\overline{\omega}_{t,h} - \widehat{\omega}_{t,h}\|_{\Sigma_{t,h}} + \|\widehat{\omega}_{t,h} - \omega_h^\star\|_{\Sigma_{t,h}} \right) \|\phi(s_{t,h}, a_{t,h})\|_{\Sigma_{t,h}^{-1}} + V^{\pi_t} \quad \text{(Cauchy-Schwartz)}$$

$$\leq H \cdot (B_{\text{noise}}^{\text{R}} + B_{\text{err}}^{\text{R}}) \cdot \sqrt{d} + H. \quad \text{(Definition 5 and lemma 5)}$$

For $\mathtt{T}_2$, we can use Cauchy-Schwartz:

$$\mathtt{T}_2 = \left| \sum_{h=1}^{H} \langle \widehat{\theta}_{t,h} - \mathcal{T}(\overline{\theta}_{t,h+1} + \overline{\omega}_{t,h+1}), \phi(s_{t,h}, a_{t,h}) \rangle \right|$$

$$\leq \sum_{h=1}^{H} \|\widehat{\theta}_{t,h} - \mathcal{T}(\overline{\theta}_{t,h+1} + \overline{\omega}_{t,h+1})\|_{\widehat{\Sigma}_{t,h}} \|\phi(s_{t,h}, a_{t,h})\|_{\widehat{\Sigma}_{t,h}^\dagger} \quad \text{(Cauchy-Schwartz, Lemma 25)}$$

$$\leq B_{\text{err}}^{\text{P}} \gamma H. \quad \text{(Assumption 4 and lemma 7)}$$

*Proof of the first statement.* We prove it by establishing a lower bound and an upper bound of $U_t(s_{t,1})$ separately. We start with the lower bound, whose derivation is similar to the second statement we just proved above:

$$U_t(s_{t,1}) \geq \sum_{h=1}^{H} \left( \langle \widehat{\theta}_{t,h} - \mathcal{T}(\overline{\theta}_{t,h+1} + \overline{\omega}_{t,h+1}), \phi(s_{t,h}, a_{t,h}) \rangle + \langle \overline{\omega}_{t,h}, \phi(s_{t,h}, a_{t,h}) \rangle \right) \quad \text{(Lemma 11)}$$

$$\geq -B_{\text{err}}^{\text{P}} \gamma H - \left| \sum_{h=1}^{H} \langle (\underline{\omega}_{t,h} - \widehat{\omega}_{t,h}) + (\widehat{\omega}_{t,h} - \omega_h^\star) + \omega_h^\star, \phi(s_{t,h}, a_{t,h}) \rangle \right|$$

$$\text{(following a similar argument as above)}$$

$$\geq -B_{\text{err}}^{\text{P}} \gamma H - \sum_{h=1}^{H} \left( \|\underline{\omega}_{t,h} - \widehat{\omega}_{t,h}\|_{\Sigma_{t,h}} + \|\widehat{\omega}_{t,h} - \omega_h^\star\|_{\Sigma_{t,h}} \right) \|\phi(s_{t,h}, a_{t,h})\|_{\Sigma_{t,h}^{-1}}$$

$$\text{(Cauchy-Schwartz)}$$

$$\geq -B_{\text{err}}^{\text{P}} \gamma H - H \cdot (B_{\text{noise}}^{\text{R}} + B_{\text{err}}^{\text{R}}) \cdot \sqrt{d}. \quad \text{(Lemma 8)}$$

The upper bound of $U_t(s_{t,1})$ is a consequence of the second statement we just proved above:

$$U_t(s_{t,1}) \leq \mathbb{E}[\overline{V}_t(s_{t,1}) \,|\, \mathfrak{E}^{\text{high}}] \quad \text{(by definition)}$$

$$\leq B_{\text{err}}^{\text{P}} \gamma H + H \cdot (B_{\text{noise}}^{\text{R}} + B_{\text{err}}^{\text{R}}) \cdot \sqrt{d} + H.$$

We finish the proof by combining the lower and upper bounds. $\qquad \square$

## C.3 EXPLORATION IN THE NULL SPACE

**Lemma 14** (optimism with constant probability). *For any $t \in [T]$, denote $\mathfrak{E}_t^{\text{optm}}$ as the event that*

$$V^\star(s_{t,1}) \leq \overline{V}_t(s_{t,1}) + B_{\text{err}}^{\text{P}} \gamma H.$$

*Then, under Assumption 4 and conditioning on the high-probability event $\mathfrak{E}^{\text{high}}$, we have*

$$\Pr\left( \mathfrak{E}_t^{\text{optm}} \right) \geq \Gamma^2(-1)$$

*where $\Gamma(\cdot)$ is the CDF of the standard normal distribution.*

**Proof of Lemma 14.** By Lemma 10, we have:

$$V^\star(s_{t,1}) - \overline{V}_t(s_{t,1}) \leq \sum_{h=1}^{H} \left( \overline{V}_{t,h+1}(s_{t,h+1}^\star) - \langle \overline{\theta}_{t,h}, \phi(s_{t,h}^\star, a_{t,h}^\star) \rangle + \langle \omega_h^\star - \overline{\omega}_{t,h}, \phi(s_{t,h}^\star, a_{t,h}^\star) \rangle \right)$$

$$= \underbrace{\sum_{h=1}^{H} \left( \overline{V}_{t,h+1}(s_{t,h+1}^\star) - \langle \widehat{\theta}_{t,h}, \phi(s_{t,h}^\star, a_{t,h}^\star) \rangle \right)}_{(i)} - \underbrace{\sum_{h=1}^{H} \langle \xi_{t,h}^{\text{P}}, \phi(s_{t,h}^\star, a_{t,h}^\star) \rangle}_{(ii)}$$

$$+ \underbrace{\sum_{h=1}^{H} \langle \omega_h^\star - \widehat{\omega}_{t,h}, \phi(s_{t,h}^\star, a_{t,h}^\star) \rangle}_{\text{(iii)}} - \underbrace{\sum_{h=1}^{H} \langle \xi_{t,h}^{\mathrm{R}}, \phi(s_{t,h}^\star, a_{t,h}^\star) \rangle}_{\text{(iv)}}.$$

Note that, given any state-action-state triple $(s, a, s')$, we have

$$\overline{V}_{t,h+1}(s') - \langle \widehat{\theta}_{t,h}, \phi(s,a) \rangle = \langle \mathcal{T}(\overline{\omega}_{t,h+1} + \overline{\theta}_{t,h+1}) - \widehat{\theta}_{t,h}, \phi(s,a) \rangle = \langle \eta_{t,h}, \phi(s,a) \rangle.$$

Plugging this back to (i), we obtain

$$\text{(i)} - \text{(ii)} \le \sum_{h=1}^{H} \langle \eta_{t,h} - \xi_{t,h}^{\mathrm{P}}, \phi(s_{t,h}^\star, a_{t,h}^\star) \rangle =: \sum_{h=1}^{H} \langle \eta_{t,h} - \xi_{t,h}^{\mathrm{P}}, \phi_h^\star \rangle$$

where we abbreviate $\phi_h^\star := \phi(s_{t,h}^\star, a_{t,h}^\star)$. Next, we split it into two parts:

$$\text{(i)} - \text{(ii)} \le \sum_{h=1}^{H} \langle \eta_{t,h}, P_{t,h} \phi_h^\star \rangle + \sum_{h=1}^{H} \langle \eta_{t,h}, (I - P_{t,h}) \phi_h^\star \rangle - \sum_{h=1}^{H} \langle \xi_{t,h}^{\mathrm{P}}, \phi_h^\star \rangle$$

$$\le \sum_{h=1}^{H} \|\eta_{t,h}\|_{\widehat{\Sigma}_{t,h}} \|P_{t,h} \phi_h^\star\|_{\widehat{\Sigma}_{t,h}^\dagger} + \sum_{h=1}^{H} \|\eta_{t,h}\|_{\Lambda_{t,h}} \|(I - P_{t,h}) \phi_h^\star\|_{\Lambda_{t,h}^{-1}} - \sum_{h=1}^{H} \langle \xi_{t,h}^{\mathrm{P}}, \phi_h^\star \rangle$$

$$\text{(Cauchy-Schwartz, Lemma 25)}$$

$$\le B_{\mathrm{err}}^{\mathrm{P}} \gamma H + \sum_{h=1}^{H} \|\eta_{t,h}\|_{\Lambda_{t,h}} \|(I - P_{t,h}) \phi_h^\star\|_{\Lambda_{t,h}^{-1}} - \sum_{h=1}^{H} \langle \xi_{t,h}^{\mathrm{P}}, \phi_h^\star \rangle$$

$$\text{(Assumption 4 and Lemmas 7 and 26)}$$

$$\le B_{\mathrm{err}}^{\mathrm{P}} \gamma H + \sqrt{H \sum_{h=1}^{H} \|\eta_{t,h}\|_{\Lambda_{t,h}}^2 \|(I - P_{t,h}) \phi_h^\star\|_{\Lambda_{t,h}^{-1}}^2} - \sum_{h=1}^{H} \langle \xi_{t,h}^{\mathrm{P}}, \phi_h^\star \rangle \quad \text{(Cauchy-Schwartz)}$$

Recall that $\xi_{t,h}^{\mathrm{P}}$ is sampled from $\mathcal{N}(0, \sigma_h^2 (I - P_{t,h}) \Lambda_{t,h}^{-1} (I - P_{t,h}))$. Therefore,

$$\sum_{h=1}^{H} \langle \xi_{t,h}^{\mathrm{P}}, \phi_h^\star \rangle \sim \mathcal{N}\left(0, \sum_{h=1}^{H} \sigma_h^2 \|(I - P_{t,h}) \phi_h^\star\|_{\Lambda_{t,h}^{-1}}^2\right).$$

Since $\sigma_h \ge \sqrt{H} \|\eta_{t,h}\|_{\Lambda_{t,h}}$ under high-probability event $\mathfrak{E}^{\mathrm{high}}$, we have

$$\Pr\left(\text{(i)} - \text{(ii)} \le B_{\mathrm{err}}^{\mathrm{P}} \gamma H\right) \ge \Gamma(-1).$$

Next, we consider $\text{(iii)} - \text{(iv)}$. By a similar argument, we have

$$\text{(iii)} - \text{(iv)} = \sum_{h=1}^{H} \langle \omega_h^\star - \widehat{\omega}_{t,h}, \phi_h^\star \rangle - \sum_{h=1}^{H} \langle \xi_{t,h}^{\mathrm{R}}, \phi_h^\star \rangle$$

$$\le \sum_{h=1}^{H} \|\omega_h^\star - \widehat{\omega}_{t,h}\|_{\Sigma_{t,h}} \|\phi_h^\star\|_{\Sigma_{t,h}^{-1}} - \sum_{h=1}^{H} \langle \xi_{t,h}^{\mathrm{R}}, \phi_h^\star \rangle$$

$$\le \sqrt{H \cdot \sum_{h=1}^{H} \|\omega_h^\star - \widehat{\omega}_{t,h}\|_{\Sigma_{t,h}}^2 \|\phi_h^\star\|_{\Sigma_{t,h}^{-1}}^2} - \sum_{h=1}^{H} \langle \xi_{t,h}^{\mathrm{R}}, \phi_h^\star \rangle.$$

Recall that $\xi_t^{\mathrm{R}}$ is sampled from $\mathcal{N}(0, \sigma_{\mathrm{R}}^2 \Sigma_{t,h}^{-1})$, and thus, we have

$$\sum_{h=1}^{H} \langle \xi_t^{\mathrm{R}}, \phi_h^\star \rangle \sim \mathcal{N}\left(0, \sum_{h=1}^{H} \sigma_{\mathrm{R}}^2 \|\phi_h^\star\|_{\Sigma_{t,h}^{-1}}^2\right).$$

Therefore, since $\sigma_{\mathrm{R}} \ge \sqrt{H} \|\omega_h^\star - \widehat{\omega}_{t,h}\|_{\Sigma_t}$ (Lemma 9), we have

$$\Pr\left(\text{(iii)} - \text{(iv)} \le 0\right) \ge \Gamma(-1).$$

Since the two events are independent, the probability that both events happen is at least $\Gamma^2(-1)$. $\quad\square$

**Lemma 15.** *The number of times $\mathfrak{E}_t^{\mathrm{span}}$ does not hold will not exceed $dH$, i.e.,*

$$\sum_{t=1}^{T} \mathbf{1}\left\{(\mathfrak{E}_t^{\mathrm{span}})^{\mathsf{C}}\right\} \le dH.$$

**Proof.** By definition, when $\mathfrak{E}_t^{\mathrm{span}}$ does not hold, there exists $h \in [H]$ such that $\phi(s_{t,h}, a_{t,h})$ is not in the span of $\{\phi(s_{i,h}, a_{i,h})\}_{i=1}^{t-1}$. That means, the dimension of the span should increase by exactly one after this iteration, i.e.,

$$\dim\left(\mathrm{span}\left(\{\phi(s_{i,h}, a_{i,h})\}_{i=1}^{t}\right)\right) = \dim\left(\mathrm{span}\left(\{\phi(s_{i,h}, a_{i,h})\}_{i=1}^{t-1}\right)\right) + 1.$$

However, the dimension cannot exceed $d$, so it can only increase at most $d$ times. This argument holds for any $h \in [H]$, and thus, the total number of times $\mathfrak{E}_t^{\mathrm{span}}$ does not happen will not exceed $dH$. $\qquad\square$

**Lemma 16.** *For any $h \in [H]$, it holds that*

$$\sum_{t=1}^{T} \|\phi(s_{t,h}, a_{t,h})\|_{\Sigma_{t,h}^{-1}} \le d\sqrt{2T\log(T+1)} =: B_\phi^{\mathrm{R}},$$

$$\sum_{t=1}^{T} \mathbf{1}\{\mathfrak{E}_t^{\mathrm{span}}\} \|\phi(s_{t,h}, a_{t,h})\|_{\widehat{\Sigma}_{t,h}^{\dagger}} \le \gamma d\sqrt{2dT\log\left(2T\gamma^2\right)} =: B_\phi^{\mathrm{P}}.$$

**Proof of Lemma 16.** We prove the two inequalities separately.

*Proof of the first inequality.* For any $t \in [T]$ and $h \in [H]$, we have the following bound on the norm of features (Lemma 5):

$$\|\phi(s_{t,h}, a_{t,h})\|_{\Sigma_{t,h}^{-1}} \le \|\phi(s_{t,h}, a_{t,h})\|_{\Lambda^{-1}} \le \sqrt{d}.$$

Hence, by Cauchy-Schwartz, we have

$$
\begin{aligned}
\sum_{t=1}^{T} \|\phi(s_{t,h}, a_{t,h})\|_{\Sigma_{t,h}^{-1}} &\le \sqrt{T \cdot \sum_{t=1}^{T} \|\phi(s_{t,h}, a_{t,h})\|_{\Sigma_{t,h}^{-1}}^2} \\
&= \sqrt{T \cdot \sum_{t=1}^{T} \min\left\{\|\phi(s_{t,h}, a_{t,h})\|_{\Sigma_{t,h}^{-1}}^2, d\right\}} \\
&\le \sqrt{Td \cdot \sum_{t=1}^{T} \min\left\{\|\phi(s_{t,h}, a_{t,h})\|_{\Sigma_{t,h}^{-1}}^2, 1\right\}} \\
&\le \sqrt{Td \cdot 2d\log(T+1)} \qquad \text{(elliptical potential lemma, Lemma 21)} \\
&= d\sqrt{2T\log(T+1)}.
\end{aligned}
$$

*Proof of the second inequality.* We divide the rounds into $d$ consecutive blocks, in each of which the rank of $\widehat{\Sigma}_{t,h}$ remains the same. To be specific, let $t_1, t_2, \ldots, t_d, t_{d+1}$ be a sequence of integers such that for any $i \in [d]$ and any $t \in \{t_i, t_{i+1}, \ldots, t_{i+1} - 1\}$, we have $\mathrm{rank}(\widehat{\Sigma}_{t,h}) = i$.

We will apply the elliptical potential lemma to each block separately. Now let's fix $i \in [d]$ and consider the $i$-th block. Let the reduced eigen-decomposition of $\widehat{\Sigma}_{t_i,h}$ be $\widehat{\Sigma}_{t_i,h} = UDU^{\top}$ where $U \in \mathbb{R}^{d \times i}$ and $D \in \mathbb{R}^{i \times i}$. For each $t \in \{t_i, t_{i+1}, \ldots, t_{i+1} - 1\}$, since $\phi(s_{t,h}, a_{t,h})$ is in the span of $\widehat{\Sigma}_{t,h}$ conditioning on $\mathfrak{E}_t^{\mathrm{span}}$, there exists a vector $x_t$ such that $\phi(s_{t,h}, a_{t,h}) = Ux_t$.

For any $t \in \{t_i, t_{i+1}, \ldots, t_{i+1} - 1\}$, we have

$$
\begin{aligned}
\|\phi(s_{t,h}, a_{t,h})\|_{\widehat{\Sigma}_{t,h}^{\dagger}}^2 &= \phi(s_{t,h}, a_{t,h})^{\top} \widehat{\Sigma}_{t,h}^{\dagger} \phi(s_{t,h}, a_{t,h}) \\
&= \phi(s_{t,h}, a_{t,h})^{\top} \left(\widehat{\Sigma}_{t_i,h} + \sum_{j=t_i}^{t-1} \phi(s_{j,h}, a_{j,h})\phi^{\top}(s_{j,h}, a_{j,h})\right)^{\dagger} \phi(s_{t,h}, a_{t,h})
\end{aligned}
$$

$$= x_t^\top U^\top \left( UDU^\top + \sum_{j=t_i}^{t-1} Ux_j x_j^\top U^\top \right)^\dagger Ux_t$$

$$= x_t^\top \left( D + \sum_{j=t_i}^{t-1} x_j x_j^\top \right)^{-1} x_t.$$

Define $D_t = D + \sum_{j=t_i}^{t-1} x_j x_j^\top$. Hence, we have

$$\sum_{t=t_i}^{t_{i+1}-1} \mathbf{1}\{\mathfrak{E}_t^{\mathrm{span}}\} \|\phi(s_{t,h}, a_{t,h})\|_{\widehat{\Sigma}_{t,h}^\dagger} = \sum_{t=t_i}^{t_{i+1}-1} \mathbf{1}\{\mathfrak{E}_t^{\mathrm{span}}\} \|x_t\|_{D_t^{-1}}.$$

By Assumption 4, the eigenvalues of $D$ are lower bounded by $1/\gamma^2$. And clearly, its eigenvalues are upper bounded by $t_i \le T$. Therefore, we have

$$\sum_{t=t_i}^{t_{i+1}-1} \mathbf{1}\{\mathfrak{E}_t^{\mathrm{span}}\} \|x_t\|_{D_t^{-1}} \le \sqrt{T \cdot \sum_{t=t_i}^{t_{i+1}-1} \mathbf{1}\{\mathfrak{E}_t^{\mathrm{span}}\} \|x_t\|_{D_t^{-1}}^2}$$

$$= \sqrt{T \cdot \sum_{t=t_i}^{t_{i+1}-1} \mathbf{1}\{\mathfrak{E}_t^{\mathrm{span}}\} \min\left\{ \|x_t\|_{D_t^{-1}}^2, \gamma^2 \right\}}$$

$$\le \gamma \sqrt{T \cdot \sum_{t=t_i}^{t_{i+1}-1} \mathbf{1}\{\mathfrak{E}_t^{\mathrm{span}}\} \min\left\{ \|x_t\|_{D_t^{-1}}^2, 1 \right\}}$$

$$\le \gamma\sqrt{T \cdot 2d\log\left(T\gamma^2(1+1/d)\right)} \quad \text{(elliptical potential lemma, Lemma 21)}$$

$$\le \gamma\sqrt{T \cdot 2d\log\left(2T\gamma^2\right)}.$$

This finishes the summation of one block. Notice that we have $d$ such blocks, we complete the proof by multiplying the above by $d$. $\qquad\square$

## C.4 MAIN STEPS OF THE PROOF

Let $\widetilde{V}_t(s_{t,1})$ denote an i.i.d. copy of $\overline{V}_t$ conditioned on initial state $s_{t,1}$ and $\widetilde{\mathfrak{E}}_t^{\mathrm{optm}}$ and $\widetilde{\mathfrak{E}}^{\mathrm{high}}$ denote the counterparts of $\mathfrak{E}_t^{\mathrm{optm}}$ and $\mathfrak{E}^{\mathrm{high}}$ but for $\widetilde{V}_t(s_{t,1})$.

The proof starts with the following decomposition of the regret:

$$\mathbb{E}\left[ \sum_{t=1}^T \left( V^\star(s_{t,1}) - V^{\pi_t}(s_{t,1}) \right) \right] \le \mathbb{E}\left[ \mathbf{1}\{\mathfrak{E}^{\mathrm{high}}\} \sum_{t=1}^T \mathbf{1}\{\mathfrak{E}_t^{\mathrm{span}}\} \left( V^\star(s_{t,1}) - V^{\pi_t}(s_{t,1}) \right) \right]$$

$$+ \mathbb{E}\left[ \mathbf{1}\{(\mathfrak{E}^{\mathrm{high}})^{\mathsf{C}}\} \sum_{t=1}^T \left( V^\star(s_{t,1}) - V^{\pi_t}(s_{t,1}) \right) \right]$$

$$+ \mathbb{E}\left[ \sum_{t=1}^T \mathbf{1}\{(\mathfrak{E}_t^{\mathrm{span}})^{\mathsf{C}}\} \left( V^\star(s_{t,1}) - V^{\pi_t}(s_{t,1}) \right) \right]$$

We will later show that the second and third terms can be easily bounded separately by observing the following two fact: (1) the probability that $\mathfrak{E}^{\mathrm{high}}$ doesn't hold is very small, and (2) the number of times $\mathfrak{E}_t^{\mathrm{span}}$ doesn't hold is also small. Hence, it remains to bound the first term, which is the most challenging. The most of the proof below is devoted to bounding it.

As the first step, we will add some necessary event conditions to the first term, using the following lemma.

**Lemma 17** (Adding necessary event conditions)**.** *It holds that*

$$\mathbb{E}\left[ \mathbf{1}\{\mathfrak{E}^{\mathrm{high}}\} \sum_{t=1}^T \mathbf{1}\{\mathfrak{E}_t^{\mathrm{span}}\} \left( V^\star(s_{t,1}) - V^{\pi_t}(s_{t,1}) \right) \right]$$

$$\le \frac{1}{\Gamma^2(-1)} \mathbb{E}\left[ \sum_{t=1}^T \mathbb{E}_{\widetilde{V}_t}\left[ \mathbf{1}\{\widetilde{\mathfrak{E}}^{\mathrm{high}} \cap \widetilde{\mathfrak{E}}_t^{\mathrm{span}} \cap \mathfrak{E}^{\mathrm{high}}\} \widetilde{V}_t(s_{t,1}) - \mathbf{1}\{\mathfrak{E}_t^{\mathrm{span}} \cap \mathfrak{E}^{\mathrm{high}} \cap \widetilde{\mathfrak{E}}^{\mathrm{high}} \cap \widetilde{\mathfrak{E}}_t^{\mathrm{span}}\} U_t(s_{t,1}) \right] \right]$$

$$+ \frac{1}{\Gamma^2(-1)} \cdot \left( dHB_V + B_{\mathrm{err}}^{\mathrm{P}} \gamma H + (B_{\mathrm{noise}}^{\mathrm{R}} + B_{\mathrm{err}}^{\mathrm{R}}) \cdot B_\phi^{\mathrm{R}} + dH^2 + 1 \right)$$

*where the expectation $\mathbb{E}_{\widetilde{V}_t}$ is taken over the randomness of $\widetilde{V}_t$ (an i.i.d. copy of $\overline{V}_t$) only.*

**Proof of Lemma 17.** We have

$$\mathbb{E}\left[ \mathbf{1}\{\mathfrak{E}^{\mathrm{high}}\} \sum_{t=1}^{T} \mathbf{1}\{\mathfrak{E}_t^{\mathrm{span}}\} \left( V^\star(s_{t,1}) - V^{\pi_t}(s_{t,1}) \right) \right]$$

$$\leq \mathbb{E}\left[ \mathbf{1}\{\mathfrak{E}^{\mathrm{high}}\} \sum_{t=1}^{T} \left( V^\star(s_{t,1}) - \mathbf{1}\{\mathfrak{E}_t^{\mathrm{span}}\} V^{\pi_t}(s_{t,1}) \right) \right] \qquad (V^\star \text{ is non-negative})$$

Plugging the condition on $\widetilde{\mathfrak{E}}_t^{\mathrm{optm}}$ (Lemma 14), we get

$$\leq \mathbb{E}\left[ \mathbf{1}\{\mathfrak{E}^{\mathrm{high}}\} \sum_{t=1}^{T} \mathbb{E}_{\widetilde{V}_t} \left[ \min\{H, \widetilde{V}_t(s_{t,1})\} - \mathbf{1}\{\mathfrak{E}_t^{\mathrm{span}}\} V^{\pi_t}(s_{t,1}) \,\big|\, \widetilde{\mathfrak{E}}_t^{\mathrm{optm}} \right] \right] + B_{\mathrm{err}}^{\mathrm{P}} \gamma H$$

We aim to add two event indicators, $\widetilde{\mathfrak{E}}^{\mathrm{high}}$ and $\widetilde{\mathfrak{E}}_t^{\mathrm{span}}$, and thus split the whole thing into several terms:

$$\leq \mathbb{E}\left[ \mathbf{1}\{\mathfrak{E}^{\mathrm{high}}\} \sum_{t=1}^{T} \mathbb{E}_{\widetilde{V}_t} \left[ \mathbf{1}\{\widetilde{\mathfrak{E}}^{\mathrm{high}} \cap \widetilde{\mathfrak{E}}_t^{\mathrm{span}}\} \left( \widetilde{V}_t(s_{t,1}) - \mathbf{1}\{\mathfrak{E}_t^{\mathrm{span}}\} V^{\pi_t}(s_{t,1}) \right) \,\big|\, \widetilde{\mathfrak{E}}_t^{\mathrm{optm}} \right] \right]$$

$$+ \mathbb{E}\left[ \mathbf{1}\{\mathfrak{E}^{\mathrm{high}}\} \sum_{t=1}^{T} \mathbb{E}_{\widetilde{V}_t} \left[ \mathbf{1}\{(\widetilde{\mathfrak{E}}^{\mathrm{high}})^{\mathsf{C}}\} \left( \min\{H, \widetilde{V}_t(s_{t,1})\} - \mathbf{1}\{\mathfrak{E}_t^{\mathrm{span}}\} V^{\pi_t}(s_{t,1}) \right) \,\big|\, \widetilde{\mathfrak{E}}_t^{\mathrm{optm}} \right] \right]$$

$$+ \mathbb{E}\left[ \mathbf{1}\{\mathfrak{E}^{\mathrm{high}}\} \sum_{t=1}^{T} \mathbb{E}_{\widetilde{V}_t} \left[ \mathbf{1}\{(\widetilde{\mathfrak{E}}_t^{\mathrm{span}})^{\mathsf{C}}\} \left( \min\{H, \widetilde{V}_t(s_{t,1})\} - \mathbf{1}\{\mathfrak{E}_t^{\mathrm{span}}\} V^{\pi_t}(s_{t,1}) \right) \,\big|\, \widetilde{\mathfrak{E}}_t^{\mathrm{optm}} \right] \right]$$

$$+ B_{\mathrm{err}}^{\mathrm{P}} \gamma H$$

$$=: \mathtt{T}_1 + \mathtt{T}_2 + \mathtt{T}_3 + B_{\mathrm{err}}^{\mathrm{P}} \gamma H.$$

Below we bound each term separately.

*Bounding* $\mathtt{T}_1$. To bound $\mathtt{T}_1$, we will first drop the conditioning event $\widetilde{\mathfrak{E}}_t^{\mathrm{optm}}$ to make things cleaner. To that end, we re-arange it in the following way

$$\mathtt{T}_1 = \mathbb{E}\left[ \mathbf{1}\{\mathfrak{E}^{\mathrm{high}}\} \sum_{t=1}^{T} \mathbb{E}_{\widetilde{V}_t} \left[ \underbrace{\mathbf{1}\{\widetilde{\mathfrak{E}}^{\mathrm{high}} \cap \widetilde{\mathfrak{E}}_t^{\mathrm{span}}\} \left( \widetilde{V}_t(s_{t,1}) - \mathbf{1}\{\mathfrak{E}_t^{\mathrm{span}}\} U_t(s_{t,1}) \right) + \mathbf{1}\{(\mathfrak{E}_t^{\mathrm{span}})^{\mathsf{C}}\} \cdot B_V}_{(*)} \,\Big|\, \widetilde{\mathfrak{E}}_t^{\mathrm{optm}} \right] \right]$$

$$+ \mathbb{E}\left[ \mathbf{1}\{\mathfrak{E}^{\mathrm{high}}\} \sum_{t=1}^{T} \mathbb{E}_{\widetilde{V}_t} \left[ \mathbf{1}\{\widetilde{\mathfrak{E}}^{\mathrm{high}} \cap \widetilde{\mathfrak{E}}_t^{\mathrm{span}}\} \mathbf{1}\{\mathfrak{E}_t^{\mathrm{span}}\} \left( U_t(s_{t,1}) - V^{\pi_t}(s_{t,1}) \right) \,\big|\, \widetilde{\mathfrak{E}}_t^{\mathrm{optm}} \right] \right]$$

$$- \mathbb{E}\left[ \mathbf{1}\{\mathfrak{E}^{\mathrm{high}}\} \sum_{t=1}^{T} \mathbb{E}_{\widetilde{V}_t} \left[ \mathbf{1}\{(\mathfrak{E}_t^{\mathrm{span}})^{\mathsf{C}}\} \cdot B_V \,\big|\, \widetilde{\mathfrak{E}}_t^{\mathrm{optm}} \right] \right]$$

$$=: \mathtt{T}_{1.1} + \mathtt{T}_{1.2} + \mathtt{T}_{1.3}.$$

The reason we did this is that we want to make sure $(*)$ is non-negative, so we can drop the conditioning event $\widetilde{\mathfrak{E}}_t^{\mathrm{optm}}$. To see why it is non-negative, we consider two cases: first, if $\mathfrak{E}_t^{\mathrm{span}}$ holds, then we already have $\mathbf{1}\{\widetilde{\mathfrak{E}}^{\mathrm{high}}\} \left( \widetilde{V}_t(s_{t,1}) - \mathbf{1}\{\mathfrak{E}_t^{\mathrm{span}}\} U_t(s_{t,1}) \right) \geq 0$ by definition of $U_t(s_{t,1})$; second, if $\mathfrak{E}_t^{\mathrm{span}}$ does not hold, then we have $\mathbf{1}\{\widetilde{\mathfrak{E}}^{\mathrm{high}} \cap \widetilde{\mathfrak{E}}_t^{\mathrm{span}}\} \widetilde{V}_t(s_{t,1}) + \mathbf{1}\{(\mathfrak{E}_t^{\mathrm{span}})^{\mathsf{C}}\} \cdot B_V \geq 0$ by Lemma 13.

Hence, for $\mathtt{T}_{1.1}$, we can drop the conditioning event using the following rule (for non-negative variable $X$):

$$\mathbb{E}[X \,|\, \mathfrak{E}] = \mathbb{E}[X \cdot \mathbf{1}\{\mathfrak{E}\}] / \Pr(\mathfrak{E}) \leq \mathbb{E}[X] / \Pr(\mathfrak{E})$$

and using Lemma 14 to get

$$\mathtt{T}_{1.1} \leq \frac{1}{\Gamma^2(-1)} \mathbb{E}\left[ \mathbf{1}\{\mathfrak{E}^{\mathrm{high}}\} \sum_{t=1}^{T} \mathbb{E}_{\widetilde{V}_t} \left[ \mathbf{1}\{\widetilde{\mathfrak{E}}^{\mathrm{high}} \cap \widetilde{\mathfrak{E}}_t^{\mathrm{span}}\} \left( \widetilde{V}_t(s_{t,1}) - \mathbf{1}\{\mathfrak{E}_t^{\mathrm{span}}\} U_t(s_{t,1}) \right) + \mathbf{1}\{(\mathfrak{E}_t^{\mathrm{span}})^{\mathsf{C}}\} \cdot B_V \right] \right]$$

$$
\begin{aligned}
&= \frac{1}{\Gamma^2(-1)} \, \mathbb{E}\left[ \mathbf{1}\{\mathfrak{E}^{\mathrm{high}}\} \sum_{t=1}^{T} \mathop{\mathbb{E}}_{\widetilde{V}_t} \left[ \mathbf{1}\{\widetilde{\mathfrak{E}}^{\mathrm{high}} \cap \widetilde{\mathfrak{E}}_t^{\mathrm{span}}\} \big( \widetilde{V}_t(s_{t,1}) - \mathbf{1}\{\mathfrak{E}_t^{\mathrm{span}}\} U_t(s_{t,1}) \big) \right] \right] \\
&\quad + \frac{1}{\Gamma^2(-1)} \, \mathbb{E}\left[ \mathbf{1}\{\mathfrak{E}^{\mathrm{high}}\} \sum_{t=1}^{T} \mathbf{1}\{(\mathfrak{E}_t^{\mathrm{span}})^{\mathsf{C}}\} \cdot B_V \right] \\
&\leq \frac{1}{\Gamma^2(-1)} \, \mathbb{E}\left[ \mathbf{1}\{\mathfrak{E}^{\mathrm{high}}\} \sum_{t=1}^{T} \mathop{\mathbb{E}}_{\widetilde{V}_t} \left[ \mathbf{1}\{\widetilde{\mathfrak{E}}^{\mathrm{high}} \cap \widetilde{\mathfrak{E}}_t^{\mathrm{span}}\} \big( \widetilde{V}_t(s_{t,1}) - \mathbf{1}\{\mathfrak{E}_t^{\mathrm{span}}\} U_t(s_{t,1}) \big) \right] \right] \\
&\quad + \frac{1}{\Gamma^2(-1)} \cdot dH B_V \qquad\qquad\qquad\qquad\qquad\qquad\qquad\qquad\text{(Lemma 15)}
\end{aligned}
$$

For $\mathtt{T}_{1.2}$, we apply Lemma 12 to get

$$
\begin{aligned}
\mathtt{T}_{1.2} &\leq \mathbb{E}\left[ \mathbf{1}\{\mathfrak{E}^{\mathrm{high}}\} \sum_{t=1}^{T} \mathop{\mathbb{E}}_{\widetilde{V}_t} \left[ \mathbf{1}\{\widetilde{\mathfrak{E}}^{\mathrm{high}} \cap \widetilde{\mathfrak{E}}_t^{\mathrm{span}}\} \mathbf{1}\{\mathfrak{E}_t^{\mathrm{span}}\} \big( \overline{V}_t(s_{t,1}) - V^{\pi_t}(s_{t,1}) \big) \,\big|\, \widetilde{\mathfrak{E}}_t^{\mathrm{optm}} \right] \right] \\
&\qquad\qquad\qquad\qquad\qquad\qquad\qquad\qquad (\overline{V}_t \geq U_t \text{ conditioning on } \mathfrak{E}^{\mathrm{high}}) \\
&\leq B_{\mathrm{err}}^{\mathrm{P}} \gamma H + (B_{\mathrm{noise}}^{\mathrm{R}} + B_{\mathrm{err}}^{\mathrm{R}}) \cdot B_\phi^{\mathrm{R}}.
\end{aligned}
$$

We simply upper bound $\mathtt{T}_{1.3}$ by zero. Plugging all these upper bounds back, we obtain

$$
\begin{aligned}
\mathtt{T}_1 &\leq \frac{1}{\Gamma^2(-1)} \, \mathbb{E}\left[ \mathbf{1}\{\mathfrak{E}^{\mathrm{high}}\} \sum_{t=1}^{T} \mathop{\mathbb{E}}_{\widetilde{V}_t} \left[ \mathbf{1}\{\widetilde{\mathfrak{E}}^{\mathrm{high}} \cap \widetilde{\mathfrak{E}}_t^{\mathrm{span}}\} \big( \widetilde{V}_t(s_{t,1}) - \mathbf{1}\{\mathfrak{E}_t^{\mathrm{span}}\} U_t(s_{t,1}) \big) \right] \right] \\
&\quad + \frac{1}{\Gamma^2(-1)} \cdot dH B_V + B_{\mathrm{err}}^{\mathrm{P}} \gamma H + (B_{\mathrm{noise}}^{\mathrm{R}} + B_{\mathrm{err}}^{\mathrm{R}}) \cdot B_\phi^{\mathrm{R}} \\
&= \frac{1}{\Gamma^2(-1)} \, \mathbb{E}\left[ \sum_{t=1}^{T} \mathop{\mathbb{E}}_{\widetilde{V}_t} \left[ \mathbf{1}\{\widetilde{\mathfrak{E}}^{\mathrm{high}} \cap \widetilde{\mathfrak{E}}_t^{\mathrm{span}} \cap \mathfrak{E}^{\mathrm{high}}\} \widetilde{V}_t(s_{t,1}) - \mathbf{1}\{\mathfrak{E}_t^{\mathrm{span}} \cap \mathfrak{E}^{\mathrm{high}} \cap \widetilde{\mathfrak{E}}^{\mathrm{high}} \cap \widetilde{\mathfrak{E}}_t^{\mathrm{span}}\} U_t(s_{t,1}) \right] \right] \\
&\quad + \frac{1}{\Gamma^2(-1)} \cdot dH B_V + B_{\mathrm{err}}^{\mathrm{P}} \gamma H + (B_{\mathrm{noise}}^{\mathrm{R}} + B_{\mathrm{err}}^{\mathrm{R}}) \cdot B_\phi^{\mathrm{R}}
\end{aligned}
$$

This is the final bound of $\mathtt{T}_1$ we need. Next, we go back to bound $\mathtt{T}_2$ and $\mathtt{T}_3$.

*Bounding* $\mathtt{T}_2$. We upper bound the value function inside the expectation by $H$ and obtain

$$
\begin{aligned}
\mathtt{T}_2 &\leq H \cdot \mathbb{E}\left[ \mathbf{1}\{\mathfrak{E}^{\mathrm{high}}\} \sum_{t=1}^{T} \mathop{\mathbb{E}}_{\widetilde{V}_t} \left[ \mathbf{1}\{(\widetilde{\mathfrak{E}}^{\mathrm{high}})^{\mathsf{C}}\} \,\big|\, \widetilde{\mathfrak{E}}_t^{\mathrm{optm}} \right] \right] \\
&\leq H \cdot \mathbb{E}\left[ \sum_{t=1}^{T} \mathop{\mathbb{E}}_{\widetilde{V}_t} \left[ \mathbf{1}\{(\widetilde{\mathfrak{E}}^{\mathrm{high}})^{\mathsf{C}}\} \,\big|\, \widetilde{\mathfrak{E}}_t^{\mathrm{optm}} \right] \right] \qquad\qquad \text{(dropping } \mathfrak{E}^{\mathrm{high}}) \\
&= H \cdot \mathbb{E}\left[ \sum_{t=1}^{T} \mathrm{Pr}\big( (\widetilde{\mathfrak{E}}^{\mathrm{high}})^{\mathsf{C}} \cap \widetilde{\mathfrak{E}}_t^{\mathrm{optm}} \big) \big/ \mathrm{Pr}\big( \widetilde{\mathfrak{E}}_t^{\mathrm{optm}} \big) \right] \\
&\leq \frac{HT}{\Gamma^2(-1)} \cdot \mathrm{Pr}\big( (\mathfrak{E}^{\mathrm{high}})^{\mathsf{C}} \big) \\
&\leq \frac{1}{\Gamma^2(-1)}. \qquad\qquad\qquad\qquad\qquad\qquad\qquad\qquad \text{(Lemma 8)}
\end{aligned}
$$

*Bounding* $\mathtt{T}_3$. Similar, we upper bound the value function inside the expectation by $H$ and obtain

$$
\begin{aligned}
\mathtt{T}_3 &\leq H \cdot \mathbb{E}\left[ \mathbf{1}\{\mathfrak{E}^{\mathrm{high}}\} \sum_{t=1}^{T} \mathop{\mathbb{E}}_{\widetilde{V}_t} \left[ \mathbf{1}\{(\widetilde{\mathfrak{E}}_t^{\mathrm{span}})^{\mathsf{C}}\} \,\big|\, \widetilde{\mathfrak{E}}_t^{\mathrm{optm}} \right] \right] \\
&\leq H \cdot \mathbb{E}\left[ \sum_{t=1}^{T} \mathop{\mathbb{E}}_{\widetilde{V}_t} \left[ \mathbf{1}\{(\widetilde{\mathfrak{E}}_t^{\mathrm{span}})^{\mathsf{C}}\} \,\big|\, \widetilde{\mathfrak{E}}_t^{\mathrm{optm}} \right] \right] \qquad\qquad \text{(dropping } \mathfrak{E}^{\mathrm{high}}) \\
&= H \cdot \mathbb{E}\left[ \sum_{t=1}^{T} \mathop{\mathbb{E}}_{\widetilde{V}_t} \left[ \mathbf{1}\{(\widetilde{\mathfrak{E}}_t^{\mathrm{span}})^{\mathsf{C}} \cap \widetilde{\mathfrak{E}}_t^{\mathrm{optm}}\} \right] \big/ \mathrm{Pr}(\widetilde{\mathfrak{E}}_t^{\mathrm{optm}}) \right] \\
&\leq H \cdot \mathbb{E}\left[ \sum_{t=1}^{T} \mathop{\mathbb{E}}_{\widetilde{V}_t} \left[ \mathbf{1}\{(\widetilde{\mathfrak{E}}_t^{\mathrm{span}})^{\mathsf{C}}\} \right] \big/ \mathrm{Pr}(\widetilde{\mathfrak{E}}_t^{\mathrm{optm}}) \right]
\end{aligned}
$$

$$\leq \frac{H}{\Gamma^2(-1)} \cdot \mathbb{E}\left[\sum_{t=1}^T \mathbf{1}\{(\mathfrak{E}_t^{\mathrm{span}})^{\complement}\}\right] \qquad \text{(tower rule)}$$

$$\leq \frac{dH^2}{\Gamma^2(-1)} \qquad \text{(Lemma 15)}$$

Plugging all these back, we conclude the proof. $\qquad\square$

The following lemma refines the event conditions established in Lemma 17 to make the whole thing more manageable.

**Lemma 18** (Refining event conditions). *It holds that*

$$\mathbb{E}\left[\sum_{t=1}^T \mathop{\mathbb{E}}_{\widetilde{V}_t}\left[\mathbf{1}\{\widetilde{\mathfrak{E}}^{\mathrm{high}} \cap \widetilde{\mathfrak{E}}_t^{\mathrm{span}} \cap \mathfrak{E}^{\mathrm{high}}\}\widetilde{V}_t(s_{t,1}) - \mathbf{1}\{\mathfrak{E}_t^{\mathrm{span}} \cap \mathfrak{E}^{\mathrm{high}} \cap \widetilde{\mathfrak{E}}^{\mathrm{high}} \cap \widetilde{\mathfrak{E}}_t^{\mathrm{span}}\}U_t(s_{t,1})\right]\right]$$

$$\leq \mathbb{E}\left[\sum_{t=1}^T \mathop{\mathbb{E}}_{\widetilde{V}_t}\left[\mathbf{1}\{\widetilde{\mathfrak{E}}^{\mathrm{high}} \cap \widetilde{\mathfrak{E}}_t^{\mathrm{span}}\}\widetilde{V}_t(s_{t,1}) - \mathbf{1}\{\mathfrak{E}_t^{\mathrm{span}} \cap \mathfrak{E}^{\mathrm{high}}\}U_t(s_{t,1})\right]\right]$$
$$+ dHB_V + 2B_V/H.$$

**Proof of Lemma 18.** We start with refining the event conditions on the first term. We remove unneeded events by splitting the first term into two parts:

$$\mathbb{E}\left[\sum_{t=1}^T \mathop{\mathbb{E}}_{\widetilde{V}_t}\left[\mathbf{1}\{\widetilde{\mathfrak{E}}^{\mathrm{high}} \cap \widetilde{\mathfrak{E}}_t^{\mathrm{span}} \cap \mathfrak{E}^{\mathrm{high}}\}\widetilde{V}_t(s_{t,1}) - \mathbf{1}\{\mathfrak{E}_t^{\mathrm{span}} \cap \mathfrak{E}^{\mathrm{high}} \cap \widetilde{\mathfrak{E}}^{\mathrm{high}} \cap \widetilde{\mathfrak{E}}_t^{\mathrm{span}}\}U_t(s_{t,1})\right]\right]$$

$$= \mathbb{E}\left[\sum_{t=1}^T \mathop{\mathbb{E}}_{\widetilde{V}_t}\left[\mathbf{1}\{\widetilde{\mathfrak{E}}^{\mathrm{high}} \cap \widetilde{\mathfrak{E}}_t^{\mathrm{span}}\}\widetilde{V}_t(s_{t,1}) - \mathbf{1}\{\mathfrak{E}_t^{\mathrm{span}} \cap \mathfrak{E}^{\mathrm{high}} \cap \widetilde{\mathfrak{E}}^{\mathrm{high}} \cap \widetilde{\mathfrak{E}}_t^{\mathrm{span}}\}U_t(s_{t,1})\right]\right]$$

$$- \mathbb{E}\left[\sum_{t=1}^T \mathop{\mathbb{E}}_{\widetilde{V}_t}\left[\mathbf{1}\{\widetilde{\mathfrak{E}}^{\mathrm{high}} \cap \widetilde{\mathfrak{E}}_t^{\mathrm{span}} \cap (\mathfrak{E}^{\mathrm{high}})^{\complement}\}\widetilde{V}_t(s_{t,1})\right]\right]$$

Here, using Lemma 13, the last term can be bounded by

$$- \mathbb{E}\left[\sum_{t=1}^T \mathop{\mathbb{E}}_{\widetilde{V}_t}\left[\mathbf{1}\{\widetilde{\mathfrak{E}}^{\mathrm{high}} \cap \widetilde{\mathfrak{E}}_t^{\mathrm{span}} \cap (\mathfrak{E}^{\mathrm{high}})^{\complement}\}\widetilde{V}_t(s_{t,1})\right]\right] \leq \mathbb{E}\left[\sum_{t=1}^T \mathbf{1}\{(\mathfrak{E}^{\mathrm{high}})^{\complement}\}B_V\right] \leq B_V/H$$

where we used Lemma 8 in the last inequality.

Now we seek to remove unneeded event conditions on $U_t$ as well. We notice the following decomposition

$$\mathbf{1}\{\mathfrak{E}_t^{\mathrm{span}} \cap \mathfrak{E}^{\mathrm{high}} \cap \widetilde{\mathfrak{E}}^{\mathrm{high}} \cap \widetilde{\mathfrak{E}}_t^{\mathrm{span}}\}U_t(s_{t,1})$$
$$\geq \mathbf{1}\{\mathfrak{E}_t^{\mathrm{span}} \cap \mathfrak{E}^{\mathrm{high}}\}U_t(s_{t,1})$$
$$- \mathbf{1}\{\mathfrak{E}_t^{\mathrm{span}} \cap \mathfrak{E}^{\mathrm{high}} \cap (\widetilde{\mathfrak{E}}^{\mathrm{high}})^{\complement}\}U_t(s_{t,1})$$
$$- \mathbf{1}\{\mathfrak{E}_t^{\mathrm{span}} \cap \mathfrak{E}^{\mathrm{high}} \cap (\widetilde{\mathfrak{E}}_t^{\mathrm{span}})^{\complement}\}U_t(s_{t,1}).$$

Plugging this back, we obtain

$$\mathbb{E}\left[\sum_{t=1}^T \mathop{\mathbb{E}}_{\widetilde{V}_t}\left[\mathbf{1}\{\widetilde{\mathfrak{E}}^{\mathrm{high}} \cap \widetilde{\mathfrak{E}}_t^{\mathrm{span}} \cap \mathfrak{E}^{\mathrm{high}}\}\widetilde{V}_t(s_{t,1}) - \mathbf{1}\{\mathfrak{E}_t^{\mathrm{span}} \cap \mathfrak{E}^{\mathrm{high}} \cap \widetilde{\mathfrak{E}}^{\mathrm{high}} \cap \widetilde{\mathfrak{E}}_t^{\mathrm{span}}\}U_t(s_{t,1})\right]\right]$$

$$\leq \mathbb{E}\left[\sum_{t=1}^T \mathop{\mathbb{E}}_{\widetilde{V}_t}\left[\mathbf{1}\{\widetilde{\mathfrak{E}}^{\mathrm{high}} \cap \widetilde{\mathfrak{E}}_t^{\mathrm{span}}\}\widetilde{V}_t(s_{t,1}) - \mathbf{1}\{\mathfrak{E}_t^{\mathrm{span}} \cap \mathfrak{E}^{\mathrm{high}}\}U_t(s_{t,1})\right]\right]$$

$$+ \mathbb{E}\left[\sum_{t=1}^T \mathop{\mathbb{E}}_{\widetilde{V}_t}\left[\mathbf{1}\{\mathfrak{E}_t^{\mathrm{span}} \cap \mathfrak{E}^{\mathrm{high}} \cap (\widetilde{\mathfrak{E}}^{\mathrm{high}})^{\complement}\}U_t(s_{t,1})\right]\right]$$

$$+ \mathbb{E}\left[\sum_{t=1}^T \mathop{\mathbb{E}}_{\widetilde{V}_t}\left[\mathbf{1}\{\mathfrak{E}_t^{\mathrm{span}} \cap \mathfrak{E}^{\mathrm{high}} \cap (\widetilde{\mathfrak{E}}_t^{\mathrm{span}})^{\complement}\}U_t(s_{t,1})\right]\right]$$

$$+ B_V/H$$

The first term is exactly what we want. Now we bound the middle two terms separately below:

$$\mathbb{E}\left[\sum_{t=1}^{T}\mathop{\mathbb{E}}_{\widetilde{V}_t}\left[\mathbf{1}\{\mathfrak{C}_t^{\mathrm{span}}\cap\mathfrak{C}^{\mathrm{high}}\cap(\widetilde{\mathfrak{C}}^{\mathrm{high}})^{\complement}\}U_t(s_{t,1})\right]\right]$$

$$\leq\mathbb{E}\left[\sum_{t=1}^{T}\mathop{\mathbb{E}}_{\widetilde{V}_t}\left[\mathbf{1}\{\mathfrak{C}_t^{\mathrm{span}}\cap\mathfrak{C}^{\mathrm{high}}\cap(\widetilde{\mathfrak{C}}^{\mathrm{high}})^{\complement}\}B_V\right]\right] \qquad \text{(Lemma 13)}$$

$$\leq T\cdot\Pr((\widetilde{\mathfrak{C}}^{\mathrm{high}})^{\complement})B_V$$

$$\leq B_V/H \qquad \text{(Lemma 8)}$$

and for the other term we also have

$$\mathbb{E}\left[\sum_{t=1}^{T}\mathop{\mathbb{E}}_{\widetilde{V}_t}\left[\mathbf{1}\{\mathfrak{C}_t^{\mathrm{span}}\cap\mathfrak{C}^{\mathrm{high}}\cap(\widetilde{\mathfrak{C}}_t^{\mathrm{span}})^{\complement}\}U_t(s_{t,1})\right]\right]$$

$$\leq\mathbb{E}\left[\sum_{t=1}^{T}\mathop{\mathbb{E}}_{\widetilde{V}_t}\left[\mathbf{1}\{(\widetilde{\mathfrak{C}}_t^{\mathrm{span}})^{\complement}\}\right]B_V\right] \qquad \text{(Lemma 13)}$$

$$=B_V\,\mathbb{E}\left[\sum_{t=1}^{T}\mathbf{1}\{(\mathfrak{C}_t^{\mathrm{span}})^{\complement}\}\right] \qquad \text{(tower rule)}$$

$$\leq dHB_V. \qquad \text{(Lemma 15)}$$

Hence, putting all together, we complete the proof. $\qquad\square$

The following lemma provides a final bound for the first term in Lemma 18.

**Lemma 19** (Final bound). *It holds that*

$$\mathbb{E}\left[\sum_{t=1}^{T}\mathop{\mathbb{E}}_{\widetilde{V}_t}\left[\mathbf{1}\{\widetilde{\mathfrak{C}}^{\mathrm{high}}\cap\widetilde{\mathfrak{C}}_t^{\mathrm{span}}\}\widetilde{V}_t(s_{t,1})-\mathbf{1}\{\mathfrak{C}_t^{\mathrm{span}}\cap\mathfrak{C}^{\mathrm{high}}\}U_t(s_{t,1})\right]\right]$$

$$\leq 2HB_{\mathrm{err}}^{\mathrm{P}}B_\phi^{\mathrm{P}}+2(B_{\mathrm{err}}^{\mathrm{R}}+B_{\mathrm{noise}}^{\mathrm{R}})\cdot HB_\phi^{\mathrm{R}}.$$

**Proof of Lemma 19.** By tower rule, we have

$$\mathbb{E}\left[\sum_{t=1}^{T}\mathop{\mathbb{E}}_{\widetilde{V}_t}\left[\mathbf{1}\{\widetilde{\mathfrak{C}}^{\mathrm{high}}\cap\widetilde{\mathfrak{C}}_t^{\mathrm{span}}\}\widetilde{V}_t(s_{t,1})-\mathbf{1}\{\mathfrak{C}_t^{\mathrm{span}}\cap\mathfrak{C}^{\mathrm{high}}\}U_t(s_{t,1})\right]\right]$$

$$=\mathbb{E}\left[\sum_{t=1}^{T}\mathbf{1}\{\mathfrak{C}^{\mathrm{high}}\cap\mathfrak{C}_t^{\mathrm{span}}\}\overline{V}_t(s_{t,1})-\mathbf{1}\{\mathfrak{C}_t^{\mathrm{span}}\cap\mathfrak{C}^{\mathrm{high}}\}U_t(s_{t,1})\right]$$

We plug in the result in Lemma 11 and get

$$\leq\mathbb{E}\left[\sum_{t=1}^{T}\mathbf{1}\{\mathfrak{C}^{\mathrm{high}}\cap\mathfrak{C}_t^{\mathrm{span}}\}\sum_{h=1}^{H}\left\langle\widehat{\theta}_{t,h}-\mathcal{T}(\overline{\theta}_{t,h+1}+\overline{\omega}_{t,h+1}),\phi(s_{t,h},a_{t,h})\right\rangle\right]$$

$$+\mathbb{E}\left[\sum_{t=1}^{T}\mathbf{1}\{\mathfrak{C}^{\mathrm{high}}\cap\mathfrak{C}_t^{\mathrm{span}}\}\sum_{h=1}^{H}\left\langle\mathcal{T}(\underline{\theta}_{t,h+1}+\underline{\omega}_{t,h+1})-\widehat{\theta}_{t,h},\phi(s_{t,h},a_{t,h})\right\rangle\right]$$

$$+\mathbb{E}\left[\sum_{t=1}^{T}\mathbf{1}\{\mathfrak{C}^{\mathrm{high}}\cap\mathfrak{C}_t^{\mathrm{span}}\}\sum_{h=1}^{H}\left\langle\overline{\omega}_{t,h}-\underline{\omega}_{t,h},\phi(s_{t,h},a_{t,h})\right\rangle\right]$$

Applying Cauchy-Schwartz inequality to each term yields

$$\leq\mathbb{E}\left[\sum_{t=1}^{T}\mathbf{1}\{\mathfrak{C}^{\mathrm{high}}\cap\mathfrak{C}_t^{\mathrm{span}}\}\sum_{h=1}^{H}\left\|\widehat{\theta}_{t,h}-\mathcal{T}(\overline{\theta}_{t,h+1}+\overline{\omega}_{t,h+1})\right\|_{\widehat{\Sigma}_{t,h}}\|\phi(s_{t,h},a_{t,h})\|_{\widehat{\Sigma}_{t,h}^{\dagger}}\right]$$

$$+\mathbb{E}\left[\sum_{t=1}^{T}\mathbf{1}\{\mathfrak{C}^{\mathrm{high}}\cap\mathfrak{C}_t^{\mathrm{span}}\}\sum_{h=1}^{H}\left\|\mathcal{T}(\underline{\theta}_{t,h+1}+\underline{\omega}_{t,h+1})-\widehat{\theta}_{t,h}\right\|_{\widehat{\Sigma}_{t,h}}\|\phi(s_{t,h},a_{t,h})\|_{\widehat{\Sigma}_{t,h}^{\dagger}}\right]$$

$$+\mathbb{E}\left[\sum_{t=1}^{T}\mathbf{1}\{\mathfrak{C}^{\mathrm{high}}\cap\mathfrak{C}_t^{\mathrm{span}}\}\sum_{h=1}^{H}\left(\|\overline{\omega}_{t,h}-\omega_h^\star\|_{\Sigma_{t,h}}+\|\omega_h^\star-\underline{\omega}_{t,h}\|_{\Sigma_{t,h}}\right)\|\phi(s_{t,h},a_{t,h})\|_{\Sigma_{t,h}^{-1}}\right]$$

The first two terms can be bounded by $HB_{\mathrm{err}}^{\mathrm{P}}B_{\phi}^{\mathrm{P}}$ using Lemmas 7 and 16. For the last term, conditioning on $\mathfrak{C}^{\mathrm{high}}$, we have

$$\|\overline{\omega}_{t,h} - \omega_h^{\star}\|_{\Sigma_{t,h}} \le \|\overline{\omega}_{t,h} - \widehat{\omega}_{t,h}\|_{\Sigma_{t,h}} + \|\widehat{\omega}_{t,h} - \omega_h^{\star}\|_{\Sigma_{t,h}} \le B_{\mathrm{err}}^{\mathrm{R}} + B_{\mathrm{noise}}^{\mathrm{R}}$$

and similarly for $\|\omega_h^{\star} - \underline{\omega}_{t,h}\|_{\Sigma_{t,h}}$. Also, applying Lemma 16, we have

$$\sum_{t=1}^{T}\sum_{h=1}^{H} \|\phi(s_{t,h}, a_{t,h})\|_{\Sigma_{t,h}^{-1}} \le HB_{\phi}^{\mathrm{R}}.$$

Inserting all these back, we get the upper bound of

$$2HB_{\mathrm{err}}^{\mathrm{P}}B_{\phi}^{\mathrm{P}} + 2(B_{\mathrm{err}}^{\mathrm{R}} + B_{\mathrm{noise}}^{\mathrm{R}}) \cdot HB_{\phi}^{\mathrm{R}}.$$

Hence, we complete the proof. $\qquad\square$

**Proof of Theorem 6.** We have

$$\mathbb{E}\left[\sum_{t=1}^{T}\left(V^{\star}(s_{t,1}) - V^{\pi_t}(s_{t,1})\right)\right] \le \mathbb{E}\left[\mathbf{1}\{\mathfrak{C}^{\mathrm{high}}\}\sum_{t=1}^{T}\mathbf{1}\{\mathfrak{C}_t^{\mathrm{span}}\}\left(V^{\star}(s_{t,1}) - V^{\pi_t}(s_{t,1})\right)\right]$$

$$+ \mathbb{E}\left[\mathbf{1}\{(\mathfrak{C}^{\mathrm{high}})^{\mathsf{C}}\}\sum_{t=1}^{T}\left(V^{\star}(s_{t,1}) - V^{\pi_t}(s_{t,1})\right)\right]$$

$$+ \mathbb{E}\left[\sum_{t=1}^{T}\mathbf{1}\{(\mathfrak{C}_t^{\mathrm{span}})^{\mathsf{C}}\}\left(V^{\star}(s_{t,1}) - V^{\pi_t}(s_{t,1})\right)\right]$$

$$=: \mathsf{T}_{\mathrm{A}} + \mathsf{T}_{\mathrm{B}} + \mathsf{T}_{\mathrm{C}}.$$

For $\mathsf{T}_{\mathrm{A}}$, by Lemmas 17 to 19 and re-arranging the results, we have

$$\mathsf{T}_{\mathrm{A}} \le \frac{1}{\Gamma^2(-1)} \cdot \left(2B_V(dH + 1/H) + HB_{\mathrm{err}}^{\mathrm{P}}\gamma + dH^2 + 1 + (B_{\mathrm{err}}^{\mathrm{R}} + B_{\mathrm{noise}}^{\mathrm{R}})(2H+1)B_{\phi}^{\mathrm{R}} + 2HB_{\mathrm{err}}^{\mathrm{P}}B_{\phi}^{\mathrm{P}}\right)$$

$$= \widetilde{O}\left(d^{5/2}H^{5/2} + d^2H^{3/2}\sqrt{T} + \varepsilon_1\gamma\left(dH^2 + d^{3/2}H\sqrt{T}\right)\right.$$

$$\left. + \sqrt{\varepsilon_{\mathrm{B}}}\left(d^2H^{5/2}\sqrt{T} + d^{3/2}H^{3/2}T\right) + \varepsilon_{\mathrm{B}}\gamma\left(dH^2\sqrt{T} + d^{3/2}HT\right)\right)$$

For $\mathsf{T}_{\mathrm{B}}$, by Lemma 8, we have

$$\mathsf{T}_{\mathrm{B}} \le HT \cdot \mathrm{Pr}\left((\mathfrak{C}^{\mathrm{high}})^{\mathsf{C}}\right) \le 1.$$

For $\mathsf{T}_{\mathrm{C}}$, by Lemma 15, we have

$$\mathsf{T}_{\mathrm{C}} \le H \cdot \mathbb{E}\left[\sum_{t=1}^{T}\mathbf{1}\{(\mathfrak{C}_t^{\mathrm{span}})^{\mathsf{C}}\}\right] \le dH^2.$$

Putting everything together, we complete the proof. $\qquad\square$

## D    SUPPORTING LEMMAS

**Lemma 20** (Gaussian concentration). *(Abeille & Lazaric, 2017) Let $x \sim \mathcal{N}\left(0, c\Sigma^{-1}\right)$ for $c \in \mathbb{R}^+$ and $\Sigma$ a positive definite matrix. Then, for any $\delta > 0$, we have $\mathrm{Pr}\left(\|x\|_{\Sigma} > \sqrt{2cd\log(2d/\delta)}\right) \le \delta$*

**Lemma 21** (Elliptical potential lemma). *Assume that $X \subseteq \{x : \|x\|_2 \le 1\}$ is compact and $\mathrm{span}(X) = \mathbb{R}^d$. Let $x_1, \ldots, x_T \in X$ be a sequence of vectors, $\Sigma_1$ be a positive definite matrix with each eigenvalue bounded within the range of $[a, b]$ for some $a, b > 0$, and $\Sigma_{t+1} = \Sigma_t + x_t x_t^{\top}$. Then, we have*

$$\sum_{t=1}^{T}\min\left\{1, x_t^{\top}\Sigma_t^{-1}x_t\right\} \le 2d\log\left(\frac{b}{a} + \frac{T}{ad}\right).$$

*Furthermore, if $\Sigma_1$ is constructed via optimal design, i.e., $\Sigma_1 = \mathbb{E}_{x \sim \rho} xx^\top$ where $\rho \in \Delta(X)$ is an optimal design over $X$, then we have*

$$\sum_{t=1}^{T} \min\left\{1, x_t^\top \Sigma_t^{-1} x_t\right\} \le 2d \log(T+1).$$

**Proof of Lemma 21.** First we claim that

$$\min\left\{1, x_t^\top \Sigma_t^{-1} x_t\right\} \le 2 x_t^\top \Sigma_{t+1}^{-1} x_t \tag{14}$$

To show this, we use Sherman-Morrison-Woodbury formula (Bhatia, 2013) for rank-one updates to a matrix inverse:

$$x_t^\top \Sigma_{t+1}^{-1} x_t = x_t^\top \left(\Sigma_t + x_t x_t^\top\right)^{-1} x_t = x_t^\top \left(\Sigma_t^{-1} - \frac{\Sigma_t^{-1} x_t x_t^\top \Sigma_t^{-1}}{1 + \|x_t\|_{\Sigma_t^{-1}}^2}\right) x_t = \|x_t\|_{\Sigma_t^{-1}}^2 - \frac{\|x_t\|_{\Sigma_t^{-1}}^4}{1 + \|x_t\|_{\Sigma_t^{-1}}^2} = \frac{\|x_t\|_{\Sigma_t^{-1}}^2}{1 + \|x_t\|_{\Sigma_t^{-1}}^2}.$$

Now let us consider two cases for the right-hand side of the above:

Case 1 : $x_t^\top \Sigma_t^{-1} x_t \le 1$. Then, we can lower bound the right-hand side above by $\|x_t\|_{\Sigma_t^{-1}}^2 / 2$.

Case 2 : $x_t^\top \Sigma_t^{-1} x_t \ge 1$. Then the right-hand side above is directly at least $1/2$ since the function $x/(1+x)$ is increasing in $x$.

Hence, in both cases, we have $x_t^\top \Sigma_{t+1}^{-1} x_t \ge \min\left\{1, x_t^\top \Sigma_t^{-1} x_t\right\} / 2$, which finishes the proof of (14).

Since the log-determinant function is concave, we can obtain that $\log \det(\Sigma_t) - \log \det \Sigma_{t+1} \le \operatorname{tr}\left(\Sigma_{t+1}^{-1}(\Sigma_t - \Sigma_{t+1})\right)$ via first-order Taylor approximation. This gives us the following

$$\sum_{t=1}^{T} x_t^\top \Sigma_{t+1}^{-1} x_t = \sum_{t=1}^{T} \operatorname{tr}\left(\Sigma_{t+1}^{-1}(\Sigma_{t+1} - \Sigma_t)\right) \le \sum_{t=1}^{T} \left(\log \det \Sigma_{t+1} - \log \det \Sigma_t\right) = \log\left(\frac{\det \Sigma_{T+1}}{\det \Sigma_1}\right)$$

where the last step follows from telescoping. Since each eigenvalue of $\Sigma_1$ is lower bounded by $a$, we have $\det \Sigma_1 \ge a^d$. Towards an upper bound of $\det \Sigma_{T+1} = \det(\Sigma_1 + \sum_{t=1}^{T} x_t x_t^\top)$, let $(\lambda_1, \ldots, \lambda_d)$ denote the eigenvalues of $\sum_{t=1}^{T} x_t x_t^\top$, and then we have

$$\det\left(\Sigma_1 + \sum_{t=1}^{T} x_t x_t^\top\right) \le \prod_{i=1}^{d}(b + \lambda_i) \le \left(\frac{1}{d}\sum_{i=1}^{d}(b + \lambda_i)\right)^d \le \left(b + \frac{1}{d}\operatorname{tr}\left(\sum_{t=1}^{T} x_t x_t^\top\right)\right)^d \le \left(b + \frac{T}{d}\right)^d$$

Here, the first step is Weyl's inequality, the second step is AM-GM inequality, and the last step is because the trace is bounded by $T$. Plugging this upper bound back, we have

$$\log\left(\frac{\det \Sigma_{T+1}}{\det \Sigma_1}\right) \le d \log\left(\frac{b}{a} + \frac{T}{ad}\right).$$

This completes the proof of the first statement.

For the case where $\Sigma_1$ is constructed via optimal design, we can rewrite $\Sigma_{T+1}$ in the following way:

$$\Sigma_{T+1} = \mathbb{E}_{x \sim \rho} xx^\top + \sum_{t=1}^{T} x_t x_t^\top = (T+1)\underbrace{\left(\frac{1}{1+T} \cdot \mathbb{E}_{x \sim \rho} xx^\top + \sum_{t=1}^{T} \frac{1}{1+T} \cdot x_t x_t^\top\right)}_{(*)} =: (T+1)\mathbb{E}_{x \sim \rho'} xx^\top$$

where we consider $(*)$ as an expectation of $xx^\top$ over a new distribution that we denote by $\rho'$. Recall that $\Sigma_1$ is constructed via optimal design, which implies $\det \Sigma_1 \ge \det \mathbb{E}_{x \sim \rho'} xx^\top$ (Lemma 23). This gives us

$$\log\left(\frac{\det \Sigma_{T+1}}{\det \Sigma_1}\right) = \log\left(\frac{(T+1)^d \det \mathbb{E}_{x \sim \rho'} xx^\top}{\det \Sigma_1}\right) \le \log\left((T+1)^d\right) = d \log(T+1).$$

This completes the proof. $\qquad\square$

The following inequality is well-known; we use the version stated in Zhu & Nowak (2022).

**Lemma 22** (Freedman's inequality). *Let $\{X_t\}_{t \leq T}$ be a real-valued martingale different sequence adapted to the filtration $\mathcal{F}_t$, and let $\mathbb{E}_t[\cdot] \coloneqq \mathbb{E}[\cdot \mid \mathcal{F}_{t-1}]$. If $|X_t| \leq B$ almost surely, then for any $\eta \in (0, 1/B)$, the following holds with probability at least $1 - \delta$:*

$$\sum_{t=1}^{T} X_t \leq \eta \sum_{t=1}^{T} \mathbb{E}_t[X_t^2] + \frac{B \log(1/\delta)}{\eta}.$$

**Lemma 23.** *(Lattimore & Szepesvári, 2020) Assume that $\Phi \subseteq \mathbb{R}^d$ is compact and $\mathrm{span}(\Phi) = \mathbb{R}^d$. For a distribution $\rho$ over $\Phi$, define $\Lambda(\rho) = \sum_{\phi \in \Phi} \rho(\phi) \phi \phi^\top$ and $g(\rho) = \max_{\phi \in \Phi} \|\phi\|_{\Lambda(\rho)^{-1}}^2$. Then, the following are equivalent:*

- *$\rho$ is a minimizer of $g$.*

- *$\rho$ is a maximizer of $f(\rho) \coloneqq \log \det \Lambda(\rho)$.*

- *$g(\rho) = d$.*

*Furthermore, there exists a minimizer $\rho$ of $g$ such that $|\mathrm{supp}(\rho)| \leq d(d+1)/2$.*

Below we show that the Cauchy-Schwarz inequality is still valid when the matrix is not invertible under some conditions. We start with the following lemma.

**Lemma 24.** *Let $A$ be a positive semi-definite matrix. Let $B$ be a square root of $A$, i.e., $A = BB^\top$. Then $\mathrm{range}(A) = \mathrm{range}(B)$.*

**Proof of Lemma 24.** We first show that $\mathrm{range}(A) \subseteq \mathrm{range}(B)$. To see this, for any $y \in \mathrm{range}(A)$, there exists $x$ such that $y = Ax = BB^\top x = B(B^\top x)$. Hence $y \in \mathrm{range}(B)$. Next, we show that $\mathrm{range}(B) \subseteq \mathrm{range}(A)$. To see this, for any $y \in \mathrm{range}(B)$, there exists $x$ such that $y = Bx$. Let $x = x_0 + x_1$ where $x_0 \in \mathrm{null}(B)$ and $x_1 \in \mathrm{rowspace}(B)$. Then, $y = Bx = Bx_1$. Since $x_1 \in \mathrm{rowspace}(B)$, there exists $z$ such that $x_1 = B^\top z$. Thus, $y = Bx_1 = BB^\top z = Az$. Hence, $y \in \mathrm{range}(A)$. $\qquad\square$

**Lemma 25** (Cauchy-Schwarz under pseudo-inverse). *Let $\Sigma$ be a positive semi-definite matrix (that is unnecessarily invertible). Then, for any $x \in \mathrm{range}(\Sigma)$ and any $y \in \mathbb{R}^d$, we have*

$$x^\top y \leq \|x\|_{\Sigma^\dagger} \|y\|_\Sigma.$$

**Proof of Lemma 25.** Let $B$ denote the square root of $\Sigma$ and force $B$ to be positive semi-definite. One can verify that $BB^\dagger$ is the orthogonal projection matrix onto $\mathrm{range}(B)$, and hence, $\mathrm{range}(\Sigma)$ (recalling that $\mathrm{range}(B) = \mathrm{range}(\Sigma)$ by Lemma 24). Therefore, for any $x \in \mathrm{range}(\Sigma)$, we have $BB^\dagger x = x$. Then, we have

$$x^\top y = x^\top B^\dagger By \leq \sqrt{x^\top B^\dagger B^{\dagger\top} x} \sqrt{y^\top BBy} = \|x\|_{\Sigma^\dagger} \|y\|_\Sigma$$

where the inequality follows from the standard Cauchy-Schwarz inequality. $\qquad\square$

**Lemma 26** (Invariance under projection). *Let $\Sigma \in \mathbb{R}^{d \times d}$ be a positive semi-definite matrix of rank $r$. For any vector $\phi \in \mathbb{R}^d$, we have $\|\phi\|_{\Sigma^\dagger} = \|P\phi\|_{\Sigma^\dagger}$ where $P$ is the projection onto the range of $\Sigma$.*

**Proof of Lemma 26.** Assume the eigen-decomposition of $\Sigma = U\Lambda U^\top$, so $\Sigma^\dagger = U\Lambda^\dagger U^\top$. Without loss of generality, we assume $\Lambda$ has all its non-zero elements at the front and zero elements at the back on the diagonal. Denote $U_r$ as the matrix obtained by replacing the last $n - r$ columns of $U$ by 0. Note that the first $r$ columns of $U$ is in the range of $\Sigma$, so we must have $PU = U_r$. Then, we have the following

$$\|P\phi\|_{\Sigma^\dagger}^2 = \phi^\top P^\top \Sigma^\dagger P\phi = \phi^\top P^\top U\Lambda^\dagger U^\top P\phi = \phi^\top P^\top U_r \Lambda^\dagger U_r^\top P\phi = \phi^\top U_r \Lambda^\dagger U_r^\top \phi = \phi^\top U\Lambda^\dagger U^\top \phi = \|\phi\|_{\Sigma^\dagger}^2.$$

This completes the proof. $\qquad\square$

### D.1 PSEUDO DIMENSION AND COVERING NUMBER

**Definition 6** ($\ell_1$-Covering number). *(Definition 4 of Modi et al. (2024)) Given a hypothesis class $\mathcal{H} \subseteq (\mathcal{Z} \mapsto \mathbb{R})$ and $Z^n = (z_1, \ldots, z_n) \in \mathcal{Z}^n$, $\varepsilon > 0$, define $\mathcal{N}(\varepsilon, \mathcal{H}, Z^n)$ as the minimum cardinality of a set $\mathcal{C} \subseteq \mathcal{H}$, such that for any $h \in \mathcal{H}$, there exists $h' \in \mathcal{C}$ such that $\sum_{i=1}^n |h(z_i) - h'(z_i)|/n \leq \varepsilon$. We define $\mathcal{N}(\varepsilon, \mathcal{H}, n) = \max_{Z^n \in \mathcal{Z}^n} \mathcal{N}(\varepsilon, \mathcal{H}, Z^n)$.*

Below we define the pseudo-dimension (Haussler, 2018; Modi et al., 2024).

**Definition 7** (VC-dimension). *For hypothesis class $\mathcal{H} \subseteq (\mathcal{X} \to \{0,1\})$, we define its VC-dimension $\mathrm{VC\text{-}dim}(\mathcal{H})$ as the maximal cardinality of a set $X = \{x_1, \ldots, x_{|X|}\} \subseteq \mathcal{X}$ that satisfies $|\mathcal{H}_X| = 2^{|X|}$ (or $X$ is shattered by $\mathcal{H}$), where $\mathcal{H}_X$ is the restriction of $\mathcal{H}$ to $X$, i.e., $\{(h(x_1), \ldots, h(x_{|X|})) : h \in \mathcal{H}\}$.*

**Definition 8** (Pseudo-dimension). *For hypothesis class $\mathcal{H} \subseteq (\mathcal{X} \to \mathbb{R})$, we define its pseudo dimension $\mathrm{Pdim}(\mathcal{H})$ as $\mathrm{Pdim}(\mathcal{H}) = \mathrm{VCdim}(\mathcal{H}^+)$, where $\mathcal{H}^+ = \{(x, \xi) \mapsto \mathbf{1}[h(x) > \xi] : h \in \mathcal{H}\} \subseteq (\mathcal{X} \times \mathbb{R} \to \{0,1\})$*

The following lemma provides a bound on the covering number of a hypothesis class via pseudo dimension.

**Lemma 27.** *(Corollary 42 of Modi et al. (2024)) Given a hypothesis class $\mathcal{H} \subseteq \mathcal{Z} \mapsto [a,b]$ with $\mathrm{Pdim}(\mathcal{H}) \leq d$, then, for any $n$, we have*

$$\mathcal{N}(\varepsilon, \mathcal{H}, n) \leq \left(4e^2(b-a)/\varepsilon\right)^d.$$

*Note that the right-hand side is independent of $n$.*

## E  LINEAR MDPs AND LQRs IMPLY LINEAR BELLMAN COMPLETENESS

It is already well known that linear Bellman completeness captures linear MDPs, as demonstrated in works such as Agarwal et al. (2019); Zanette et al. (2020b). Here, we show that it also captures LQRs for a convex subset of linear functions (specifically, when the value function is parameterized by a PSD matrix). We start with the definition.

**Definition 9** (Linear Quadratic Regulator). *A linear quadratic regulator (LQR) problem is defined by a tuple $(A, B, Q, R)$ where $A \in \mathbb{R}^{d \times d}$, $B \in \mathbb{R}^{d \times m}$, $Q \in \mathbb{R}^{d \times d}$, and $R \in \mathbb{R}^{m \times m}$. The objective is to find a policy $\pi$ that minimizes the following:*

$$J(\pi) = \mathbb{E}\left[\sum_{h=1}^H x_h^\top Q x_h + u_h^\top R u_h\right]$$

*where $x_{h+1} = A x_h + B u_h + w_h$ where $w_h \sim \mathcal{N}(0, \Sigma)$.*

Let us focus on an arbitrary step $h$ and simply write the transition as the following (ignoring the subscript $h$ for notational simplicity):

$$x' = Ax + Bu + w, \quad \text{where} \quad w \sim \mathcal{N}(0, \Sigma). \tag{15}$$

We consider state-action value functions of the form:

$$Q(x, u) = \begin{bmatrix} x \\ u \end{bmatrix}^\top \begin{bmatrix} P_{xx} & P_{xu} \\ P_{ux} & P_{uu} \end{bmatrix} \begin{bmatrix} x \\ u \end{bmatrix} + c. \tag{16}$$

It is linear in the quadratic feature $\phi(x, u) = [x^2, u^2, xu, x, u, 1]$. Without loss of generality, we assume $P_{xu} = P_{ux}^\top$. Note that we may not have the Bellman completeness for *any* such $Q$. However, it does hold under the restriction that $P = \begin{bmatrix} P_{xx} & P_{xu} \\ P_{ux} & P_{uu} \end{bmatrix}$ is PSD. Recall that $P$ is PSD if and only if (1) $P_{uu}$ is PSD, and (2) its Schur complement $P_{xx} - P_{xu} P_{uu}^{-1} P_{ux}$ is PSD. We note that such a set of feasible $P$ is a convex set.

Then, we consider the Bellman backup of $Q$ (ignoring the per-step reward (cost) for now):

$$\widetilde{Q}(x, u) = \mathbb{E}_{x'}\left[\min_{u'} Q(x', u')\right] \tag{17}$$

$$= \mathbb{E}_w \left[ \min_{u'} \begin{bmatrix} Ax + Bu + w \\ u' \end{bmatrix}^{\top} \begin{bmatrix} P_{xx} & P_{xu} \\ P_{ux} & P_{uu} \end{bmatrix} \begin{bmatrix} Ax + Bu + w \\ u' \end{bmatrix} + c \right] \tag{18}$$

$$= \mathbb{E}_w \left[ \min_{u'} \left\{ [Ax + Bu + w]^T P_{xx} [Ax + Bu + w] + 2 [Ax + Bu + w]^T P_{xu} u' + u'^T P_{uu} u' \right\} \right] + c. \tag{19}$$

Using first-order condition, we know that the optimal $u'$ (for a fixed $w$) satisfies

$$u' = -P_{uu}^{-1} P_{ux} (Ax + Bu + w), \tag{20}$$

which implies that, the term in (17) is equal to

$$\min_{u'} Q(x', u') = [Ax + Bu + w]^T \left[ P_{xx} - P_{xu} P_{uu}^{-1} P_{ux} \right] [Ax + Bu + w] + c \tag{21}$$

Plugging the above in (19), we get

$$\widetilde{Q}(x, u) = \mathbb{E}_w \left[ [Ax + Bu + w]^T \left[ P_{xx} - P_{xu} P_{uu}^{-1} P_{ux} \right] [Ax + Bu + w] + c \right] \tag{22}$$

$$= [Ax + Bu]^T \left[ P_{xx} - P_{xu} P_{uu}^{-1} P_{ux} \right] [Ax + Bu] + c + \mathbf{Tr} \left( \left( P_{xx} - P_{xu} P_{uu}^{-1} P_{ux} \right) \Sigma \right) \tag{23}$$

$$= [Ax + Bu]^T \left[ P_{xx} - P_{xu} P_{uu}^{-1} P_{xu}^{\top} \right] [Ax + Bu] + c' \tag{24}$$

$$= \begin{bmatrix} x \\ u \end{bmatrix}^T \begin{bmatrix} A^T \\ B^T \end{bmatrix} \left( P_{xx} - P_{xu} P_{uu}^{-1} P_{xu}^{\top} \right) \begin{bmatrix} A & B \end{bmatrix} \begin{bmatrix} x \\ u \end{bmatrix} + c' \tag{25}$$

where $c'$ is some constant. The middle matrix above is PSD if $P_{xx} \succeq P_{xu} P_{uu}^{-1} P_{xu}^{\top}$, which holds since $P$ is PSD. Thus, we conclude that $\widetilde{Q}$ is also linear for some PSD matrix.

We can also easily verify that the reward (cost) function is linear in the quadratic feature. Hence, we complete the proof.

## F  COMPUTATIONALLY EFFICIENT IMPLEMENTATIONS FOR OPTIMIZATION ORACLES

The convex programming algorithm given in Algorithm 2 is due to Bertsimas & Vempala (2004). In the following, we provide an informal description of Algorithm 2 below but refer the reader to Bertsimas & Vempala (2004) for the full details.

At an iteration $t \le T$, Algorithm 2 stars with a set $\mathcal{D}_t$ which contains the set $\mathcal{K}$, and a set of $2N$ points $\mathcal{U}_t$ sampled (approximately) uniformly from $\mathcal{D}_t$ using the SAMPLER subroutine in Algorithm 3. It then uses the first $N$ samples from $\mathcal{U}_t$ to compute an approximate centroid $z_t$ of the set $\mathcal{D}_t$ in Line 23; the remaining points from $\mathcal{U}_t$ are denoted by $\mathcal{V}_t$. It then queries the separation oracle $\mathcal{O}_{\mathcal{K}}^{\text{sep}}$ at the point $z_t$. If $z_t \in \mathcal{K}$, then we terminate and return $z_t$. Otherwise, we use the separating hyperplane between $z_t$ and $\mathcal{K}$ to shrink the set $\mathcal{D}_t$ further into $\mathcal{D}_{t+1}$ in Line 29. Finally, it calls SAMPLER again using the set of points $\mathcal{V}_t$ as a warm start to get $2N$ new (approximately) i.i.d. sample from $\mathcal{D}_{t+1}$ in Line 30. Equipped with the sets $\mathcal{D}_{t+1}$ and $\mathcal{U}_{t+1}$, another iteration of the algorithm follows.

On receiving a convex set $\mathcal{D}$ and a set of points $\mathcal{V}$, the SAMPLER protocol in Algorithm 3 first refines $\mathcal{V}$ to $\mathcal{V}'$ by disposing off any points $z \in \mathcal{V}$ that do not lie in $\mathcal{D}$. Then, it starts a random ball walk from the samples in $\mathcal{V}'$: in order to update the current point $\widehat{z}$ we first sample a point $z'$ uniformly from the ellipsoid $\widehat{z} + \eta \Lambda^{1/2} \mathbb{B}_d(1)$ (where $\Lambda$ is defined using the points in $\mathcal{V}'$) and then updates $\widehat{z} \leftarrow z'$ if $z' \in \mathcal{D}$. The analysis of Bertsimas & Vempala (2004) shows that this ball walk mixes fast to a uniform distribution over the set $\mathcal{D}$.

## G  MISSING DETAILS FROM SECTION 6.2

### G.1  COMPUTATIONALLY EFFICIENT ESTIMATION OF REWARD FUNCTION (EQN. 2)

The convex set feasibility procedure of Bertsimas & Vempala (2004) can also be used to estimate the parameters for the reward functions in Equation (2) in Algorithm 1. Note that for any time $t$ and

---

**Algorithm 2** Solving Convex Programs by Random Walks (Bertsimas & Vempala (2004))

---

**Require:** • Separation oracle $\mathcal{O}_{\mathcal{K}}^{\mathrm{sep}}$ for the convex set $\mathcal{K} \subseteq \mathbb{R}^d$.
      • Parameters $r, R, \delta$.

1: Let $T = 2d \log(R/\delta r)$ and $N = O(d \log(1/\delta))$
2: Let $\mathcal{D}_1$ be the axis-aligned cube with width $R$ with center $z_1 = 0$.
3: Sample $2N$ points $\mathcal{U}_1 := \{z_1^1, \ldots, z_1^{2N}\} \leftarrow \mathrm{Uniform}(\mathcal{D}_1)$.
4: Let $\mathcal{V}_1 := \{z_1^1, \ldots, z_1^N\}$ and $\bar{\mathcal{V}}_1 := \mathcal{U}_1 \smallsetminus \mathcal{V}_1$.
5: **for** $t = 1, \ldots, T$ **do**
6:  Compute the point $z_t \leftarrow \frac{1}{N} \sum_{z \in \mathcal{V}_t} z$.
7:  **if** $z_t \in K$ **then**
8:   **Return** $z_t$ and **terminate**.
9:  **else**
10:   // If $z_t \notin \mathcal{K}$, shrink the set $\mathcal{D}_t$ using a separating hyperplane //
11:   Let $\langle a_t, z \rangle \leq b$ be the separating hyperplane returned by $\mathcal{O}_{\mathcal{K}}^{\mathrm{sep}}(z_t)$.
12:   Let $\mathcal{D}_{t+1} \leftarrow \mathcal{D}_t \cap \mathcal{H}_t$ where $\mathcal{H}_t$ denotes the halfspace $\{z \mid \langle a_t, z \rangle \leq \langle a_t, z_t \rangle\}$.
13:   Sample $2N$ points $\mathcal{U}_{t+1} := \{z_1^1, \ldots, z_1^{2N}\} \leftarrow \mathrm{SAMPLER}(\mathcal{D}_{t+1}, N, \mathcal{V}_t)$.
14:   Let $\mathcal{V}_{t+1} := \{z_1^1, \ldots, z_1^N\}$ and $\bar{\mathcal{V}}_{t+1} := \mathcal{U}_{t+1} \smallsetminus \mathcal{V}_{t+1}$.
15:  **end if**
16: **end for**
17: **Terminate** and report that $\mathcal{K}$ is empty.

---

**Algorithm 3** SAMPLER used in Algorithm 2

---

**Require:** • Convex set $\mathcal{D}$.
      • Parameter $N$.
      • Points $\mathcal{V} = \{z^1, \ldots, z^N\}$.

1: Let step size $\eta = \Theta(1/\sqrt{d})$, and number of iterations $N' = \widetilde{O}(d^3 N)$.
2: Let $\mathcal{V}' := \{z \in V \mid z \text{ lies in } \mathcal{D}\}$, and define

$$\bar{z} = \frac{1}{|\mathcal{V}'|} z \qquad \text{and} \qquad \Lambda = \frac{1}{|\mathcal{V}'|} \sum_{z \in \mathcal{V}'} (z - \bar{z})(z - \bar{z})^T.$$

3: Let $\mathcal{U} = \varnothing$ and $\widehat{z} \in \mathcal{V}'$ be any arbitrary stating point (note that $\widehat{z} \in \mathcal{D}$).
4: **while** $|\mathcal{U}| \leq 2N$ **do**
5:  Initialize $i \leftarrow 1$.
6:  **while** $i \leq N'$ **do**
7:   Sample $z' \sim \mathrm{Uniform}(\widehat{z} + \eta \Lambda^{1/2} \mathbb{B}_d(1))$.          // Ball walk //
8:   **if** $z' \in \mathcal{D}$ **then**
9:    Update $\widehat{z} \leftarrow z'$ and $i \leftarrow i + 1$.
10:   **end if**
11:  **end while**
12:  Update $\mathcal{U} = \mathcal{U} \cup \{\widehat{z}\}$.
13: **end while**
14: **Return** $\mathcal{U}$.      // Distribution of samples in $\mathcal{U}$ closely approximates $\mathrm{Uniform}(\mathcal{D})$ //

---

horizon $h \in [H]$, the objective in Equation (1) is the optimization problem

$$\widehat{\omega}_{t,h} \leftarrow \underset{\omega \in \mathscr{O}(1)}{\mathrm{argmin}} \sum_{i=1}^{t-1} \left( \langle \omega, \phi(s_{i,h}, a_{i,h}) \rangle - r_{i,h} \right)^2. \tag{27}$$

In the following, we provide a computationally efficient procedure, based off on Algorithm 2, to approximately solve the above squared loss minimization problem given a linear optimization oracle over the feature space (Assumption 6). Note that since $r_{i,h} \in [0,1]$, the constraint on the point $\omega$ implies that the objective value in Equation (27) is at most 2. Thus, we can solve the above

---

**Algorithm 4** Computationally Efficient Implementation of $\mathcal{O}^{\text{sq}}_{\text{apx}}$ for Value Estimation

**Require:** • Data samples $\{(s_i, a_i, u_i)\}_{i \leq t}$.
- Convex domain $\mathscr{O}(W)$.
- Approximation parameter $\varepsilon$.
- Linear optimization oracle $\mathcal{O}^{\text{lin}}$ defined in Assumption 6.

1: // Convert Square Loss Minimization into a Set Feasibility Problem //
2: Define the convex set

$$\mathcal{K}_{\text{APX}} := \left\{ \theta \in \mathbb{R}^d \ \middle| \ \begin{array}{l} \langle \theta, \phi(s_i, a_i) \rangle - u_i \leq \varepsilon \text{ for all } i \leq t \\ \langle \theta, \phi(s_i, a_i) \rangle - u_i \geq -\varepsilon \text{ for all } i \leq t \\ |\langle \theta, \phi(s, a) \rangle| \leq W_h + \varepsilon \text{ for all } s, a \end{array} \right\} \tag{26}$$

3: // Define a Separation Oracle for the set $\mathcal{K}_{\text{APX}}$ using $\mathcal{O}^{\text{lin}}$ //
4: **Definition** $\mathcal{O}^{\text{sep}}_{\mathcal{K}_{\text{APX}}}$ (Input: parameter $\theta \in \mathbb{R}^d$)
- For all $i \leq t$, verify if $-\varepsilon \leq \langle \theta, \phi(s_i, a_i) \rangle - u_i \leq \varepsilon$ for all $i \leq t$.
  - ▸ Output any violating constraint as a separating hyperplane. **Terminate**.
- Then, verify if $\max\{\max_{s,a} \langle \theta, \phi(s, a) \rangle, \max_{s,a} \langle -\theta, \phi(s, a) \rangle\} \leq W + \varepsilon$ using the linear optimization oracle $\mathcal{O}^{\text{lin}}$ (Assumption 6).
  - ▸ If violated, use $\mathcal{O}^{\text{lin}}$ to compute a violating constraint and return it as the separating hyperplane. **Terminate**.
  - ▸ Otherwise, return that the point $\theta \in \mathcal{K}_{\text{APX}}$. **Terminate**.

5: **EndDefinition**
6: // Find a feasible point in $\mathcal{K}_{\text{APX}}$ //
7: Invoke Algorithm 2 to return a feasible point in the set $\mathcal{K}_{\text{APX}}$ with $\mathcal{O}^{\text{sep}}_{\mathcal{K}_{\text{APX}}}$ as the separation oracle.

---

optimization problem upto precision $\varepsilon$, by iterating over the set $\Delta \in \{0, \varepsilon, 2\varepsilon, \ldots, 2 - \varepsilon, 2\}$ in order to solve the set feasibility problem

$$\mathcal{K}^{\Delta}_{\text{APX}} := \left\{ \omega \in \mathbb{R}^d \ \middle| \ \begin{array}{l} \sum_{i=1}^{t-1} (\langle \omega, \phi(s_{i,h}, a_{i,h}) \rangle - r_{i,h})^2 \leq \Delta + \varepsilon \\ |\langle \omega, \phi(s, a) \rangle| \leq 1 + \varepsilon \text{ for all } s, a \end{array} \right\} \tag{28}$$

and stopping at the smallest point $\Delta$ for which $\mathcal{K}^{\Delta}_{\text{APX}}$ has a feasible solution. It is easy to see that for any $\Delta$, either $\mathcal{K}^{\Delta}_{\text{APX}}$ is empty or the shifted cube $\widehat{\omega}_{t,h} + \mathbb{R}_{\infty}(\varepsilon) \subseteq \mathcal{K}^{\Delta}_{\text{APX}}$. Furthermore, under Assumption 7 we also have that $\mathcal{K}^{\Delta}_{\text{APX}} \subseteq \mathbb{R}_{\infty}(R)$ for any $\Delta$. Thus, for any $\Delta$, whenever a feasible solution exists, the set $\mathcal{K}^{\Delta}_{\text{APX}}$ satisfies the prerequisites for Theorem 4, where recall that we can tolerate the parameter $R$ to be exponential in the dimension $d$ or the horizon $H$. Furthermore, a separation oracle $\mathcal{O}^{\text{sep}}_{\mathcal{K}^{\Delta}_{\text{APX}}}$ can be easily implemented by using the linear optimization oracle $\mathcal{O}^{\text{lin}}$ w.r.t. the feature space (Assumption 6) and by explicitly constructing a separation oracle for the ellipsoidal constraint

$$\sum_{i=1}^{t-1} (\langle \omega, \phi(s_{i,h}, a_{i,h}) \rangle - r_{i,h})^2 \leq \Delta + \varepsilon.$$

We provide the implementation of the above in Algorithm 5, which relies on Algorithm 2 for solving the corresponding set feasibility problems. The guarantee in Theorem 4 to find a feasible point in $\mathcal{K}^{\Delta}_{\text{APX}}$ (for each $\Delta$) gives the following guarantee on computational efficiency for Algorithm 5.

**Theorem 7.** *Let $\varepsilon > 0$, $\delta \in (0, 1)$, and suppose Assumption 7 holds with some parameter $R > 0$. Additionally, suppose Assumption 6 holds with the linear optimization oracle denoted by $\mathcal{O}^{\text{lin}}$. Then, for any $t \in [T]$ and $h \in [H]$, Algorithm 5 returns a point $\widehat{\omega}_{t,h}$ that, with probability at least $1 - \delta$, satisfies*

$$\sum_{i=1}^{t-1} (\langle \widehat{\omega}, \phi(s_{i,h}, a_{i,h}) \rangle - r_{i,h})^2 \leq \min_{\omega \in \mathscr{O}(1)} \sum_{i=1}^{t-1} (\langle \omega, \phi(s_{i,h}, a_{i,h}) \rangle - r_{i,h})^2 + \varepsilon \qquad \text{and} \qquad \widehat{\omega}_{t,h} \in \mathscr{O}(1 + \varepsilon).$$

*Furthermore, Algorithm 5 takes $O(\frac{d^7}{\varepsilon} \log(\frac{R}{\delta \varepsilon}))$ time in addition to $O(\frac{d}{\varepsilon} \log(\frac{THR}{\delta \varepsilon}))$ calls to $\mathcal{O}^{\text{lin}}$.*

---

**Algorithm 5** Computationally Efficient Implementation of $\mathcal{O}^{\mathrm{sq}}_{\mathrm{apx}}$ for Reward Estimation

---

**Require:** • Data samples $\{(s_i, a_i, r_i)\}_{i \le t}$.
       • Convex domain $\mathcal{O}(1)$.
       • Approximation parameter $\varepsilon$.
       • Linear optimization oracle $\mathcal{O}^{\mathrm{lin}}$ defined in Assumption 6.

1: **for** $\Delta \in \{0, \varepsilon, 2\varepsilon, \ldots, 2 - \varepsilon, 2\}$ **do**
2:    // Define a Set Feasibility Problem using $\Delta$ //
3:    Define the convex set

$$\mathcal{K}^{\Delta}_{\mathrm{APX}} := \left\{ \omega \in \mathbb{R}^d \,\middle|\, \begin{array}{l} \sum_{i=1}^{t-1} \left( \langle \omega, \phi(s_i, a_i) \rangle - r_i \right)^2 \le \Delta + \varepsilon \\ |\langle \omega, \phi(s, a) \rangle| \le 1 + \varepsilon \text{ for all } s, a \end{array} \right\} \tag{29}$$

4:    // Define a Separation Oracle for $\mathcal{K}^{\Delta}_{\mathrm{APX}}$ using $\mathcal{O}^{\mathrm{lin}}$ //
5:    **Definition** $\mathcal{O}^{\mathrm{sep}}_{\mathcal{K}^{\Delta}_{\mathrm{APX}}}$ (Input: parameter $\omega \in \mathbb{R}^d$)
         • Verify if $\sum_{i=1}^{t-1} \left( \langle \omega, \phi(s_i, a_i) \rangle - r_i \right)^2 \le \Delta + \varepsilon$.
            ▸ If not, return a separating hyperplane for the ellipsoid $\sum_{i=1}^{t-1} \left( \langle \omega, \phi(s_i, a_i) \rangle - r_i \right)^2 \le \Delta + \varepsilon$ w.r.t. $\omega$. **Terminate**.
         • Then, verify if $\max\{\max_{s,a}\langle \omega, \phi(s,a) \rangle, \max_{s,a}\langle -\omega, \phi(s,a) \rangle\} \le 1 + \varepsilon$ using the linear optimization oracle $\mathcal{O}^{\mathrm{lin}}$ (Assumption 6).
            ▸ If violated, use $\mathcal{O}^{\mathrm{lin}}$ to compute a violating constraint and return it as the separating hyperplane. **Terminate**.
            ▸ Otherwise, return that the point $\omega \in \mathcal{K}^{\Delta}_{\mathrm{APX}}$. **Terminate**.
6:    **EndDefinition**
7:    // Find a feasible point in $\mathcal{K}^{\Delta}_{\mathrm{APX}}$ //
8:    Invoke Algorithm 2 with $\mathcal{O}^{\mathrm{sep}}_{\mathcal{K}^{\Delta}_{\mathrm{APX}}}$ as the separation oracle.
         • If succeeded in finding a feasible point $\widehat{\omega} \in \mathcal{K}^{\Delta}_{\mathrm{APX}}$. Return $\widehat{\omega}$ and terminate.
         • Else, continue.
9: **end for**

---

