# OpenReview forum: "Computationally Efficient RL under Linear Bellman Completeness for Deterministic Dynamics"
_ICLR.cc/2025/Conference — ICLR 2025 Oral_

### Official Review · Reviewer_tw1d · 2024-10-24

**Soundness:** 3
**Presentation:** 3
**Contribution:** 4
**Rating:** 8
**Confidence:** 4

**Summary:**

The paper considers the problem of learning in an MDP that is endowed with a property known in literature of _linear Bellman completeness_. Intuitively, this property ensures that, whenever a function is linear in a given feature map, it remains linear after applying the  Bellman optimality operator.

Even if existing algorithms (Eleanor) achieve optimal regret guarantees for this setting, no algorithm is known to solve it with polynomial _computational complexity_ in every parameter.

In this paper, the authors solve the problem in case the process is also deterministic.

**Strengths:**

The paper introduces an algorithm based an a very original and novel idea, that is to perturb the parameters of the algorithm with a noise that is sampled from the null-space of the features spanned up to the given episode. This idea, which the authors call Null Space Randomization, may have a broader impact in the statistical learning community.

The paper also introduces some novelty in the proofs of the statistical complexity bounds. Related to the previous point, the authors introduce a Span Argument: indeed, the new features of the state-action pairs visited in the process do not lie outside of the span of the past features too often. In fact, for every time-step h, this can happen at most $d$ times. Using this simple observation, it is possible to avoid exponential blowout in $H$.
As the problem of exponential blowout in the horizon appears in a wide family of RL with function approximation works, and is usually solved by truncating the value function, which needs to introduce unrealistic assumptions to ensure that truncation does not bring us outside of the function class, I think this is also a major contribution.

Lastly, it is remarkable that the papers deals also with the case of Low Inherent Bellman Error, where the linear model is affected by a misspecification. This scenario, while particularly challenging, is by far more general than assuming exact linearity.

Overall, it is worth mentioning that the algorithm proposed enjoys a regret guarantee which is order-optimal, i.e. scales with $\sqrt T$ (which is the best dependency) in the case of no misspecification. At the opposite side [1], which is perhaps the most recent "competitor" of this work, proves a statistical complexity bound that scales as

$$\varepsilon^{-O(A)},$$

being therefore exponentially dependent on the number of actions and potentially heavily suboptimal even for $A=2$ (due to the $O(\cdot)$). I see this as a major point for this work over the state of the art.


[1]Linear bellman completeness suffices for efficient online reinforcement
learning with few actions.

**Weaknesses:**

The only big weakness of the paper is that the setting is very restrictive, and the contribution does not _at first sight_ seem a big improvement over the state of the art.

1. Assuming deterministic dynamics is very restrictive and, most importantly, does not marry very well with Bellmann completeness. Indeed, given the for of the Bellman optimality operator,
$$\mathcal Tf (s,a)=\int_S p(s'|s,a)\max_{a'\in A}f(s',a')\ ds',$$

One may hope that a well-behaved density function $p(s'|s,a)$ helps the result to approach the span of the features by smoothing everything (a smooth function is indeed easier too approximate with a linear feature map) thus reducing the inherent bellmann error. This does not happen in case of deterministic dynamics, as we just get

$$\mathcal Tf (s,a)=\max_{a'\in A}f(p(s,a),a'),$$

which may be only piecewise smooth due to the maximum.

2. As explained in the paragraph "Deterministic Rewards or Deterministic Initial State", an efficient algortihm was already known for the case where the initial state was fixed; generalizing to the case of random $s_1$ is usually easy for general MDPs. Still, it must be noted that the assumption of linear Bellman completeness does not allow for an easy reduction: just creating a fictitious initial state does not work in this case.

Overall, I think these limitations should not divert the attention for the value of the paper, as they are just a consequence of the hardness of the setting.

**Questions:**

In appendix E: LINEAR MDPS AND LQRS IMPLY LINEAR BELLMAN COMPLETENESS, the authors prove a result about the Linear Quadratic Regulator.

At line 2040, the authors claim that, due to the structure of the problem, the optimal actions writes as $Kx$, where $K$ is some matrix. Even if it is true that in the LQR the optimal policy takes this form, I am not convinced from that passage: indeed, what the authors are showing is that

$$E[\phi(x',\pi^*(x'))^\top \theta] = \phi(x,u)^\top \theta',$$

for some $\theta'$, where $\pi^*$ is the optimal policy. While in fact, what they needed to prove is that

$$E[\min_u \phi(x',u)^\top \theta] = \phi(x,u)^\top \theta'.$$

There is a subtle difference between the two things, as first we are using the optimal policy of the MDP, while in the second case $u$ is given by the best action _with respect to the given $\theta$._ Please, correct me if I am wrong.

---

> ### Author Response · Authors · 2024-11-23
>
> We appreciate your thorough review of our manuscript and your insightful feedback. Your comments that this technique may have broader applications and your point about deterministic dynamics potentially making Bellman backups less ideal, are particularly valuable. In addition, we would like to share our thoughts below.
>
> > Assuming deterministic dynamics is very restrictive and, most importantly, does not marry very well with Bellmann completeness.
>
> We really appreciate you sharing your insight into this problem. Your intuition that deterministic transitions may make the Bellman operator less ideal—making it harder to smooth things so estimation using linear functions may be harder—is insightful. We believe that future research on both deterministic transitions and linear Bellman completeness should carefully consider it.
>
>
> > Still, it must be noted that the assumption of linear Bellman completeness does not allow for an easy reduction: just creating a fictitious initial state does not work in this case.
>
> Thank you for bringing this point! We will make sure to incorporate it.
>
> > At line 2040, the authors claim that, due to the structure of the problem, the optimal actions writes as $Kx$, where $K$ is some matrix. Even if it is true that in the LQR the optimal policy takes this form, I am not convinced from that passage
>
>
>
> Thank you for your comment, and we really appreciate that you pointed this out. We think LQR satisfies Bellman completeness for a convex subset of linear functions. In particular, consider the dynamics of LQR given by:
> $$
> \begin{aligned}
> x' = Ax + Bu + \omega, \quad \text{where} \quad \omega \sim \mathcal N(0, \Sigma).
> \end{aligned}
> $$
> We consider state-action value functions of the follow form (basically the same as what we used):
> \begin{align}
> Q(x, u) = \begin{bmatrix}
> x \\\\ u
> \end{bmatrix}^\top \begin{bmatrix}
> 	P\_{xx} & P\_{xu} \\\\
> 	P\_{ux} & P\_{uu}
> \end{bmatrix}\begin{bmatrix}
> 	x \\\\ u
> \end{bmatrix} + c,
> \end{align}
> It is linear in the quadratic feature $\phi(x, u) = [x^2, u^2, xu, x, u, 1]$. Without loss of generality, we assume that $P\_{xu} = P\_{ux}^\top$; the induced $\theta$ is given by flattening our $P$ (We interchange $\theta$ and $P$ in our proof). As pointed out by the reviewer, we do not have Bellman completeness for *any* such $P$, but under the restriction that $P$ is PSD (note that the set of such $P$ or corresponding $\theta$ will be a convex set). To see this, recall that a matrix:
> \begin{align}
> P = \begin{bmatrix}
> 	P\_{xx} & P\_{xu} \\\\
> 	P\_{ux} & P\_{uu}
> \end{bmatrix}
> \end{align} is PSD iff (1) $P\_{xx}\succeq0$ and $P\_{uu}\succeq0$, and (2) its schur's component $P\_{xx} - P\_{xu} P\_{uu}^{-1} P\_{ux}\succeq 0$.
>
> Then, we can compute the Bellman backup of $Q$:
> \begin{align}
> & \widetilde Q(x, u) \\\\
> &= \mathbb{E}\_{x'} \left[{\min\_{u'} Q(x', u')}\right] \\\\
> &= \mathbb{E}\_{\omega} \left[{\min\_{u'}  \begin{bmatrix}
> 	Ax + Bu + \omega \\\\ u'
> \end{bmatrix}^\top \begin{bmatrix}
> 	P\_{xx} & P\_{xu} \\\\
> 	P\_{ux} & P\_{uu}
> \end{bmatrix}\begin{bmatrix}
> 	Ax + Bu + \omega \\\\ u'
> \end{bmatrix} + c }\right] \\\\
> &= \mathbb{E}\_{\omega} \left[{\min\_{u'}  \left\\{{\left[ Ax + Bu + \omega \right]^T P\_{xx} \left[ Ax + Bu + \omega \right] + 2 \left[ Ax + Bu + \omega \right]^T P\_{xu} u' + u'^T P\_{uu} u'}\right\\}}\right] + c
> \end{align}
>
> Using first-order condition, we know that the optimal $u'$ satisfies
> $$
> u' = - P\_{uu}^{-1} P\_{ux} \left({Ax + Bu + \omega}\right),
> $$
> which implies that
> \begin{align}
> \min\_{u'} Q(x', u') = \left[ Ax + Bu + \omega \right]^T \left[{P\_{xx} - P\_{xu} P\_{uu}^{-1} P\_{ux}}\right]\left[ Ax + Bu + \omega \right] + c
> \end{align}
>
> Plugging the above back into $\widetilde{Q}$, we get
> \begin{align}
> \widetilde Q(x, u) &= \mathbb{E}\_\omega \left[{\left[ Ax + Bu + \omega \right]^T \left[{P\_{xx} - P\_{xu} P\_{uu}^{-1} P\_{ux}}\right]\left[ Ax + Bu + \omega \right] + c }\right]  \\\\
> &=   \left[ Ax + Bu \right]^T \left[{P\_{xx} - P\_{xu} P\_{uu}^{-1} P\_{ux}}\right]\left[ Ax + Bu \right] + c~+ \text{tr}\left({\left({P\_{xx} - P\_{xu} P\_{uu}^{-1} P\_{ux}}\right)\Sigma}\right) \\\\
> &=  \left[ Ax + Bu \right]^T \left[{P\_{xx} - P\_{xu} P\_{uu}^{-1} P\_{xu}^\top}\right]\left[ Ax + Bu \right] + c'  \\\\
> &= \begin{bmatrix} x \\\\ u \end{bmatrix}^T
> \begin{bmatrix} A^T \\\\ B^T \end{bmatrix}
> \left( P\_{xx} - P\_{xu} P\_{uu}^{-1} P\_{xu}^\top \right)
> \begin{bmatrix} A & B \end{bmatrix}
> \begin{bmatrix} x \\\\ u \end{bmatrix} + c'
> \end{align}
> where $c'$ is some constant. The above matrix is PSD since $P\_{xx} - P\_{xu} P\_{uu}^{-1} P\_{xu}^\top \succeq 0$, which holds since $P$ itself is PSD. Thus,
> \begin{align}
> \widetilde{Q}(x, u) = & \begin{bmatrix}
> x \\\\ u
> \end{bmatrix}^\top \widetilde{P} \begin{bmatrix}
> 	x \\\\ u
> \end{bmatrix} + c,
> \end{align}
> for PSD matrix $\widetilde{P}$, and thus the Bellman completeness holds over this convex subset of linear functions for which the corresponding Ps are PSD.

---

> > ### Comment · Reviewer_tw1d · 2024-11-24
> > **Thank you, this nice explanation solves my doubts**
> >
> > I have no further questions.

---

### Official Review · Reviewer_FXbH · 2024-11-03

**Soundness:** 3
**Presentation:** 3
**Contribution:** 3
**Rating:** 8
**Confidence:** 3

**Summary:**

The paper proposes a learning algorithm for linear Bellman-complete MDPs with deterministic dynamics. The algorithm randomizes value function estimates to facilitate efficient exploration of the feature space, a commonly used technique. However, the distinguishing feature of this algorithm is that the random noise is injected only into the null space of the collected feature vectors, selectively exploring previously unvisited regions of the feature space and preventing the value function parameters from growing exponentially with the horizon length. Furthermore, the algorithm employs a random walk-based constrained optimization solver, enhancing the computational efficiency of the method. The derived regret bound for the algorithm is optimal in the number of episodes.

**Strengths:**

* The paper achieves computational efficiency alongside a low regret bound. As the authors point out, computational considerations in MDP learning are crucial for practical applicability, particularly given that many recent works focus only on the statistical analysis of MDP learning.
* The idea of injecting random noise exclusively within the null space of the data matrix is brilliant. The insight that this approach promotes efficient exploration of unexplored feature space without redundancy is well-explained throughout the paper. This strategy may be potentially adapted to settings beyond linear Bellman completeness.

**Weaknesses:**

* To illustrate the computational efficiency of the method, including a simple numerical demonstration would be beneficial.
* One concern is with the presentation style of the paper. The assumption of deterministic dynamics appears to be critical for the regret analysis, yet it is unclear why stochastic dynamics would impede the success of the method. Although it is mentioned that the deterministic nature of the system enables value decomposition as shown in Lemma 10, the necessity of this condition is not explicitly explained in the main body of the paper.

**Questions:**

* There are some typos: e.g., "to the learn parameters" in p.2, and "where the it is" in p.6.
* Why are the constrained squared loss minimization problems first converted to the convex feasibility problem, and then solved by random walk method that depends on a separation oracle? It seems that when $\mathcal{S}$ and $\mathcal{A}$ are infinite, then it is challenging to design an appropriate separation oracle for $\mathcal{K}$. Even when  $\mathcal{S}$ and $\mathcal{A}$ are finite, then each minimization problem has $d$ decision variables and $2|\mathcal{S}||\mathcal{A}|$ constraints, requiring any separation oracle to evaluate the membership for $2|\mathcal{S}||\mathcal{A}|$ inequality constraints.

---

> ### Author Response · Authors · 2024-11-23
>
> Thank you for your valuable feedback! We have fixed the typo you mentioned in the current draft. We will also ensure to improve the paper by elaborating more on the difficulty in stochastic dynamics as per your suggestions.
>
> In addition, we would like to provide our thoughts on your concerns below.
>
> > **One concern is with the presentation style of the paper. The assumption of deterministic dynamics appears to be critical for the regret analysis, yet it is unclear why stochastic dynamics would impede the success of the method. Although it is mentioned that the deterministic nature of the system enables value decomposition as shown in Lemma 10, the necessity of this condition is not explicitly explained in the main body of the paper.**
>
> We will ensure a more comprehensive discussion on this. In summary, the key point is: deterministic transitions are needed in our current analysis due to the *span argument* (see Appendix B, particularly B.1). The span argument divides the proof into two cases based on whether the entire trajectory generated by the current policy ($\pi_t$) falls completely within the data span or not. This step relies on the observation that, under deterministic transitions, our estimates are accurate within the data span--this will not hold under stochastic transition.
>
>
> > **There are some typos: e.g., "to the learn parameters" in p.2, and "where the it is" in p.6.**
>
> Thanks you! We have fixed them.
>
> > **Why are the constrained squared loss minimization problems first converted to the convex feasibility problem, and then solved by random walk method that depends on a separation oracle? It seems that when S and A are infinite, then it is challenging to design an appropriate separation oracle for K. Even when and are finite, then each minimization problem has decision variables and constraints, requiring any separation oracle to evaluate the membership for inequality constraints.**
>
>
> The l_infinity norm of the constraint set in our constraint optimization problem is exponential in H. Thus, to ensure computational efficiency, we need an optimization procedure that can solve this constrained optimization problem in $\text{poly}(d, \log R)$ time where $R$ is the l_infinity norm of the constraint set. The random walk-based approach from Bertsimas & Vempala (2004) gives us this guarantee; however we can easily substitute reliance on their procedure by any other procedure to solve constrained optimization problems whose running time is $\text{poly}(d, \log R)$.
>
> The procedure of Bertsimas and Vempala (2004) needs a separation oracle, and in our setting, this separation oracle can be implemented using a linear optimization oracle over the feature space. Assuming such a linear optimization oracle over the feature space is standard in RL theory. To conclude, your point is correct, our procedure is efficient conditional on solving the constrained optimization problem efficiently, which in turn relies on the access to a linear optimization oracle w.r.t. the feature space; but these assumptions are standard in RL theory.
>
> Additionally, if we are willing to assume bounded $\ell_2$-norm of Bellman backup as commonly imposed in prior works (though we did not make this assumption in our paper), the exponential blow-up will just happen in terms of $\ell_2$-norm. In that case, we can just use these $\ell_2$-norm constraints. Then, the problem is easier to solve (since it does not involve the feature anymore) so no need for the optimization oracle.
>
> We are happy to discuss further in case this does not address your concerns.

---

> > ### Comment · Reviewer_FXbH · 2024-11-27
> > **To Authors' Insightful Response**
> >
> > I appreciate the authors for addressing all my concerns. I have no further comments.

---

### Official Review · Reviewer_Z3Wg · 2024-11-03

**Soundness:** 3
**Presentation:** 4
**Contribution:** 3
**Rating:** 8
**Confidence:** 4

**Summary:**

The paper studies RL under linear Bellman completeness (a generalization of low-rank MDPs) and deterministic transition dynamics. It shows how to design the first algorithm that is both statistically and computationally efficient for this setting, and important open question given the intractability of all existing approaches. The main idea is to drive exploration by injecting noise into the least squares estimates, with the main novelty being that this is done only on the null space of the training data to avoid estimation erros from exploding.

**Strengths:**

1. Designing RL algorithms with function approximation for large state-action spaces that are not only statistically efficient but also computationally efficient is an important open problem, and this paper makes a good step forward in this direction
2. The "span argument" is interesting and novel
3. Although I only quickly went through the proofs, all results seem sound
4. The paper is very well written: the problem and its technical challenges are well motivated from the very beginning, notation is clear, and the adopted techniques are well justified
5. The examples in 3.1 are interesting to motivate why bounded norm assumptions are unreasonable, and further motivate the paper's contributions

**Weaknesses:**

1. The "span argument" is quite interesting, but it also seems quite specific for MDPs with deterministic transitions. Not clear if it extends to stochastic MDPs
2. While a good step forward and a common setting in practice, the assumption of deterministic dynamics may be a limiting factor
3. There is no experiment despite the algorithm being computationally efficient (and, I hope, implementable)
4. Not clear if the dependences on d and H are optimal in the main theorems, or if there is still work to do in that direction

**Questions:**

1. The paper mentions that "Many efficient algorithms can be applied to find approximate D-optimal designs such as the Frank-Wolfe". I think this is the case when S,A are finite, but what if they are infinite? Is there any efficient algorithm?
2. It is not clear why constrained least squares regression, instead of an unconstrained one, is needed since there is no assumption on boundedness of the true parameters. Could you clarify?
3. Even though it acts only on the null space, it was not clear to me why the injected noise (ie \sigma_h) has to be exponential in H
4. Does the algorithm need to know that the MDP has deterministic transitions? In other words, if applied to a stochastic MDP, would the algorithm still make sense or would it break for some reason?

---

> ### Author Response · Authors · 2024-11-23
>
> Thank you for your valuable feedback! We appreciate your insightful questions about the span argument and optimization. We would like to share our thought on these below:
>
> > **Not clear if the dependences on d and H are optimal in the main theorems, or if there is still work to do in that direction**
>
> This is a very good point. While we do not know either, we hope to share what is already known as a reference. We consider two cases:
>
> (1) Known reward: in this case, our regret bound is $\tilde{O}(dH^2)$ (see line 373). A naive lower bound for table MDP with deterministic transition is $\Omega(SAH)$. Since $d = SA$ in the tabular case, there seems to be a gap of $H$ in this case.
>
> (2) Unknown reward: existing results (e.g. proposition 1 in [1]) show a lower bound of $\Omega(dH\sqrt{T})$. This differs from our bound by a factor of $d\sqrt{H}$. However, the lower bound does not assume deterministic transition, so it may not be comparable in general.
>
> We acknowledge the importance of this question and believe it is an interesting future direction.
>
> >**The paper mentions that "Many efficient algorithms can be applied to find approximate D-optimal designs such as the Frank-Wolfe". I think this is the case when S,A are finite, but what if they are infinite? Is there any efficient algorithm?**
>
> Thank you for this important point. It is correct that frank-wolfe is efficient only when SA is finite. However, the running time for frank-wolfe scales as $\text{poly}(d, \log \log |SA|)$, which is tolerate for us for extremely large SA. Furthermore, under slightly sophisticated initializations, this dependence on SA can be completely eliminated. We refer to bullet points 3 and 4 in section 21.2 in the book "Bandit Algorithms" by Csaba Szepesvari and Tor Lattimore (https://tor-lattimore.com/downloads/book/book.pdf) for more details. We will clarify this in the next version of our paper.
>
> Additionally, if we are willing to assume bounded $\ell_2$-norm as commonly imposed in prior works (though we did not make this assumption in our paper), the identity matrix suffices for initialization. So there is no need for a D-optimal.
>
> >**It is not clear why constrained least squares regression, instead of an unconstrained one, is needed since there is no assumption on boundedness of the true parameters. Could you clarify?**
>
> Our algorithm is inspired by RLSVI to inject random noise to encourage exploration and achieve optimism. The key argument here is that the noise must be large enough to cancel out the estimation error. This means, when designing the noise sampling procedure, we have to know an upper bound of the estimation error--this is why we need the parameter to be bounded and hence we solved a constrained regression problem.
>
> Regarding boundedness of parameters: while we cannot control the $\ell_2$ norm of the parameter as argued in Section 3.1, we observe that the parameter is naturally bounded under what we defined as the "$\ell_\infty$-functional norm" (Line 334). So we are still able to control the size of the parameter in terms of this norm. Importantly, this norm property arises as a conclusion of linear Bellman completeness rather than being an additional assumption. And we use this norm in our regression constraints.
>
> >**Even though it acts only on the null space, it was not clear to me why the injected noise (ie \sigma_h) has to be exponential in H**
>
> As mentioned in the previous question, the noise must exceed the estimation error, which can become exponentially large because the parameter norm (of $\theta$) can grow exponentially large--this is because adding noise to the parameter scales its norm multiplicatively. All these blow-ups happen in the null space, so the noise there has to be exponentially large as well.
>
> >**Does the algorithm need to know that the MDP has deterministic transitions? In other words, if applied to a stochastic MDP, would the algorithm still make sense or would it break for some reason?**
>
> Our algorithm does not need to know that the MDP is deterministic, and will run even if the underlying MDP is stochastic. However, our performance guarantees only hold under deterministic transitions.

---

> > ### Comment · Reviewer_Z3Wg · 2024-11-26
> >
> > Thanks for your detailed response. It clarified all my doubts. My initial view was already positive, and the authors' response, together with the other reviews, strengthened this belief. I have thus raised my score and confidence.

---

### Official Review · Reviewer_VJVt · 2024-11-04

**Soundness:** 4
**Presentation:** 4
**Contribution:** 4
**Rating:** 8
**Confidence:** 4

**Summary:**

This paper designs the first computationally and statistically efficient algorithm for linear Bellman complete MDPs with deterministic transitions. Instead of adding a quadratic exploration bonus and truncating the value function, this paper adds carefully designed random noises to the regression results so that it can (a) achieve optimism, and (b) prevent error amplification.

**Strengths:**

-	This paper designs the first computationally and statistically efficient algorithm for linear Bellman complete MDPs with deterministic transitions, which serves as a solid step toward solving the long-standing open question about the statistical-computational gap in the Bellman complete setting.
-	This paper is well written, and the algorithm is straightforward and easy to understand. The technical proofs in the appendix is also well written and easy to follow.
-	This paper clearly discussed the limitations of prior works in the linear Bellman complete setting (Section 3.1 and 3.2). Although the idea of adding noise to the regression result is not new, applying this idea to the specific setting requires non-trivial analysis on the exponential blowup of the norm of the parameters, which is novel in this paper.

**Weaknesses:**

-	My major concern is the organization of this paper --- although the high-level ideas are discussed in detail, all the proofs, including the proof sketch, are deferred to the appendix. I encourage the authors to at least discuss how the key designs of the algorithm (e.g., adding noise to the null space of the current data) enable the regret analysis.
-	One of the major assumptions in this paper is that the transition is deterministic (Assumption 1). However, this assumption is not very well discussed in the paper. The only place where Assumption 1 is used is Lemma 10, which only computes the value decomposition. If I understand correctly, when the transition is not deterministic, a variant of Eq (10) still holds where one only needs to take an expectation w.r.t the randomness of the trajectory on the RHS. Therefore, it is unclear to me what is blocking the analysis from extending to MDPs with stochastic transitions.
-	In the proof sketch (line 885 in particular), the definition of U_t is deferred to Appendix C.2. However, without its definition, it is impossible to understand the proof outline in B.3.
-	Line 1662: typo in the displayed equation, mismatched parenthesis.

**Questions:**

-	Example 1 in Section 3.1 is confusing. First of all, is \mu(s_2) a typo in Line 195? It seems to me that Example 1 proves that, there exists a tabular MDP and a particular feature mapping, such that the norm of the ground-truth parameter has unbounded l2 norm. In other words, this example does not prove that, linear Bellman complete MDPs with unbounded ground-truth parameters is a strict superset of those with bounded ground-truth parameters, since it is still possible that the same MDP can be represented with a different set of features.

---

> ### Author Response · Authors · 2024-11-23
>
> Thank you for your valuable feedback! We have fixed the typo you mentioned in the current draft. We will also ensure to improve the paper by elaborating more on the key design choices and reasoning as per your suggestions.
>
> In addition, we would like to provide our thoughts on your questions below.
>
> > Therefore, it is unclear to me what is blocking the analysis from extending to MDPs with stochastic transitions.
>
> You are right regarding Eq (10). As you pointed out, when the transition is not deterministic, a variant of Eq. (10) still holds, where we take an expectation with respect to the trajectory randomness on the RHS. However, in our proof, we specifically require a version without the expectation. This is needed so that we can divides the proof into two cases based on whether the entire trajectory generated by the current policy ($\pi_t$) falls completely within the data span--this is the *span argument* in our paper (see Appendix B, particularly B.1). This step crucially relies on Eq. (10) being derived without expectation to show that our estimates are accurate within the data span. We will make sure to clarify this point in the next version of the draft.
>
> > Example 1 in Section 3.1 is confusing.
>
> Thanks for checking our examples. \mu(s_2) is not a typo. Here we are using the notation from Linear MDP (so we have $P(s_2|s_1,a_1)= \langle\phi(s_1,a_2),\mu(s_2)\rangle$).
>
> And you are right, we aim to demonstrate the existence of an MDP and a feature mapping that lead to unbounded $\ell_2$ norm. We agree it is possible that a better feature mapping for the same MDP may exist and bound the norm; however, searching for such a mapping is likely a hard problem. Additionally, the MDP in this example does not need to be tabular. Since we have formulated it as a linear MDP, we can append a true linear MDP to it to extend to infinite state space.

---

> > ### Comment · Reviewer_VJVt · 2024-11-26
> >
> > Thank you for your response. I will keep my score.

---

### Meta-Review · Area_Chair_Hrw7 · 2024-12-21

**Metareview:**

The paper proposes a computationally tractable algorithm for the linear Bellman complete setting, addressing a problem that has remained open in the literature for years.
It does so with a randomized algorithm that keeps under control the issue of error amplification.
I, and the reviewers, enthusiastically recommended the paper for acceptance.

**Additional Comments On Reviewer Discussion:**

Minor questions have been addressed

---

### Decision · Program_Chairs · 2025-01-22

Accept (Oral)